# Effects of emissions caps on the costs and feasibility of low-carbon hydrogen in the European ammonia industry

Stefano Mingolla [1] ✉, Paolo Gabrielli [2,3,12], Alessandro Manzotti [4,5,12], Matthew J. Robson [4,12], Kevin Rouwenhorst [6,7,8], Francesco Ciucci [4,9], Giovanni Sansavini [2], Magdalena M. Klemun [10,11] ✉ & Zhongming Lu [1,10] ✉

The European ammonia industry emits 36 million tons of carbon dioxide annually, primarily from steam methane reforming (SMR) hydrogen production. These emissions can be mitigated by producing hydrogen via water electrolysis using dedicated renewables with grid backup. This study investigates the impact of decarbonization targets for hydrogen synthesis on the economic viability and technical feasibility of retrofitting existing European ammonia plants for on-site, semi-islanded electrolytic hydrogen production. Results show that electrolytic hydrogen cuts emissions, on average, by 85% (36%-100% based on grid price and carbon intensity), even without enforcing emission limits. However, an optimal lifespan average well-to-gate emission cap of 1 kg carbon dioxide equivalent ($CO_2e$)/kg $H_2$ leads to a 95% reduction (92%-100%) while maintaining cost-competitiveness with SMR in renewable-rich regions (mean levelized cost of hydrogen (LCOH) of 4.1 euro/kg $H_2$). Conversely, a 100% emissions reduction target dramatically increases costs (mean LCOH: 6.3 euro/kg $H_2$) and land area for renewables installations, likely hindering the transition to electrolytic hydrogen in regions with poor renewables and limited land. Increasing plant flexibility effectively reduces costs, particularly in off-grid plants (mean reduction: 32%). This work guides policymakers in defining cost-effective decarbonization targets and identifying region-based strategies to support an electrolytic hydrogen-fed ammonia industry.

Ammonia is a widely produced chemical primarily for fertilizer production[1-3]. The Haber–Bosch (HB) process is typically used to produce ammonia[4], by combining nitrogen and hydrogen. Steam methane reforming (SMR) is the prevalent method for hydrogen synthesis in Europe[5,6] (Supplementary Fig. 1). SMR emits about 1.6–1.9 tons (t) of carbon dioxide equivalent ($CO_2e$) emissions (hereafter referred to as emissions)[3,7,8], contributing to 36 million metric tons (Mt) of $CO_2$ per year from ammonia production in Europe[9]. Over 85% of this is from SMR-derived hydrogen[1,10]. Decarbonizing hydrogen

production is therefore crucial to reduce the emissions of the ammonia industry and a necessary step to achieve carbon neutrality by 2050[11]. In addition to climate goals, the vulnerability of ammonia production costs to price shocks in fossil fuel markets also motivates a shift away from natural gas[12].

Using renewable energy for water electrolysis is considered a long-term sustainable pathway to produce low-carbon hydrogen for uses like transport, power, and energy storage. While direct electrification is often more cost-effective for road transport[13], electrolytic

hydrogen, and synthesized e-fuels are low-carbon alternatives for shipping[14] and aviation[15], where other options are currently limited. In addition, electrolytic hydrogen can be utilized in Power-to-X systems (converting electricity into other forms of energy or products), Power-to-$H_2$-to-Power systems (converting electricity to hydrogen and then back to electricity when needed)[16], long-term energy storage, and grid stabilization. However, electrolytic hydrogen and e-fuels are secondary energy carriers subject to conversion losses during production and utilization, resulting in overall efficiencies of 10% to 35% and 2-14 times more renewable electricity generation than direct alternatives[17]. Still, the high cost of water electrolysis compared to SMR currently prevents its widespread implementation[11,18–21], with only a small fraction of Europe's hydrogen production currently via electrolysis, at 0.13%[5].

Nevertheless, government funding and technological progress are expected to drive growth in electrolysis-based hydrogen production[11,18–21]. The European Commission has emphasized hydrogen as a key investment priority, projecting a 50-fold increase in electrolyzer installed capacity by 2030, with ammonia plants as the primary users[11]. The disruptions of global fuel supply chains triggered by Russia's invasion of Ukraine in early 2022, drove a significant rise in fossil fuel prices, causing the curtailment of 50% of the ammonia manufacturing capacity in Europe[12,22]. With natural gas constituting more than 80% of production costs, the price of fossil-based hydrogen tripled compared to pre-crisis levels, reaching between 4 and 7 euro per kilogram of hydrogen (EUR/kg $H_2$)[22]. The crisis also impacted the cost of hydrogen derived from natural gas coupled with carbon capture, usage, and storage (CCUS), resulting in a levelized cost of hydrogen (LCOH) ranging from 5 to 7 EUR/kg $H_2$[22]. These circumstances have further highlighted the importance and potential of electrolytic hydrogen to reduce dependency on volatile fossil fuels and enhance energy security, thus accelerating the shift towards electrolysis in industries like ammonia[22].

However, electrolysis powered exclusively by non-dispatchable renewable energy sources cannot meet the continuous, high-volume hydrogen demand of ammonia plants. To substitute SMR with water electrolysis, new system designs must be implemented, including renewable energy generation, electrolyzers, and battery storage (such as lithium-ion batteries). In addition, a surplus of hydrogen produced during peak renewable periods can be stored to balance periods with limited renewable energy supply. Combining solar and wind systems can improve energy consistency[23], but aligning fluctuating renewables with steady industrial processes is a complex challenge that drives cost increases. The levelized cost of ammonia (LCOA) is significantly higher for electrolytic hydrogen (1000–2500 EUR/t of ammonia or $NH_3$) than SMR (200–1000 EUR/t $NH_3$)[22,24]. A lower production cost of clean hydrogen is required for renewable ammonia production to be cost-competitive. Grid electricity can complement renewable resources by increasing operational hours and reducing system installed capacity and costs, especially where renewable conditions are poor[1,25]. Nevertheless, using grid electricity results in carbon emissions, especially in those countries where the electricity mix strongly relies on fossil fuels (the average carbon footprint of the European electricity mix is about 275 g of $CO_2$ per kilowatt hour (kWh) of electricity produced[26]). The production of electrolytic ammonia using a carbon-intensive electricity grid could potentially result in emissions higher than those from SMR[23,27].

The greenhouse gas content of electrolytic hydrogen has become a topic of discussion following the publication of the initial draft of the EU's sustainable finance taxonomy in 2021, which defined sustainable hydrogen as having a well-to-gate $CO_2$e content of less than 3 kg $CO_2$e/kg $H_2$[28]. In June 2023, this initial proposal was revised with the publication of two delegated acts introducing stricter regulations[29]. Renewable hydrogen must be produced exclusively with additional renewable power plants and only when these assets generate electricity (an hourly temporal correlation). Furthermore, the production

should only occur near renewable electricity assets (geographical correlation) (Supplementary Note 1). While these regulations aim to guarantee the sustainability of hydrogen, they could raise costs and limit expansion, possibly hindering the REPowerEU initiative's goals[30].

Research into the economics of electrolytic hydrogen and ammonia production has been extensive, encompassing plant-level, regional, and global analyses. Campion et al.[31] assessed hydrogen systems in three ammonia plants worldwide, finding the most cost-effective strategy combines local renewables with grid electricity, with emissions tied to grid carbon intensity. Nayak-Luke and Bañares-Alcántara[32] expanded the scope to 534 locations worldwide finding that by 2030 many could produce ammonia at costs competitive with fossil fuels, with production flexibility being crucial for reducing expenses.

Operational flexibility has been pinpointed as a method to decrease hydrogen production costs, as outlined by Guerra et al.[33], Wang et al.[23], and Fasihi et al.[34] demonstrated that flexible ammonia plant operations could mitigate costs and overcapacity[23,35]. Despite this, current ammonia production through the HB process exhibits limited adaptability, necessitating technological advancements. An alternative approach involves connecting plants to the electricity grid in a semi-islanded configuration, which can potentially reduce costs. However, this strategy risks increasing emissions unless properly constrained by emission-limiting policies.

Salmon and Bañares-Alcántara[25] studied electrolytic ammonia production in Australia, focusing on the economic and emission implications of grid connectivity. Terlouw et al.[27] investigated semi-islanded hydrogen systems in renewable-rich islands concluding that can be both cost-effective and have a low environmental burden under a specific emission cap. Ricks et al.[36] highlighted a potential pitfall: grid-connected electrolysis, although compliant with US clean-carbon regulation, may increase emissions compared to fossil-based hydrogen unless it is matched hourly with clean energy.

Another set of studies has focused on characterizing region-specific challenges arising from the transition to electrolytic hydrogen. Bartels et al.[37] showed that large-scale electrolysis facilities may have local grid impacts when only relying on grid electricity. Kakoulaki et al.[38] pointed out that despite abundant national renewable resources, regional shortages might arise. Lastly, Gabrielli et al.[39], Rosa and Gabrielli[7], and Tonelli et al.[40] illustrated how, despite global resources exceeding the amount necessary for electrolytic production, local scarcities of land and water may pose significant hurdles.

While previous research has extensively analyzed trade-offs between the technical, economic, and environmental feasibility of electrolytic hydrogen production, the influence of emission caps on these trade-offs has not been considered. Determining low-carbon hydrogen emission standards for the ammonia industry is complex since plants are spread across various regions, each with distinct cost components, electricity prices, grid emissions, and renewable potential. Accounting for this variation is vital to understanding how the stringency of emissions targets affects the size, cost, and land use of low-carbon ammonia plants. However, studies that consider regional variations in renewable energy potential and costs, employ high-resolution analysis of renewable energy profiles and plant operations, and consider future advancements in technology, are currently missing. It is therefore not well understood how regional conditions, including renewables resource profiles and grid emission intensities, shape the relationship between emission standards and costs, particularly as these emissions standards approach zero. Non-linear relationships may lead to outsized costs, grid congestion, extensive renewables curtailment, and land scarcity, impacts that could be avoided with more deliberate, model-informed policy designs.

This study fills this research gap by investigating the effect of increasingly stringent emission caps on system design and operation, hydrogen cost as well as the feasibility of retrofitting existing European

ammonia plants for semi-islanded electrolytic production. The study analyzes hydrogen production across 38 major European ammonia plants (a list of ammonia plants and locations is provided in Supplementary Table 1 and Supplementary Note 2), factoring in regional costs and historical weather data. In doing so, this work provides guidance for policymakers in defining cost-effective decarbonization targets for electrolytic hydrogen production in the ammonia industry context and identifying the regions where the transition to electrolytic hydrogen is both technically feasible and economically viable.

## Results

### Hydrogen production system and model description

This study assumes that the existing European ammonia plants will be retrofitted by replacing the SMR production system with an electrolytic hydrogen production system (hereafter EHPS). The other subsystems of the ammonia plant, including the air separator unit (ASU), the ammonia synthesis loop, and the cryogenic storage for ammonia, will be maintained as they are in the existing facility. Consequently, the EHPS is designed to deliver a continuous hydrogen supply thanks to on-site hydrogen storage in pressurized tanks and grid backup, ensuring uninterrupted operation of the ammonia plant, which requires steady-state conditions (for detailed information, see "Ammonia production process and EHPS").

The EHPS includes solar photovoltaic panels (PV), wind turbines (WT), electrolyzers, battery energy storage systems (BESS), hydrogen compressors, and high-pressure tanks for hydrogen storage (Fig. 1). Two EHPS configurations are modeled: (i) when the EHPS is mainly powered by renewable resources but also maintains a grid connection for backup, it is defined as semi-islanded configuration with hybrid (PV and WT) renewable power generation. (ii) When the plant is islanded or off-grid, operating entirely independently of any grid connection, relying solely on its renewable power generation (Fig. 1).

It is assumed that new dedicated solar PV and WT will be installed near the ammonia plant, within the same region, including dedicated transmission lines and accounting for transformer and transmission losses (see "Grid connection upgrades"). This assumption aligns with the requirements for low-carbon hydrogen highlighted by the European Union Renewable Energy Directive II delegated act[29] (Supplementary Note 1). Here, the term region is defined according to the NUTS-2 level of the Nomenclature of Territorial Units for Statistics (NUTS) system. Each European region is assigned a distinct four-letter code in accordance with the NUTS-2 classification. For example, the

Norwegian region of Sør-Østlandet is coded as NO03, while the Italian region of Emilia-Romagna is coded as ITH5. A detailed list of the regions covered in this study, along with their corresponding unique NUTS-2 code, can be found in Supplementary Table 1.

The EHPS is modeled and optimized for all 38 ammonia plants based on historical weather data and regional cost components. The objective is to minimize the lifetime system cost and, therefore, the levelized cost of hydrogen (LCOH) produced over the 2025–2050 timeframe for each of the 38 major European ammonia plants. The decision variables are the design and operation of the EHPS under different input parameter assumptions (see "Optimization model").

The effect of increasingly stringent emissions reduction targets on the LCOH is tested under various emission caps on the lifespan well-to-gate $CO_2e$ content of hydrogen: (i) <3 kg $CO_2e$/kg $H_2$ (3-cap, from the EU taxonomy[28]), (ii) <1 kg $CO_2e$/kg $H_2$ (1-cap), (iii) <0.5 kg $CO_2e$/kg $H_2$ (0.5-cap), (iv) <0.1 kg $CO_2e$/kg $H_2$ (0.1-cap), and (v) 0 kg $CO_2e$/kg $H_2$ (0-cap). These caps are chosen to cover the proposed certification and regulations for hydrogen (Supplementary Table 2). The 0-cap refers to an ammonia plant not connected to the electricity grid (off-grid or islanded) and where the electrolysis system is 100% powered by renewables. In addition, a no-cap scenario is assessed, representing no specific emission reduction targets, and includes any emissions greater than the 3 kg $CO_2e$/kg $H_2$ threshold.

Emission caps are set to limit the average well-to-gate $CO_2e$ emissions from the hydrogen produced throughout the lifetime of the plant (hereafter simply emissions) in the optimization models (see "Carbon emission caps"). Well-to-gate emissions encompass Scope 1 (direct emissions from operations, negligible in electrolytic production), Scope 2 (indirect greenhouse gas emissions from the generation of purchased electricity), and partial Scope 3 emissions (upstream activities like the extraction, refining, and transport of fuel used for electricity production, hereafter Scope 3 upstream emissions) (see "Grid electricity price and carbon intensity"). Hence, following EU regulations, the emissions from the grid considered in the study are based on the average carbon intensity of electricity consumed in the Member State where the fuel is produced[41]. Other Scope 3 emissions, such as those embedded in technology manufacturing (hereafter Scope 3 embedded emissions), are excluded, aligning with most low-carbon hydrogen certification systems (Supplementary Table 2) and the emission accounting framework for low-carbon hydrogen outlined by the International Partnership for Hydrogen and Fuell Cells in the Economy (IPHE)[22].

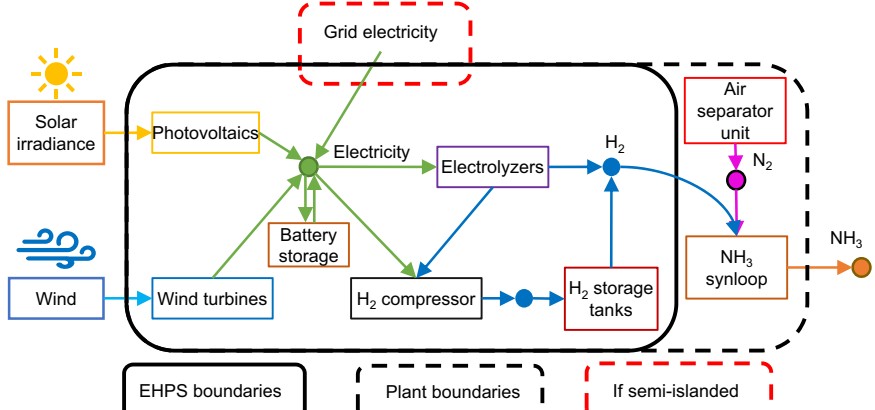

**Fig. 1 | System boundaries for the electrolytic hydrogen production system (EHPS) and the ammonia plant.** The EHPS is primarily powered by newly dedicated renewable installations (semi-islanded with grid connection for backup), with the 0-cap scenario being an off-grid exception with no grid electricity import. This study focuses on the EHPS, accounting for over 85% of energy consumption and emissions in the electrolytic hydrogen production process. Green, blue, pink, and orange lines represent electricity, hydrogen ($H_2$), nitrogen ($N_2$), and ammonia ($NH_3$) flows. Details regarding ammonia plant operation and subsystems in "Ammonia production process and EHPS". The ammonia plant is assumed to operate continuously at full load. Hence, the EHPS must supply a constant volume of hydrogen to the ammonia synloop.

The well-to-gate emissions from electrolytic hydrogen production are compared with the ones from SMR. The direct emissions (Scope 1) of hydrogen production from natural gas through SMR are around 9–10 kg of $CO_2$/kg $H_2$[22]. Further emissions occur in natural gas production, processing, and transport. Scope 3 upstream emissions for natural gas can vary widely on a country base, depending on production methods and emission mitigation efforts (median value 2.4 kg $CO_2$e/kg $H_2$[22]). Therefore, considering direct and upstream emissions, the total well-to-gate emissions from hydrogen production via SMR are 12 kg $CO_2$e/kg $H_2$.

Future electricity prices for industrial users are projected using historical data and simulated through Geometric Brownian Motion (GBM) and Monte Carlo methods. Instead, the future carbon intensity of grid electricity is determined based on upcoming EU targets for the carbon intensity of electricity production in Europe (see "Grid electricity price and carbon intensity").

Three cases are defined to provide a comprehensive understanding of hydrogen production costs, accounting for the uncertainty of the model's input parameters, such as future electricity price and equipment cost. First, a pessimistic case is investigated wherein the input parameter values are from the lower end of the cost and performance estimates gathered from the literature, resulting in the highest LCOH. Conversely, an optimistic case is also defined by considering the values of each parameter that result in the lowest LCOH. Finally, a reference case is examined by calculating the mean of the uncertain parameters based on the available literature (see "Methods"). In this study, the results presented refer to the reference case, unless otherwise specified. A total of 684 optimizations were conducted for the main analysis, considering 3 cases (optimistic, reference, and pessimistic), 6 emission caps (including the no-cap case), and 38 ammonia plants.

## Electrolytic $H_2$ can reach SMR costs except for the 0-cap

The mean LCOH for electrolytic hydrogen across Europe is 3.90 EUR/kg $H_2$ when no emission caps are enforced (no-cap), 3.97 EUR/kg $H_2$ for the 3-cap, 4.13 EUR/kg $H_2$ for the 1-cap, 4.23 EUR/kg $H_2$ for the 0.5-cap, 4.58 EUR/kg $H_2$ for the 0.1-cap and 6.34 EUR/kg $H_2$ for the 0-cap (Supplementary Table 3). The lowest LCOH is with a semi-islanded configuration in Norway (Sør-Østlandet region or NO03) with 1.99 EUR/kg $H_2$, while the highest LCOH is 12.61 EUR/kg $H_2$ for off-grid plants in Slovakia (Západné Slovensko region or SK02). This range aligns with recent estimates from the International Energy Agency (IEA)[22] (2–10 EUR/kg $H_2$ in Europe). Assuming continuous operation of ammonia plants, these values translate into an ammonia production cost ranging from around 700 EUR/t $NH_3$ under less stringent emission caps to 1200 EUR/t $NH_3$ for off-grid plants. These estimates are consistent with cost projections found in other studies[22].

Throughout a 25-year plant lifetime, the total cost of the EHPS, comprising both capital expenditures (CAPEX) and operating expenditures (OPEX), averages 6.7 billion EUR for the no-cap scenario, with OPEX making up 55% and CAPEX 45% of the total (Supplementary Fig. 2). In contrast, for the 0-cap scenario, the total cost rises to ~10.8 billion EUR, with a higher proportion attributable to CAPEX at 65% and a smaller portion to OPEX at 35%. These estimates align with anticipated investments for major proposed projects in renewable-based ammonia production, which range between 4 and 11 billion EUR[42–45].

The cost of imported electricity is the most significant expense under both the no-cap and 3-cap scenarios, comprising 30% and 25% of the total cost, respectively (Supplementary Fig. 2). Consequently, in the absence of any emission thresholds (no-cap), the lowest LCOH is recorded in Norway and Poland (1.99 EUR/kg $H_2$ in NO03; 2.98 EUR/kg $H_2$ PL42; and 3.05 EUR/kg $H_2$ PL61), mainly driven by the below-average price of electricity. The weight of grid import diminishes rapidly with the enforcement of stricter emission caps. Despite regional variations in local cost components and renewable capacity factors, a direct

correlation exists between the LCOH and the average price of grid electricity in the region where the plant is situated. However, this correlation weakens as the emission caps become more stringent, dropping from an r-squared value of 0.55 in the no-cap scenario to 0.01 in the 0.1-cap scenario (Supplementary Fig. 3).

More stringent emissions targets generally increase the LCOH (Fig. 2a for the reference case and Supplementary Table 3 for pessimistic and optimistic), but this increase is particularly pronounced for the 0-cap. The mean LCOH for the 0-cap is 63% higher than the case without emission constraints, and 38% higher than the 0.1-cap. In addition, more stringent targets also result in a broader uncertainty range in the LCOH due to the larger variation in the required installed capacity of the EHPS components. To better illustrate similarities across locations, European ammonia plants can be clustered based on the grid characteristics (cost and carbon intensity) of the corresponding NUTS-2 regions (Fig. 2b).

Plants in regions with cheap, low-carbon grid electricity, such as Alsace (FRF1, France), Sør-Østlandet (NO03, Norway; highlighted in Fig. 2b), and Aragón (ES24, Spain), experience negligible increases in LCOH with a more stringent emission cap. NO03 is the only region where electrolytic hydrogen is estimated to be cost-competitive with SMR hydrogen produced at 2.7 EUR/kg $H_2$ in 2021 in Europe (in August 2022, costs of SMR hydrogen reached 10 EUR/kg $H_2$[46]), which oscillates between 1.4 and 1.8 EUR/kg $H_2$[46]. However, a 0-cap brings about a sharp increase in the LCOH. For example, the ammonia plant in Sør-Østlandet (NO03) presents the lowest LCOH of all plants, with above-average use of grid electricity (54% of the electricity comes from the grid); here, grid electricity has the lowest carbon intensity in Europe and lower price compared to the mean value. However, when the plant is off-grid, the installed capacity of wind turbines and electrolyzers increases by 369% and 417%, respectively, to balance the lack of grid backup, resulting in a 236% increase in LCOH.

Plants in regions with cheap but carbon-intensive grid electricity, such as Severozápad (PL81, Poland; Fig. 2b) and Lubelskie (PL81, Poland), tend to cover a significant portion of their energy demand with grid electricity when no emission caps are imposed. In this case, the share of grid electricity is gradually reduced with a more stringent emission cap, whereas installed capacities and LCOH gradually increase.

Finally, plants in regions with a high grid electricity price, e.g., Tees Valley and Durham (UKC1, Fig. 2b), consume less grid electricity as installing renewable infrastructure is more economical. Hence, there is a relatively small difference in LCOH across the emission caps.

These results corroborate previous research conducted at select sites in Europe[27] and the United States[47], which found that producing hydrogen via water electrolysis, when powered by a combination of dedicated renewable energy sources with a grid backup for uninterrupted plant operations, is generally more cost-effective than using exclusively additional renewable energy sources and produces fewer carbon emissions than relying solely on grid electricity. The degree of these benefits is influenced by factors such as the cost of electricity from the grid, the grid's carbon intensity, and the availability of renewable resources.

By assessing the implications of increasingly stringent emission targets, this research enhances the comprehensive understanding of the variability in LCOH across different European locations. The subsequent sections detail the findings, which are essential for defining suitable emission standards necessary for guiding the ammonia industry's shift toward low-carbon electrolytic hydrogen production.

## The largest installations are required to meet the 0-cap

Capital costs of the electrolyzers and wind turbines are the second and third largest cost components, which range on average between 14% and 12%, respectively, for the no-cap scenario and increase to 18% and 21% for the 0-cap scenario (Supplementary Fig. 2). With increasingly

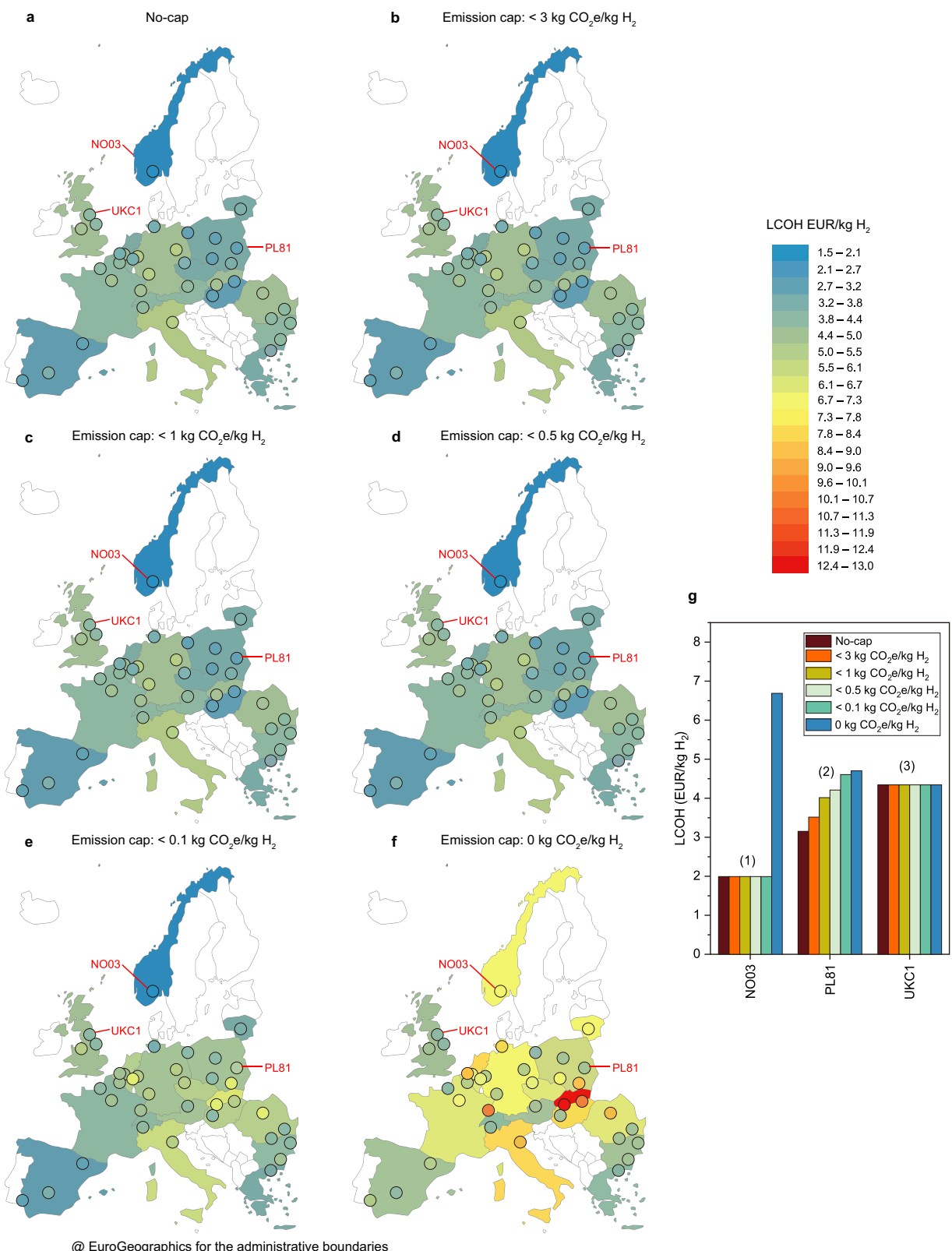

@ EuroGeographics for the administrative boundaries

stringent caps, the installed capacity of these components exhibits exponential growth (Fig. 3 and Supplementary Fig. 4). Under the 0-cap, the installed capacity is nearly double that of the average installed capacity under the 0.1-cap. This trend is particularly pronounced in wind-dominated regions. The installed capacity of wind turbines is estimated to approximately triple from the 0.1-cap to the 0-cap scenario, while the installed capacity of electrolyzers quadruples. PV

installations, however, grow more steadily from the no-cap to the 0.1-cap scenarios. On average, PV systems constitute 57% of the renewable power capacity for semi-islanded configurations, whereas this share drops to 46% for islanded (off-grid) systems. This result stems from the difference between solar and wind resource profiles. Under the 0-cap, the consistent availability of wind energy leads to an increased use of wind installations, reducing the reliance on solar

**Fig. 2 | Minimum levelized cost of hydrogen (LCOH) across Europe.** Circles indicate the minimum LCOH (unit: euro per kilogram of hydrogen, EUR/kg $H_2$) for each ammonia plant given emission constraints (caps) on the lifespan well-to-gate carbon dioxide equivalent ($CO_2$e) content of hydrogen ($H_2$) (unit: kilogram $CO_2$e per kilogram $H_2$, kg $CO_2$e/kg $H_2$). Country-level results were obtained by averaging the LCOH of the individual plants within a country. Pessimistic and optimistic results are presented in Supplementary Table 3. **a** No emission cap (no-cap). **b** For all plants, a 3 kg $CO_2$e/kg $H_2$ emission cap (3-cap). **c** In all, 1 kg $CO_2$e/kg $H_2$ emission cap (1-cap). **d** In all, 0.5 kg $CO_2$e/kg $H_2$ emission cap (0.5-cap). **e** In all, 0.1 kg $CO_2$e/kg $H_2$ emission cap (0.1-cap). **f** In all, 0 kg $CO_2$e/kg $H_2$ emission cap (0-cap). **g** Ammonia plants are clustered based on the grid characteristics of the corresponding regions. Three examples are given: Sør-Østlandet coded as NO03, Lubelskie coded as PL81,

and Tees Valley and Durham coded as UKC1 in NUTS-2 (Nomenclature of Territorial Units for Statistics level 2). Ammonia plant locations were primarily sourced from the comprehensive map of major fertilizer plants in Europe provided by Fertilizer Europe[103]. These locations were then cross-referenced and validated with information obtained from the websites of key producers, including Yara, Fertiberia, BASF, and Borealis, as well as checked against the European Commission[70] and Fuel Cell Hydrogen Observatory (FCHO)[5] databases to identify any additional steam methane reforming (SMR) hydrogen production sites not reported in the initial data (see Supplementary Note 2). Geospatial data for the NUTS-0 (Nomenclature of Territorial Units for Statistics level 0) and NUTS-2 European regions were obtained in the form of shapefiles from Eurostat, the statistical office of the European Union[104].

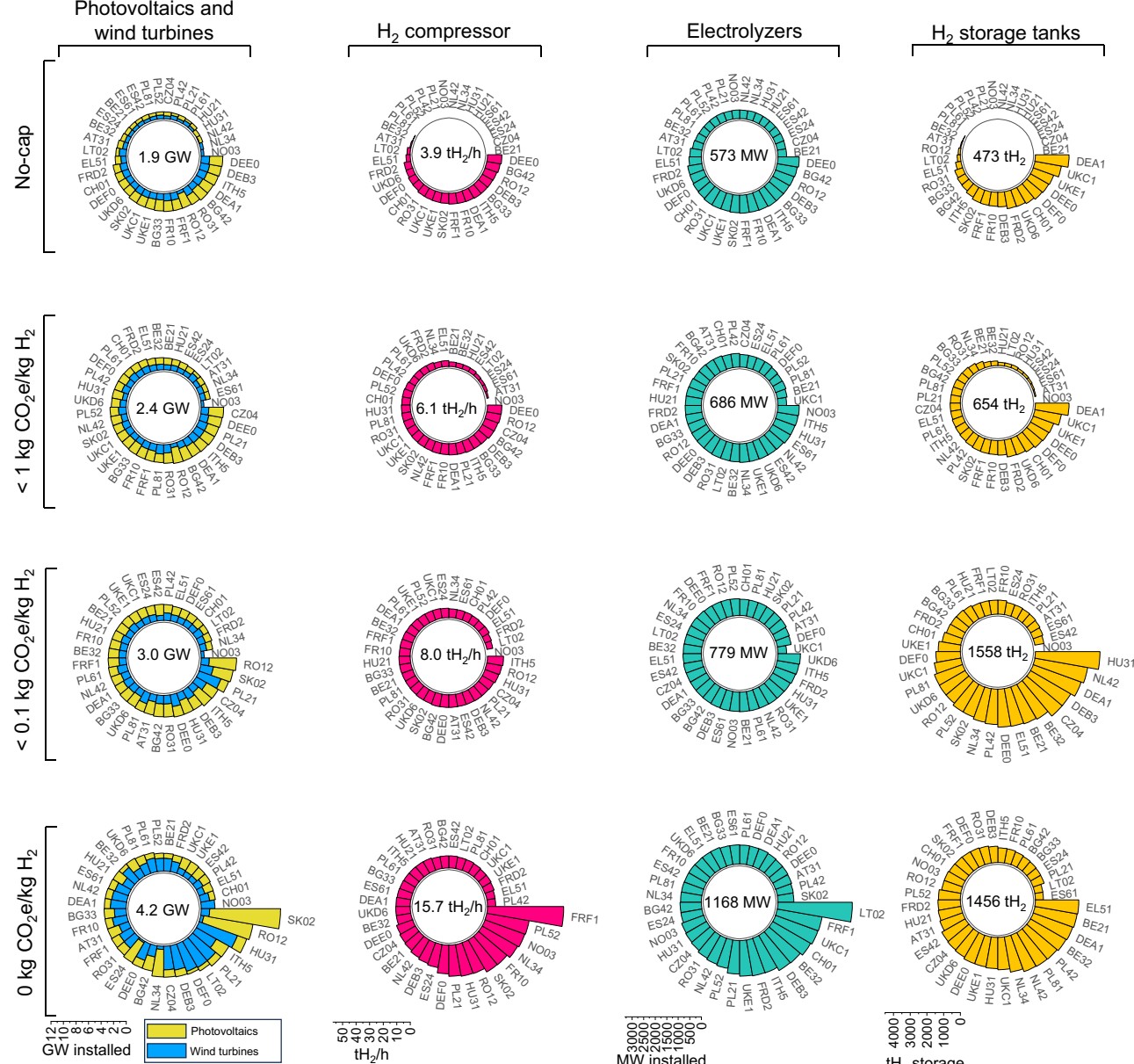

**Fig. 3 | Optimal electrolytic hydrogen production system (EHPS) design across Europe.** Regions coded according to the NUTS-2 level of the Nomenclature of Territorial Units for Statistics (NUTS) system (Supplementary Table 1) and ordered by installed capacity. Values inside the spirals refer to the mean installed capacity of the respective component in European plants given emission constraints (caps) on the lifespan well-to-gate carbon dioxide equivalent ($CO_2$e) content of

hydrogen ($H_2$) (unit: kilogram $CO_2$e per kilogram $H_2$, kg $CO_2$e/kg $H_2$). Overall, the installed capacity of various EHPS components, including photovoltaics and wind turbines (unit: Gigawatt, GW), hydrogen ($H_2$) compressor (unit: tons $H_2$ per hour, t$H_2$/h), electrolyzers (unit: Megawatt, MW), and $H_2$ storage tanks (unit: tons of $H_2$, t$H_2$) exhibits exponential growth with increasingly stringent caps.

energy. Consequently, costs in wind-dominated regions surge significantly more than in solar-rich areas. For instance, Norway's NO03 region (Sør-Østlandet) shows a 236% cost increase from the 0.1-cap to the 0-cap scenario. Meanwhile, Greece's EL51 region (Eastern Macedonia and Thrace) sees a mere 10% increase in cost from the no-cap to the 0.1-cap scenario and an even smaller 1% increase from the 0.1-cap to the 0-cap.

Compressed hydrogen is favored over Li-ion batteries. Although a significant cost reduction in utility-scale batteries is forecasted in the coming years[48], hydrogen storage is estimated to be cheaper for large-scale applications, as also previously shown by other studies[49–51]. One notable observation is that the greater availability of wind energy throughout the day reduces the need for storage in wind-rich regions compared to regions primarily reliant on solar energy.

Generally, despite variations across regions, off-grid plants necessitate significantly larger installations compared to semi-islanded plants without any emission caps (Fig. 3 and Supplementary Fig. 4). The increase in infrastructure is substantial: on average, photovoltaic (PV) power capacity expands by 75%, wind turbines by 177%, electrolyzers by 105%, compressors size (kg $H_2$/h) by 304%, and storage tanks volume by 208%. These figures underscore the substantially higher resource requirements for fully off-grid setups (Supplementary Fig. 4). These results align with findings from other research. For instance, Campion et al.[31] demonstrated that in Northern Chile, a region dominated by solar energy, the installed capacity of PV and electrolyzers is more than twice as high for off-grid plants compared to semi-islanded configurations.

The costs associated with retrofitting the hydrogen system constitute a relatively minor part of the total expenditure, accounting for ~1% of the overall cost (Supplementary Fig. 2). These expenses include the upgrading and replacement of specific components (such as electric start-up heaters and steam generators) to ensure optimal operation with electrolytic hydrogen. They also involve enhancements of transmission lines, which are needed to accommodate an increase in grid demand. Furthermore, the sunk costs associated with decommissioning the SMR hydrogen production system are included in these retrofitting expenses (see "Retrofitting costs").

Increasingly stringent emission caps not only affect the optimal design and costs of the EHPS but also affect the renewables curtailment rate. The curtailment rate increases exponentially with more stringent emission targets due to the larger over-capacities installed for a limited number of low-resource hours per year. For instance, the curtailment rate rises from 14% with no emission cap, to 22% with a 0.1-cap, and to 39% under a 0-cap when the plant operates off-grid (Supplementary Fig. 4). An increase in curtailment rate is typically observed during peak renewable energy production periods when the plant cannot handle the excess energy.

## Electrolytic $H_2$ reduces emissions even without emission caps

Without any emission target, on average, 32% of the EHPS's total annual energy demand comes from the grid (Supplementary Fig. 4). The reason for the low use of grid electricity import, even when emission caps would permit increased usage, is due to the lower levelized cost of electricity (LCOE) from newly installed renewable energy sources as compared to that of grid electricity. For example, an EHPS fully powered by grid electricity, using the average European grid electricity price of 115 EUR per megawatt hour (MWh), would result in an LCOH exceeding 7 EUR/kg $H_2$, which is 75% higher than the average LCOH of semi-islanded plants. Most European ammonia plants exhibit similar LCOH under both the 3-cap and the no-cap. In fact, 30/38 plants emit less than 3 kg $CO_2$e/kg $H_2$ with electrolytic hydrogen production, even without the imposition of emission caps. In regions outside Europe where the energy mix is more carbon-intensive and less expensive, higher emission caps might be more appropriate to study, as the impact on costs and emissions could be more pronounced and thus more relevant.

Replacing SMR with water electrolysis for all the European ammonia plants results in an average emission reduction of about 85%, even without enforcing any emission caps attributable to the average well-to-gate carbon content of hydrogen being 1.85 kg $CO_2$e/kg $H_2$ (Fig. 4a and Supplementary Table 4). Deep emission reductions without stringent targets are not achieved in all regions. Plants in regions with cheap and carbon-intensive grid electricity achieve significantly lower emission reductions than the European average. An example of this is the Małopolskie (PL21, Poland) plant, which shows only a 36% emissions reduction while importing 67% of its energy demand from the grid (PL21, Fig. 4b). As regulations become more stringent, the fraction of energy that powers the EHPS imported from the grid decreases. The grid reliance drops to 16% under the 1-cap, and predictably to zero when the plant operates off-grid (Supplementary Fig. 4). Grid imports are prevalent during prolonged periods with suboptimal renewable energy conditions.

There is an average 2% increase in LCOH when the first emission cap is enforced (3-cap), resulting in a 90% emission reduction compared to hydrogen from SMR. Tightening the emission cap from 3-cap to 1-cap is associated with an average LCOH increment of 4%, facilitating a further 6% reduction in emissions. When the cap is further tightened, decreasing from 1- to 0.5-cap, there is, on average, a 2% increase in LCOH to achieve a 97% emission reduction. From the 0.5- to the 0.1-cap, there is, on average, an 8% increase in LCOH to reach a 99% emission reduction. Finally, to eliminate the final 1% of emissions, an additional 41% increase in LCOH is observed. In other words, eliminating the final 1% incurs the highest cost. For some plants, this last-mile emissions reduction results in a dramatic increase in costs: Sør-Østlandet (NO03, Fig. 4c), Schleswig-Holstein (DEF0), and Alsace (FRF1) present a 236%, 128%, and 116% increase in LCOH from the 0.1 to the 0-cap, respectively.

Large-scale renewable installations, in the gigawatt range, involve substantial direct and indirect land use, especially when stringent regulations demand significant over-capacities to offset periods of suboptimal renewable energy production (see "Renewable power generation"). The average area allocated to renewables varies from 160 square kilometers ($km^2$) under the no-cap scenario to 406 $km^2$ for the 0-cap scenario, marking a 153% increase (Supplementary Fig. 4). Despite the considerable size, the direct impact on the land is smaller, given that the direct land usage is ~25% of the entirety of the designated area. This allows for the possibility of multi-purpose land use, such as agrophotovoltaics. Nevertheless, it underscores the sheer scale of such projects, particularly under stringent emissions caps.

## 1-cap is a feasible cost-effective emission reduction target

The cost-effective emission cap is identified based on the abatement cost (AC) implied by different caps. The AC is calculated by dividing the difference between the LCOH of electrolytic hydrogen and the LCOH of SMR hydrogen (reference LCOH being 2 EUR/kg $H_2$) by the difference in their carbon content. In essence, this ratio quantifies the additional expenditure per ton of $CO_2$e abated (see "Abatement cost"). The AC is computed for all European plants, and the industry-wide average is determined. The lowest AC in Europe is under the 3-cap, with 183 EUR/t $CO_2$e abated, closely followed by the 1-cap with 187 EUR/t $CO_2$e abated, approximately half of the AC for the 0-cap (362 EUR/t) (Supplementary Fig. 6 and Supplementary Table 5).

More stringent caps induce higher AC, except for regions with carbon-intensive grid electricity. For example, Polish region PL42 records an average LCOH of 2.98 EUR/kg $H_2$ under no-cap, and a 52% reduction in emissions compared to hydrogen produced via SMR. The AC is 152 EUR/t $CO_2$e. While a stricter 3-cap causes a cost increase of only 7%, avoided emissions increase by 44% compared to the no-cap scenario, and the estimated AC drops to 132 EUR/t $CO_2$e. The trend observed in Poland is a result of the large increase in avoided emissions

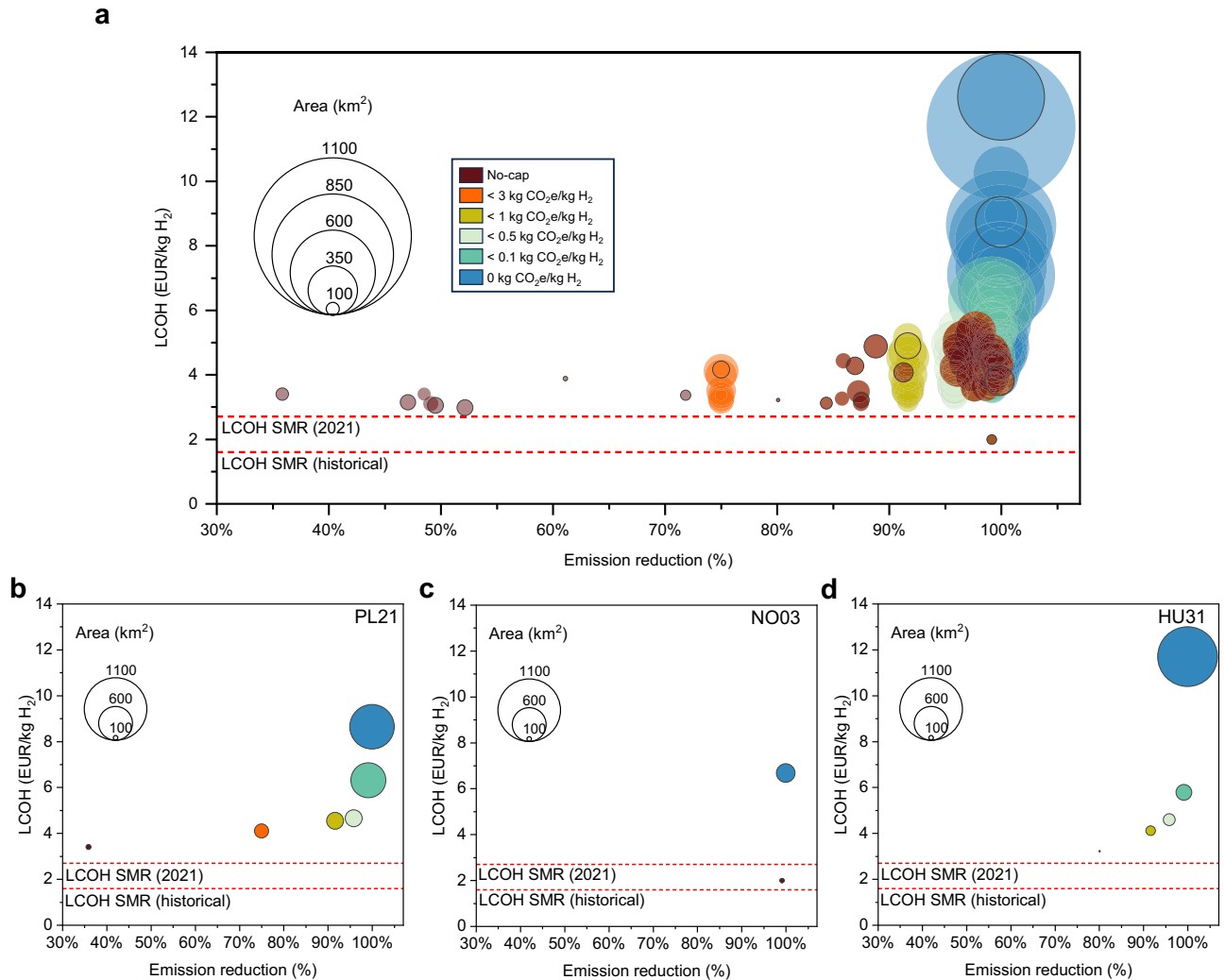

**Fig. 4 | Cost-effective emission cap. a** Minimum levelized cost of hydrogen (LCOH, in euro per kilogram of hydrogen, EUR/kg $H_2$) and area (unit: square kilometers, km$^2$) needed for system installation (size of circles) as a function of carbon emission reduction for all ammonia plants in Europe, under different emission constraints (caps) on the lifespan well-to-gate carbon dioxide equivalent ($CO_2$e) content of hydrogen ($H_2$) (unit: kilogram $CO_2$e per kilogram $H_2$, kg $CO_2$e/kg $H_2$). Pessimistic and optimistic cases in Supplementary Fig. 4. **b** Małopolskie in Poland coded as PL21 in NUTS-2 (Nomenclature of Territorial Units for Statistics level 2) of European regions. **c** Sør-Østlandet in Norway coded as NO03. **d** Észak-Magyarország in Hungary coded as HU31. Excluding the 0-cap, the emission reduction is not constant over time due to the projected decarbonization of national electricity (Supplementary Fig. 5).

compared to the relatively small increase in cost when the first cap is implemented, ultimately leading to a lower AC.

On average, the 3, 1, and 0.5-caps exhibit comparable AC due to a balance between cost and emission reductions. Stricter caps cost more but reduce more emissions. On the other hand, less stringent caps lead to smaller emission reductions but are associated with a lower average LCOH. As deep decarbonization of the industry is a pivotal target, this study investigates which emission cap can achieve the most significant emission reduction compared to the no-cap scenario with the smallest associated cost increase. As Supplementary Fig. 6 demonstrates, enforcing the 3-cap results in an additional 6% emission reduction compared to SMR production. Under this 3-cap scenario, the average carbon content is 1.25 kg $CO_2$e/kg $H_2$. Interestingly, the implementation of a 1-cap further cut these emissions by 51%, corresponding to a modest additional cost increase of 2% from the 3-cap. This results in an average carbon content of hydrogen of 0.62 kg $CO_2$e/kg $H_2$, thereby ensuring that all plants emit less than 1 kg of $CO_2$e per kg of $H_2$ over their operational lifetime. The 1-cap thus emerges as a potentially effective strategy for achieving considerable emission reduction with a tolerable increase in cost.

## Robustness analysis

A robustness analysis is performed to verify the consistency of the results under varying input conditions, as the input parameters of the optimization model can vary within a broad range of values that reflect both current and future uncertainty. The parameters tested (Fig. 5, $y$ axis) are: (i) price of grid electricity, (ii) electrolyzer price, efficiency, maintenance costs, and lifetime, (iii) solar PV price, (iv) wind turbines price, and (v) carbon intensity of grid electricity.

The analysis was conducted by consecutively assigning pessimistic, and optimistic values (Fig. 5, $x$ axis) to each parameter while keeping the other parameters at their reference values. To reduce the computational effort of testing all possible climate conditions found in all geographical regions, five representative regions were identified (see "Representative regions"), each with extreme weather conditions including (i) wind-dominated, (ii) solar-dominated, (iii) low-capacity, (iv) median-capacity, and (v) high-capacity (both solar and wind) regions. The introduction of representative regions with identical input parameters, except for the capacity factor of solar and wind energy, enables a more generalized understanding of where hydrogen

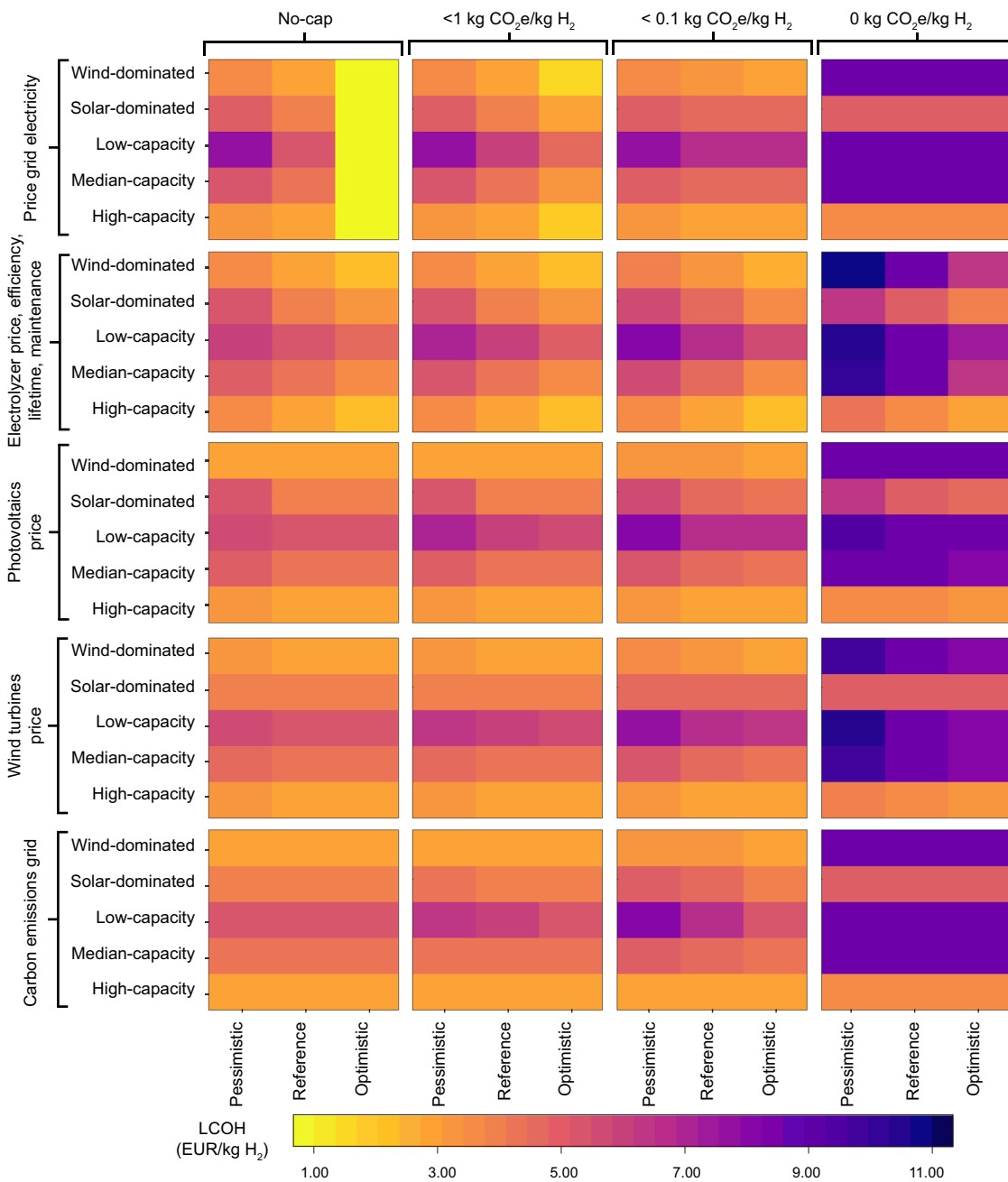

**Fig. 5 | Robustness analysis of levelized cost of hydrogen (LCOH, in euro per kilogram of hydrogen, EUR/kg H₂) sensitivity to input variations.** The investigated variables include: (i) grid electricity price, (ii) electrolyzer cost, efficiency, maintenance expenses, and lifespan, (iii) solar photovoltaic system cost, (iv) wind turbine cost, and (v) carbon intensity of grid electricity (y-axes). The range of values for these parameters is provided in Supplementary Table 6. The robustness is tested under different emission constraints (caps) on the lifespan well-to-gate production appears to be more cost-effective, thereby assisting in explicating the regional results.

carbon dioxide equivalent (CO₂e) content of hydrogen (H₂) (unit: kilogram CO₂e per kilogram H₂, kg CO₂e/kg H₂) for the five representative regions (High-capacity, Median-capacity, Low-capacity, Solar-dominated, and Wind-dominated). The analysis confirms that the steepest increase in LCOH and installed capacity occurs from the 0.1 kg CO₂e/kg H₂ (0.1-cap) to 0 kg CO₂e/kg H₂ (0-cap) despite variations in input parameters.

A total of 300 optimizations were additionally performed for the robustness analysis, considering the five input parameters, three cases for each input parameter (optimistic, reference, and pessimistic), four emission caps (no-cap, 1-cap, 0.1-cap, and 0-cap), and five representative regions. The robustness analysis demonstrates that while variations in input parameter values can affect both LCOH and the optimal system design (Fig. 5), the main trends and conclusions remain valid. Specifically, the steepest increase in

hydrogen cost and installed capacity occurs from the 0.1- to the 0-cap.

Wind energy is preferable for hydrogen production at continuous output, leading to lower LCOH. Regions characterized by very high wind capacity but poor solar energy yield similar LCOH results to regions with both high solar and wind capacities, albeit to a lesser degree. It is observed that high wind capacity can compensate for poor solar capacity; however, the reverse is not true—high solar capacity cannot make up for low wind capacity. Interestingly, under a 0-cap scenario, wind-dominated regions experience a 160% increase in costs

compared to the 0.1-cap scenario, while solar-dominated regions only witness an 81% cost increase. This suggests that while wind energy is generally favorable for hydrogen production, the cost implications under the most stringent emission cap are less favorable for wind-dominated regions compared to solar-dominated ones. The reason is the need for large wind installations (more expensive than PV per power capacity) and electrolyzers.

Among the input parameters evaluated, the price of grid electricity is found to be the most impactful under less stringent emission caps, with its significance diminishing under more stringent caps. Under the no-cap scenario, high electricity prices result in an average 21% increase in LCOH (Supplementary Table 7). This increase is primarily driven by a 39% rise in low-capacity regions, which cannot leverage inexpensive renewable generation, compared to an 8% increase in high-capacity regions. Conversely, low electricity prices result in an average reduction in LCOH of 73%, a trend that is more homogeneous across all regions. Low electricity prices would lead to predominantly using grid electricity, eliminating the need for additional renewable installations when optimizing costs without any emission constraints.

The second most influential parameter in determining the LCOH is the cost and performance of the Alkaline (ALK) electrolyzer, with it causing a variation in LCOH of approximately ±20% (Supplementary Table 7), depending on the specific case under consideration. Given the substantial impact of electrolyzer costs and performance on the LCOH, further analyses were conducted to delve deeper into this relationship and potentially identify strategies for optimizing these parameters to enhance the economic viability of hydrogen production.

For this study, ALK electrolyzers, were selected given their maturity and widespread use. However, other alternative electrolyzer technologies may become commercially viable in the future (see "Electrolyzers"). To account for this, additional analyses have assessed two additional electrolyzer technologies. The first model represents a low-cost but less efficient electrolyzer (i.e., membrane-less (ML): 54% cheaper but also 29% less efficient than ALK[52,53]), while the second represents a more expensive yet highly efficient electrolyzer (i.e., solid oxide electrolyzer (SOE): 280% more expensive and 17% more efficient than ALK[6]).

Despite their superior efficiency, the significantly higher costs associated with SOE electrolyzers result in a higher LCOH across all regions and emission cap scenarios, with increases ranging from 7% to 43% (Supplementary Table 8). The lowest increase, 7%, is observed in low-capacity regions, where higher efficiency can help to reduce the larger installed capacity of renewable infrastructure as well as reliance on grid import. Conversely, in high-capacity regions and in regions dominated by wind energy, the deployment of SOE simply leads to a higher LCOH. This can be attributed to the fact that the high renewable potential results in smaller installation sizes. As such, the benefits of increased efficiency cannot offset the higher costs of the more efficient electrolyzers. Similarly, electrolyzers that are less efficient but also less expensive (ML), generally result in a higher LCOH. However, the pattern here is opposite to SOE. A less efficient electrolyzer implies a higher energy demand, which in turn necessitates larger renewable installations. This effect is particularly pronounced in low-capacity regions, where it results in a 28% cost increase. On the other hand, regions with high renewable capacity, either solar or wind, might experience a slight (1%) reduction in LCOH under the 0-cap scenario. This happens because in these regions, under the most stringent emission cap, the electrolyzer capacities are exponentially larger than under the less stringent caps, allowing the lower electrolyzer costs to offset the impact of reduced efficiency.

To summarize, despite the current options either being too expensive or having low efficiency, in general, more efficient electrolyzers provide higher benefits in regions with low renewable

energy availability. This is primarily because the enhanced efficiency can leverage the reduction in renewable installations needed. Conversely, cheaper electrolyzers can reduce the LCOH in renewable-rich regions, particularly under stringent emission caps as these regions with plentiful renewables can deploy more capacity at lower costs, thus allowing the lower equipment costs to offset the impact of lesser efficiency. Therefore, the choice of electrolyzer technology should be carefully matched to the local conditions, particularly the availability of renewable resources and the emission cap in place.

The reference permit prices from the European Union Emissions Trading System (EU-ETS) were used to study the effect of carbon pricing on estimated hydrogen costs. With a reference price of 86 EUR/t $CO_2e$, the LCOH increases by an average of 1.85% with no emission constraint and by 1.28% under the 1-cap scenario (Supplementary Table 9). Conversely, with the same value for the EU-ETS, the price of hydrogen produced by SMR increases by 52%, to an average cost of 3.03 EUR/kg $H_2$. This stark difference underscores the sensitivity of SMR-produced hydrogen to carbon pricing. In contrast, the effect of high EU-ETS costs on electrolytic hydrogen production is modest, especially under stringent emission caps.

Lastly, the feasibility of flexible plant operation was explored. As detailed in "Ammonia production process and EHPS", ammonia plants typically operate at full capacity to satisfy the steady-state conditions required by the Haber–Bosch process. It is therefore assumed a continuous hydrogen supply to the synthesis loop thanks to hydrogen storage and grid backup to prevent operational disruptions. Recent industry efforts, however, have been directed towards investigating the potential for more adaptable plant operations, specifically aligning ammonia production with the variable output of intermittent renewable energy sources. The extent of operational flexibility is directly correlated with the capability to adjust to fluctuating energy inputs. This flexibility is limited by the least flexible component within the system (i.e., air separator unit and ammonia synloop), suggesting that constraints on any single technology's operations can limit the flexibility of the entire ammonia plant.

Technological innovations are making ammonia plants more adaptable to variable power inputs. Electric heaters, variable load compressors, and better catalysts for ammonia synthesis allow for quicker adjustments to power input changes, more manageable load variations, and improved operational ramp-up and ramp-down. Future electrolytic ammonia plants could operate more efficiently and with greater flexibility than today's standard.

In light of these developments, the impact of partially relaxing the hourly output constraint of the EHPS was tested. Specifically, the plants were assumed to operate down to a 50% minimum load while maintaining the same total annual hydrogen production, as detailed in "Robustness analysis". Results indicate that even partially flexible ammonia production can lead to cost reductions, attributable to the downsizing of renewable energy installations, grid import, and hydrogen storage. The extent of cost reduction varies, ranging from 6% under the no-cap to 32% when operating off-grid (0-cap). The observed decrease in LCOH for flexible off-grid operations closely aligns with the estimated 25–40% reduction previously reported in case studies in Australia, Argentina, and Chile[23,31,50]. The reduction is especially pronounced in regions dependent on wind energy, with a reduction of up to 46%, while solar-dominated regions experience a smaller impact, with a 19% reduction (Supplementary Table 10). This result is significant as it demonstrates that increasing the flexibility of the plant can partially mitigate the substantial cost increments encountered under a 0-cap (off-grid) scenario. These preliminary findings underscore the importance of prioritizing plant flexibility in the design of next-generation plants as a strategic approach to decrease the production costs and land requirements of off-grid electrolytic ammonia plants.

The robustness analysis revealed the impact of grid electricity price, electrolyzer technology selection, and flexible operation of ammonia production on the LCOH across a variety of emission caps and different degrees of renewable energy availability. These insights deliver a thorough assessment of the economic viability for grid-connected (non-zero caps) and off-grid (0-cap) electrolytic hydrogen and ammonia production. This enhanced understanding fills a gap in the literature[23] and informs strategic decision-making and policy development for sustainable transition towards net-zero electrolytic hydrogen and ammonia production.

### Challenges in transitioning to electrolytic hydrogen

Additional analysis was conducted to evaluate the practical feasibility of shifting from SMR to EHPS under various emission caps. The analysis considered three key factors: additional grid capacity needed, renewable energy needs, and land requirement in each plant's region (Fig. 6a) (Supplementary Table 11). Such requirements were compared with the availability of corresponding resources (see "Feasibility analysis").

Results show that, under the 1-cap, the EHPS mean grid electricity demand is 491 gigawatt hour (GWh) per year (min 26 GWh, max 1488 GWh across Europe). The shift to electrolytic hydrogen production does increase the electricity demand compared to the SMR process (on average, 56 MWh are imported from the grid each hour). While this increased electricity consumption necessitates infrastructure upgrades on the plant side, as previously discussed, the plant's grid energy demand is relatively small, representing, on average, 6.3% of the current energy demand at the regional level and 0.8% at a national level. This increase is unlikely to have a significant impact on regional consumption patterns, causing grid congestions, or to influence price dynamics notably[37]. According to the EU's 2030 and 2050 targets, it is anticipated that future expansions of the grid will be predominantly accommodated by a greater proportion of renewable energy sources, complemented by widespread adoption of utility-scale storage technologies[54].

Nevertheless, four plants in Hungary (HU21 and HU31), the Netherlands (NL34), and Greece (EL51) would result in a local demand increase of over 15%, with the Dutch plant in Zeeland (NL34) potentially reaching up to 24.5%. While the overall energy demand of the grid may not be significantly affected, the peak power demand during operation can be substantial[37] (see "Feasibility analysis"). There is a need to investigate how regional or local grids will adapt to the increased demand of the EHPS. This includes the necessary expansion of power generation capacity and the adjustment of pricing policies to ensure that hydrogen production via electrolysis remains cost-effective. In these analyses, the marginal grid emission factors should be used in optimizing the expansion of power generation capacity and grid dispatch to accurately evaluate and minimize the emissions resulting from the EHPS-induced demand[55,56].

In terms of renewable energy requirements, the ammonia plant would require, on average, 1.7% of the technical renewable potential of the country and 27% of the region (i.e., the maximum electricity generation that can be produced by renewables[38]). Under the 0-cap, this percentage grows to 44%. Notably, while under the 1-cap only one region (NL42) would exceed the technical renewable potential, this number increases to five regions under the 0-cap scenario. This indicates that while transitioning to renewable hydrogen production for ammonia is feasible in many areas, it could present challenges in

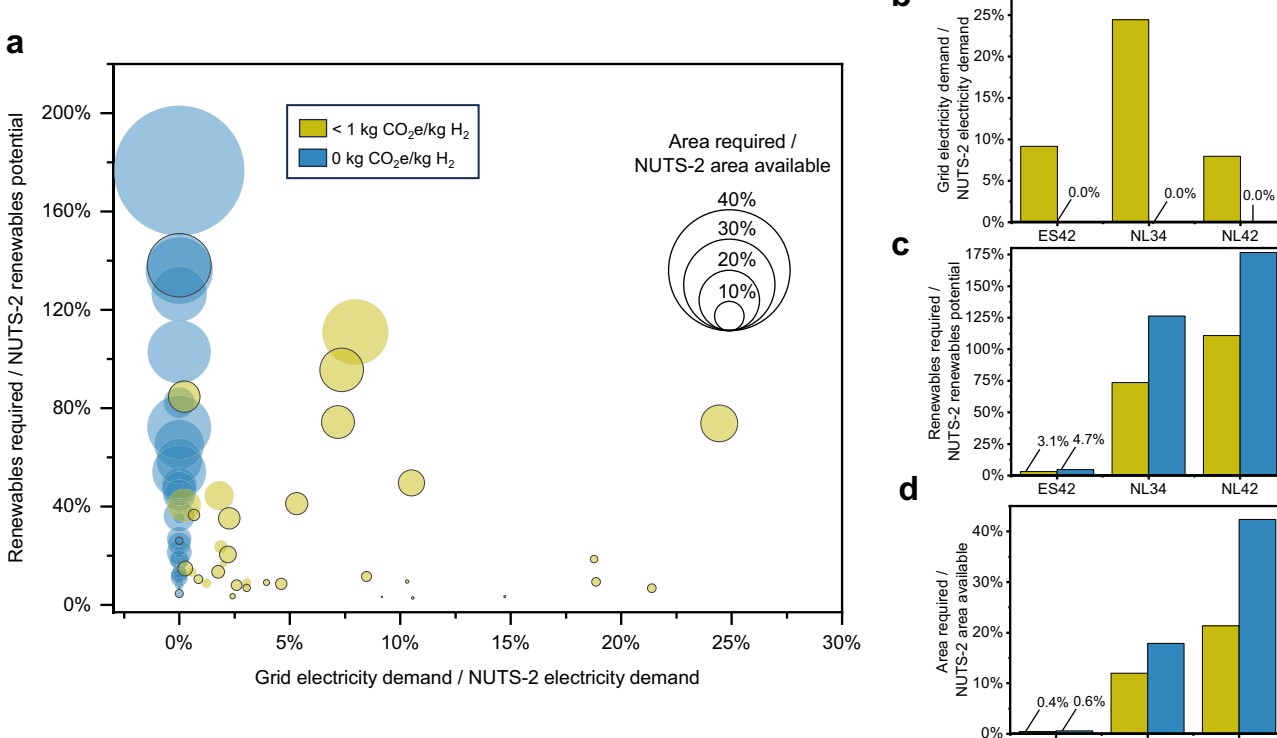

**Fig. 6 | Feasibility analysis. a** x axis: impact on grid electricity demand; y axis: renewable requirements vs. renewable potential; Circles area: land requirements for renewable installations vs. available. Analysis conducted at Nomenclature of Territorial Units for Statistics, level 2 (NUTS-2) for the average scenario under two emission constraints (caps) on the lifespan well-to-gate carbon dioxide equivalent ($CO_2$e) content of hydrogen ($H_2$) (unit: kilogram $CO_2$e per kilogram $H_2$, kg $CO_2$e/kg $H_2$): 1 kg $CO_2$e/kg $H_2$ (1-cap) and 0 kg $CO_2$e/kg $H_2$ (0-cap). **b** Focus on grid electricity demand. **c** Renewable requirements vs. renewable potential. **d** Land requirements for renewable installations vs. available. **b–d** Show three ammonia plants located in (i) Castilla-La Mancha coded as ES42, (ii) Zeeland coded as NL34, and (iii) Limburg coded as NL42 according to NUTS-2.

regions with lower renewable potential, particularly under stringent emission caps and when considering future renewable demand from other, yet-to-decarbonize industries.

In addition, the installation would occupy, on average, 5.2% (min 0.4%, max 21.4%) of the available land in the region under the 1-cap. The computation of available land adopted the methodology described in ref. 39 utilizing country-specific land data provided by the Food and Agriculture Organization of the United Nations (FAO), as detailed in "Methods". Under the 0-cap, the results show an average 87% increase in available land occupancy compared to the 1-cap to 9.6% (min 0.6%, max 42.4%). A strong direct correlation ($r$-squared = 0.85) exists between renewable potential and available land for renewables, meaning either value can effectively predict the other. More available land generally indicates higher renewable potential, and vice versa.

These results point to land area required for renewable installations as one major area of concern when considering the production of hydrogen from renewables, since ammonia plants require massive installations (on average 206 km$^2$ for the 1-cap and 405 km$^2$ for the 0-cap), with 1.4 and 2.5 times these capacities in the low-renewable regions. As a reference, the median size of an urban area with a minimum population of one million in Europe is 300 km$^2$. In comparison, the largest solar park in the world (Bhadla Solar Park in India), has a nominal capacity of 2245 MW and covers an area of 56 km$^2$, while the Asian Renewable Energy Hub in Australia, currently under planning, will span over 6500 km$^2$ for 15 GW solar and wind installations.

However, multi-functional land use can be integrated to maximize land utilization. One example of multi-functional land use is agrophotovoltaics, which involves the integration of agriculture and photovoltaic systems. The cost of land has been factored into the LCOH calculations (as detailed in "Methods"), representing on average 5% (no-cap) to 7% (0-cap) of the total EHPS cost. It should be noted that multi-functional land use like agrophotovoltaics has the potential to serve as a revenue stream, thereby reducing overall costs.

The results of the feasibility analysis indicate that the transition is technically feasible in almost all European ammonia plants (35/36 for the 1-cap and 31/36 for the 0-cap), despite requiring a significant amount of land for renewable energy installations, particularly under stricter emission caps. This may present local challenges in regions with limited land availability or that require larger renewable installations (e.g., Zeeland–NL34 and Limburg–NL42, highlighted in Fig. 6b–d). Other regions, instead, present favorable conditions to produce electrolytic hydrogen. Castilla-La Mancha (ES42, Fig. 6b– d), for example, presents a combination of high renewable potential (smaller installations are needed) and abundant land, making it an optimal location for low-carbon hydrogen production. These findings align with those produced by Tonelli et al.[40], whose global study also identified similar outcomes. They particularly noted land availability constraints in certain regions, underscoring the shared challenges in the transition to renewable energy sources.

## Discussion

This study extensively investigates the impact of emission targets for hydrogen synthesis for ammonia production by identifying the minimum cost of producing hydrogen via water electrolysis for different decarbonization targets across 38 European ammonia plants. Results show that transitioning from SMR to semi-islanded electrolytic hydrogen powered by dedicated renewables installed near the plant can cut emissions from hydrogen synthesis, on average, by 85%, even without setting an emission cap. However, without any standards, some plants in regions with cheap and carbon-intensive electricity may continue causing substantial carbon pollution despite the transition to electrolytic production. Within this context, the 1 kg CO$_2$e/kg H$_2$ (1-cap) is identified as the optimal emission cap for the European ammonia industry, with an average hydrogen cost of 4.2 EUR/kg H$_2$, cutting emissions by on average 95% compared to SMR.

Even though electrolytic hydrogen can mitigate significant variability in operational costs caused by fluctuations in natural gas prices and may even be cost-competitive with SMR in regions with clean, inexpensive grid electricity and abundant wind energy (e.g., NO03 or Sør-Østlandet, Norway), it is, on average, more expensive than SMR-produced hydrogen. Policy support could help close this gap through subsidies and tax benefits. An example is the European Hydrogen Bank initiated by the European Commission, which has set 800 million EUR for subsidies tied to renewable hydrogen production, provided certain criteria are met. This research indicates that with well-defined emission caps and adequate financial incentives, ammonia plants have the potential to initiate their hydrogen decarbonization processes. This proactive approach allows them not to rely solely on the broader grid's expansion and decarbonization efforts to meet the growing renewable electricity demand from multiple sectors.

However, more stringent regulations like the 0-cap lead to substantial cost increases when maintaining continuous production without any grid backup, rendering the subsidies insufficient to offset the cost hike. Land requirements associated with large-scale renewable energy installation may also pose challenges. There is a significant increase in system component size under the 0-cap, which results in an average 133% increase in the land area required. This is particularly challenging in some regions with limited land availability. Hence, stringent emission targets may hinder the phase-out of fossil fuel-based hydrogen synthesis in energy-intensive industries, in contrast with the European strategy to quickly scale up electrolytic hydrogen production. To summarize, although 100% decarbonization is the ultimate emission target for the ammonia industry, it is challenging to implement at this point without disrupting the entire European supply chain.

It is crucial to underscore that a 0-cap scenario not only implies higher costs and increased land usage for renewables but can also potentially lead to higher overall life cycle emissions compared to scenarios with less stringent caps when accounting for emissions associated with the manufacturing of components such as electrolyzers, wind turbines PV, and hydrogen storage systems. Present policies do not account for Scope 3 embedded technology emissions from component manufacturing, and this study therefore also omits these emissions. However, under the 0-cap scenario, where renewable installations are significantly larger, the total carbon content of hydrogen—when accounting for emissions from component manufacturing (including Scope 3)—is 33% higher compared to the life cycle emissions under the 1-cap scenario, given global average emissions components as shown in Supplementary Fig. 7 (details about the calculation of Scope 3 embedded emissions in "Scope 3 embedded emissions" and Supplementary Note 3). As we look to the future, the cleaner the grid of the EU becomes, the more significant the relative contribution of component manufacturing to overall emissions will become. This suggests that future research could usefully adapt this model to inform the next generation of policy regulations, considering the full life cycle emissions of hydrogen production systems.

To reduce costs, land requirements, and life cycle emissions associated with hydrogen production at off-grid plants, it is essential to prioritize increasing plant flexibility. Enhanced flexibility allows plants to align production more closely with the availability of renewable energy, thus reducing the need for extensive renewable generation installations (such as PV and wind turbines) and storage capacity. Although current ammonia plants offer limited operational flexibility, focused efforts from industry and targeted government research and development can stimulate advancements in this area. Enhanced plant flexibility could lead to improved competitiveness of islanded electrolytic plants and support the achievement of deep decarbonization goals in regions with favorable renewable resources, especially with high wind potential.

Conducting the analysis at the regional level can help tailor emission caps and similar policies to local conditions. Some countries present a combination of favorable conditions for the deployment of electrolytic hydrogen that can allow near-zero emissions while bearing minimal or absent cost increases compared to fossil-based hydrogen production. Policymakers may agree to set more stringent decarbonization targets for some countries, while others may be subjected to less stringent or delayed measures. Regionally diversified and phased policy approaches can avoid excessively penalizing local industries. European governments may also decide to support strategies other than producing renewable electricity in the vicinity of current ammonia plants. Due to the heterogeneity of electrolytic hydrogen costs, it is imaginable that some ammonia producers may consider locating renewable infrastructure in regions with a higher renewable potential to reduce costs. However, this could potentially deplete local renewable resources. Further investigations are warranted to ascertain the regions and conditions under which ammonia plants might be given precedence for the use of limited renewable resources over other sectors.

Another viable strategy could involve retaining the current plants while directly importing low-carbon hydrogen from regions with a superior renewable potential and lower production costs. The large-scale transportation of hydrogen would necessitate substantial pipeline infrastructure, introducing additional carbon emissions, costs, and risks, factors that could be incorporated into the model. For a detailed regional analysis, it is essential to utilize precise, high-resolution geospatial data to investigate strategies for providing low-carbon, cost-effective electrolytic hydrogen. This work could serve as a foundation for future studies to determine whether and how regions capable of producing low-cost hydrogen could meet the demand from other areas. The study could be extended beyond Europe, considering a global supply chain.

The study can be further improved in multiple aspects to formulate rigorous emission cap regulations for sustainable investment in low-carbon electrolytic hydrogen for ammonia production. First, an analysis of how varying timeframes for measuring $CO_2e$ emissions, from hourly to yearly, influence the LCOH is crucial. Hourly emission caps may be challenging in industrial processes that require continuous output, and a dedicated investigation is needed. Second, investigating the potential of flexible ammonia production is essential. This approach could more efficiently harness intermittent renewable energy, influencing both LCOH and the determination of cost-effective emission caps. Third, the performance of the optimization model (see "Optimization model") can be enhanced by incorporating robust optimization techniques to better handle the uncertainties associated with renewable energy variability. In this study, robustness analyses were instead performed to verify the consistency of the main findings (see "Robustness analysis"). Finally, the study should be extended to other hard-to-abate sectors with heavy hydrogen usage. The feasible cost-effect emission caps for these hard-to-abate sectors can be different from the findings for ammonia production in Europe, and these differences should be thoroughly investigated to formulate sector-based emission regulations.

## Methods

### Study area and time horizon
This study focuses on the major ammonia plants in Europe that produce hydrogen on-site through large-scale SMR systems (Supplementary Table 1). There are 39 major ammonia plants located in 37 regions across 19 countries (Supplementary Note 2) (note that the ammonia plant in Croatia is not included in the analysis due to a lack of capacity factor data; therefore, only 38 plants are considered). The time horizon ranges from 2024 up to December 31, 2050. Each ammonia plant is assumed to substitute the SMR system with an EHPS in 2024, starting operations on January 1, 2025. Therefore, the LCOH

(EUR/kg $H_2$) refers to the electrolytic hydrogen produced from January 1, 2025, until December 31, 2050 (8760 h/year * 26 years = 227,760 h).

EHPS component costs and features (efficiencies, lifetimes) were collected from the literature and industrial/governmental reports. When multiple sources were available for the same year, the most recent was prioritized. If a reference began its projections before or after 2024, the 2024 values were estimated using linear extrapolation from the nearest value. If publications only provided values for specific years, linear interpolation was used to fill in values for in-between years to create annual projections.

### Optimization model
Single-objective optimization models are used to identify the optimal design (technology selection and size) and operation (e.g., imported grid electricity, stored energy/hydrogen, etc.) of each EHPS to minimize the LCOH at the ammonia plant level. Figure 7 summarizes key inputs for this optimization model.

In energy system modeling, mixed-integer linear programming (MILP) has emerged as the predominant optimization approach for the design and operation of multi-energy systems[25,31,57,58]. MILP stands out due to its ability to effectively solve systems of linear equations, while including nonlinearities via binary variables, thus ensuring a trade-off between computational efficiency and solution robustness[25,58–60]. A MILP was formulated in Python with the Gurobipy package[61] and solved with the commercial solver Gurobi[62].

The objective function is to minimize the lifetime system cost (Eq. (1)), while the decision variables are the design (technology selection and size) and operation (amount of imported grid electricity, hourly storage operations) of the ammonia plants under different input parameter assumptions.

$$\min z_{cost} = \sum_{k \in K} I_k + \sum_{k \in K} v_k I_k + \sum_{l=1}^{L} \sum_{t=0}^{T} p_E\left(M_{E,t}\right) \quad (1)$$

where $z_{cost}$ represents the total lifetime cost of the EHPS, consisting of the sum of capital cost $\sum_{k \in K} I_k$, operation and maintenance cost (O&M) $\sum_{k \in K} v_k I_k$, and grid power purchase $\sum_{l=1}^{L} \sum_{t=0}^{T} p_E(M_{E,t})$. In detail, $L$ is the lifetime of the plant in years (25 years), $T$ is the number of hours per year (8760), $I_k$ is the installation cost of technology $k$, where $k$ belongs to the set of technologies $K$; $v_k$ is a fraction of the installation cost for annual maintenance, $p_E$ is the price of grid electricity for industrial end-users, and $M_{E,t}$ is the quantity of imported electricity from the grid. Further details regarding cost calculation are reported in Supplementary Eq. (1) in Supplementary Note 4.

$$LCOH = \frac{z_{cost}}{\sum_{l=1}^{L} \sum_{t=0}^{T} D_{H2,t}} \quad (2)$$

The LCOH is computed through (Eq. 2). The LCOH represents the average cost per unit of hydrogen produced, considering the expected operational lifetime of the production facility. It is assumed that the total amount of hydrogen produced $\sum_{l=1}^{L} \sum_{t=0}^{T} D_{H2,t}$, where $D_{H2,t}$ is the hourly output of the EHPS, remains consistent regardless of the chosen assumptions regarding input parameters. This allows for a meaningful comparison of LCOH across diverse scenarios. Note that this equation does not include financing and discounting issues. Discount rates are often included in cost-benefit analyses when comparing projects, usually on different timescales. In this case, the discounting rate does not impact the analysis and interpretation since the same time horizon is considered for all the plants, as well as the year of installation and replacement of the components.

The model incorporates several constraints, including the balance of hydrogen mass, energy balances, and technology behavior (i.e., the performance of energy supply, storage technology, and hydrogen production technologies) (see Supplementary Eqs. (2–13) in

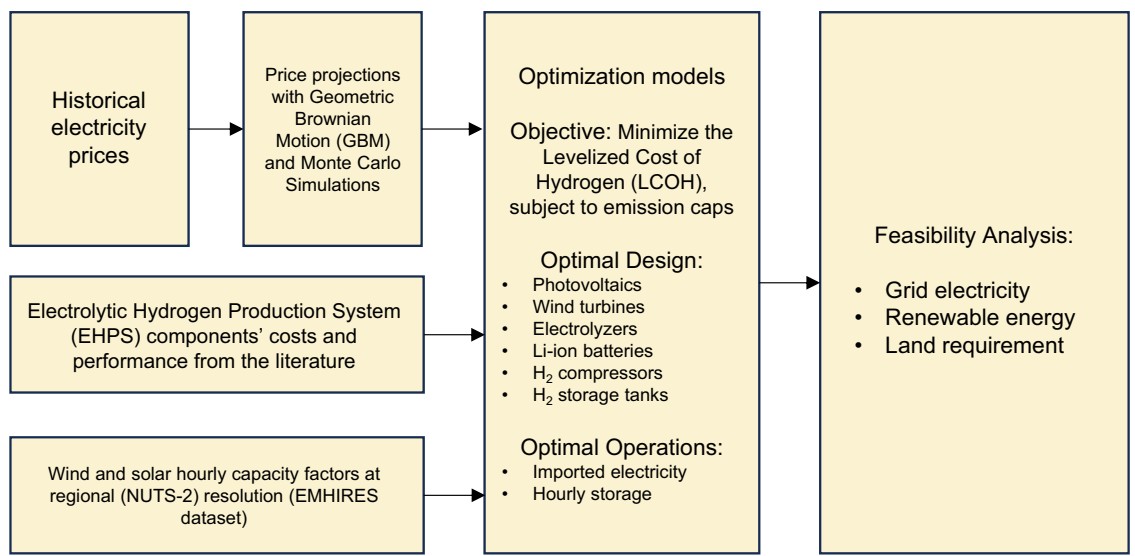

**Fig. 7 | Methodological framework.** Historical electricity prices were sourced from the official European repository, while future price trends were estimated using Geometric Brownian Motion (GBM) and Monte Carlo simulations. Cost and performance metrics for the Electrolytic Hydrogen Production Systems (EHPS) were compiled from literature reviews and reports from industrial and governmental entities. In contrast, solar and wind capacity factors were derived from the European Meteorological derived High-Resolution RES generation time series (EMHIRES) dataset. These diverse datasets served as inputs for the optimization models, which minimize the levelized cost of hydrogen by optimizing the design and operational parameters for each European ammonia plant subject to emissions limits on the carbon dioxide equivalent content of hydrogen. The study also evaluates the feasibility of transitioning European ammonia production from traditional fossil fuels to electrolytic ammonia synthesis, with assessments conducted at both plant and regional scales. In this context, the regions correspond to the NUTS-2 level, which stands for Nomenclature of Territorial Units for Statistics, level 2. NUTS-2 is a geocode standard developed by the European Union to define subdivisions of countries for statistical purposes.

Supplementary Note 5). Furthermore, constraints on the well-to-gate $CO_2e$ content of hydrogen are incorporated (see "Carbon emission caps"). The operations of the EHPS were optimized using hourly data spanning an entire year, i.e., 8760 h, to account for the role of weather conditions randomly distributed throughout the year (see "Wind and solar capacity factors"). A weak negative correlation between the annual solar and wind mean capacity factor and LCOH was found (Supplementary Fig. 8), demonstrating the importance of high temporal resolution in optimizing the installed capacity required to satisfy the ammonia plant demand and, in turn, estimating the costs.

Additional tests were conducted using other optimization techniques (i.e., heuristic optimization). Specifically, local, global, and the combination of the two (i.e., hybrid approach) were examined (Supplementary Note 6 and Supplementary Fig. 9). While heuristic methods, and particularly a hybrid approach combining Differential Evolution and Nelder-Mead, provided results equivalent to those of the MILP method, they were significantly slower in finding the solution (less than 1 min for MILP versus ~20 min for the Differential Evolution and Nelder-Mead combination; see Supplementary Table 12).

**Carbon emission caps**
The well-to-gate $CO_2e$ content for each unit of hydrogen $\Gamma_{H2}$ is calculated by dividing the total lifetime operational emissions from grid-imported electricity ($\sum_{l=1}^{L}\sum_{t=0}^{T}\gamma_E M_{E,t}$) by the total amount of hydrogen produced over the lifetime of the EHPS ($\sum_{l=1}^{L}\sum_{t=0}^{T}D_{H2,t}$) (Eq. (3)).

$$\Gamma_{H2} = \frac{\sum_{l=1}^{L}\sum_{t=0}^{T}\gamma_E M_{E,t}}{\sum_{l=1}^{L}\sum_{t=0}^{T}D_{H2,t}} \qquad (3)$$

where $M_{E,t}$ is the imported grid electricity at time $t$ and $\gamma_E$ is the average life cycle $CO_2e$ emission intensity of the grid from 2025 to 2050 calculated for each country, based on the historical data provided by the European Commission (see "Grid electricity price and carbon intensity"). Thus, the EHPS can import grid electricity if the average well-to-gate $CO_2e$ emissions of hydrogen do not exceed the emission cap constraint considered in this study (3-cap, 1-cap, 0.5-cap, 0.1-cap, 0-cap) (Eq. (4)).

$$\sum_{l=1}^{L}\sum_{t=0}^{T}\gamma_E M_{E,t} \le \varepsilon_{H2}\left(\sum_{l=1}^{L}\sum_{t=0}^{T}D_{H2}\right) \qquad (4)$$

where $\varepsilon_{H2}$ represents the emission cap under consideration in kg $CO_2e$/kg $H_2$.

**Scope 3 embedded emissions**
Including emissions from the manufacturing of components introduces significant uncertainty, largely due to the intricate nature of tracing emissions along the comprehensive technology supply chain. This complexity is particularly pronounced for technologies such as electrolyzers, solar PV modules, or wind turbine components, often subject to international trade[22]. Furthermore, actual emission inventory data are frequently unavailable or inaccessible[22].

While there are inherent complexities in estimating emissions related to the manufacturing of key technology components, several studies offer valuable insights into average emissions[63–65]. Emissions embedded in technology $\gamma_k$ are calculated for PV, WT, Li-ion batteries, electrolyzers, hydrogen compressors, and storage tanks, and multiplied by the respective installed capacity $\hat{P}_k$ (see Supplementary Note 3).

It is noteworthy that due to anticipated reductions in the emission intensity of electricity generation, as projected by the IEA scenarios[22], emissions from material production and technology manufacturing are expected to decrease. Consequently, indirect emissions from materials and manufacturing processes involved in hydrogen production could be less in the future than today.

**Ammonia production process and EHPS**
Ammonia production primarily relies on the HB process, which synthesizes ammonia from nitrogen and hydrogen. The process begins with the extraction of nitrogen from the air using an air separation unit

(ASU) through cryogenic distillation. Hydrogen is typically derived from natural gas through steam reforming, where high-temperature steam (700–1000 °C) separates hydrogen atoms from methane. Hydrogen and nitrogen are then combined in the ammonia synthesis loop (synloop), where under high pressure (150–200 atmospheres) and temperature (400–500 °C), they react to form ammonia. The presence of an iron catalyst facilitates this reaction. The synthesized ammonia is cooled and condensed into a liquid for storage and transportation.

In an electrolytic ammonia production plant, the process differs in the source of the hydrogen. Here, hydrogen is derived from water electrolysis instead of SMR. In this procedure, water is split into its constituent elements, hydrogen and oxygen, by applying an electric current.

Ammonia plants operate continuously at full load[66]. The HB process has been optimized over decades for steady-state operation, requiring consistent reactant supply, product removal, and specific temperature and pressure conditions[67]. This standard has been enabled by the dispatchability of natural gas. In line with this, even when switching to electrolysis for hydrogen production, it is assumed to maintain the continuous plant operation. This approach aligns with many of today's large-scale electrolytic plant projects (an example is the Greenko, GIC, and Gentari partnership for renewable-based electrolytic ammonia production in India[68]).

The European ammonia industry is characterized by high homogeneity regarding plant size, technology, operations, and hydrogen and ammonia demand and supply, with a few dominant players. Considering the average production rate of 7500 kg $H_2$/h for 24/7 operations in European ammonia plants[2,69–71], it is assumed that each EHPS must meet the same hourly hydrogen output ($D_{H2}$) as the current SMR system to avoid disruptions in existing plant operations.

Despite recent efforts to develop more flexible plants that can adjust operations following renewable energy availability, achieving this flexibility at a large scale presents inherent challenges, particularly with the air separation unit and the ammonia synloop[23,35,66]. In large-scale plants, air separation units typically employ cryogenic distillation and have a minimum load limit between 50 and 70%, below which extensive ramp-up times would be required[23]. The ammonia reactor and compressors also pose operational challenges due to the need to maintain precise conditions, including specific hydrogen-to-nitrogen ratios, temperatures, and pressures[66,72]. If these conditions fall below certain thresholds, it could take up to 24 h to resume operations, as demonstrated during plant inspections. Consequently, while efforts are being made to increase plant flexibility, the prevailing industry trend still favors continuous operation with hydrogen/energy storage support.

Nevertheless, the robustness analysis examines the implications of relaxing the constraint on hydrogen production output. Specifically, $D_{H2}$, which was a fixed input parameter in the main analysis, is now treated as a decision variable optimized each hour $D_{H2,t}$. However, the total annual hydrogen demand, remains consistent with the continuous operation scenario.

A minimum load parameter, denoted as $\delta_{H2}$, has been set at 50% to guarantee that the hydrogen output to the ammonia synthesis loop from the EHPS never falls below a safe minimum load (Eq. (5)). It should be noted that this assumption of the minimum load is rather optimistic. Wang et al.[23], for example, consider a minimum load of 60% with ramp rates of 20% per hour.

$$\delta_{H2}D_{H2,t} \le D_{H2,t} \le \frac{1}{\delta_{H2}}D_{H2,t} \forall\ t \in \{0,\ldots,T\} \qquad (5)$$

Compared to the reference analysis with continuous production, the costs associated with the ASU, ammonia synthesis loop, and

cryogenic storage tanks ($\Delta_f$) must be included in the LCOH (Eq. (6)).

$$LCOH = \frac{z_{cost} + \Delta_f}{\sum_{l=1}^{l}\sum_{t=0}^{T}D_{H2,t}} \qquad (6)$$

Given that production may occur at a 50% load during certain hours, it is necessary to oversize the ASU and synthesis loop to accommodate higher loads during peak production times. The maximum hydrogen volume entering the synthesis loop is set to be smaller than $1/\delta_{H2}$. This constraint ensures the installed capacities of the ASU and the synthesis loop do not exceed twice the size required for continuous operations. For large-scale ammonia plants, the doubling of installed capacity would mean that the ASU costs approximately 150 million EUR, and the costs for the ammonia synthesis loop, inclusive of auxiliaries and balance of system, amount to 300 million EUR.

However, this analysis remains preliminary and somewhat simplistic. The installed capacity and operation of both subsystems should be modeled and optimized in greater detail, taking into account other cost factors such as control systems improvements. Furthermore, as per Wang's findings[23], variations from nominal load in the HB loop will likely result in reduced efficiency. Therefore, additional energy losses must be contemplated, alongside the need for more intermediate storage for nitrogen and other minor enhancements to prevent reactor poisoning.

Despite the simplicity of the model, such increases in energy demand and inefficiencies are expected to have a minimal impact on the LCOH and LCOA, as the majority of electricity usage and costs are predominantly incurred during the production of hydrogen.

## Grid connection upgrades

Existing fossil-based ammonia plants are already grid-connected, albeit with a few MW of power capacity. However, when retrofitting these plants–replacing SMR hydrogen with electrolytic hydrogen production–the peak capacity demand from the grid will increase. This means that when a large amount of energy is required, the grid should be capable of supplying it. As a result, grid connection upgrades to accommodate this increase in peak capacity demand are included in the LCOH calculation. This includes the installation of new high voltage alternating current (HVAC) wires and a transformer to step down from high voltage to low voltage at the plant side (unit cost in Supplementary Table 13).

To account for energy losses in transmission, an increase in the price of delivered electricity has been incorporated, mirroring the approach taken by Salmon and Bañares-Alcántara in their work[25]. The additional power required to compensate for these transmission losses leads to a proportional increase in the cost of power. Given the typical location of existing plants in industrial areas, proximity to the grid connection point (within 5 km) is assumed, and line losses are negligible due to the short distance. Hence, only transformer losses (1%, high to low voltage) and rectifier losses (5% on the plant side) are included.

## Electrolyzers

Electrolyzers are devices that use electricity to split water into hydrogen and oxygen through a process known as electrolysis. Currently, there are two types of commercially available electrolyzers, which differ in terms of costs, efficiencies, maintenance, and operation costs: alkaline (ALK) and proton exchange membrane (PEM). Additionally, other electrolyzer technologies, such as membrane-less (ML) and solid oxide electrolysis cells (SOEC), are in the early stages of development and typically tested at a small scale.

The choice of electrolyzer depends on various factors such as the required production rate, purity of hydrogen gas needed, efficiency, cost, and operational conditions. No single electrolyzer technology outperforms the others in all aspects, and the best choice depends on

the specific requirements and priorities of the application[73]; however, for bigger applications where the industrial processes require a stable hydrogen supply (like in ammonia plants), ALK is the most suitable[1,6,37,74] and therefore was considered in this analysis. PEM electrolyzers are gaining momentum, but the availability of scarce metals for manufacture limits large-scale installations[75]. SOECs are a promising technology for hydrogen production but are not yet considered mature. They operate at high temperatures and pressures, which allows for high efficiency[76]. However, some challenges still need to be addressed before they can be widely commercialized, such as increasing their durability and lifetime[76]. Lastly, ML electrolyzers may offer the lowest costs per installed unit and less reliance on rare materials[52,53]. However, the largest prototypes are currently only at the kW scale[52,53].

Reference costs for ALK electrolyzer collected from IEA[6] are shown in Supplementary Table 14, while further technical details and future cost projections for the ALK electrolyzer are listed in Supplementary Note 7 and Supplementary Fig. 10. The initial electrolyzer installation cost includes bare erected cost, engineering, procurement, construction cost, process and project contingencies, and overnight cost, comprising an additional 69% of the overall system cost[53]. Only the stack component is substituted when replacement is needed (ALK stack lifetime is 10 years), which usually represents 50% of the system cost[77]. Anticipated increases in production volumes, among other factors, are expected to induce substantial cost reductions over the coming decades (e.g., IEA, 2022[20]). In this analysis, future electrolyzer costs are exogenously specified, assuming a roughly 60% cost reduction in per-kW stack costs over the 2022–2050 period based on data from Supplementary Table 14. Annual routine maintenance costs are assumed to be 2% of the system cost[77].

Once the ALK stack component requires replacement, it will be replaced with another ALK. This is because the balance of plant (BoP) is designed around the use of an ALK electrolyzer. Consequently, replacing it with a different type of electrolyzer would necessitate substantial modifications to most of the BoP to accommodate differing outlet pressures, temperatures, potential ramp rates, and maintenance procedures.

While the model primarily considers ALK electrolyzers, the scope is broadened in the robustness analysis by including SOEC and ML based on cost and performance projections collected from refs. 6 and 52, respectively. Cost and performance data are in Supplementary Table 15.

**Renewable power generation**
Utility-scale solar PV and onshore wind turbines are installed to power the electrolyzers. For both PV and wind turbines, data were collected from the International Renewable Energy Agency (IRENA) database[78] (Supplementary Table 13). IRENA's database is one of the most comprehensive data sources containing cost and performance data (including, among others, the balance of system, transformers, grid connection, wiring, and power electronics) of most renewable projects worldwide. While PV and wind turbines have become mature technologies, regional cost variation persists. IRENA provides a detailed breakdown of total utility-scale PV installed costs, ranging from hardware to installation and soft costs for each country. The average value of the IRENA European dataset was used when country-based data were unavailable. Due to the long life of solar PV and wind turbines, the infrastructure will not be replaced during the projection period; thus, only 2022 cost data were used in this work. Linear interpolation was used to obtain 2024 data from the available 2020[78] database and 2030 projections[79]. Pessimistic and optimistic cost values were obtained from the range provided by IRENA at a global scale and applied to all national total installed costs.

The IRENA dataset only provides the total installed costs of onshore wind turbine installations worldwide without country-specific information. To enable similar treatment of wind and solar input data, the same

cost differences (in percentage) across Europe identified for solar PV installations were also assumed for wind turbines. Annual maintenance costs were also collected from the IRENA dataset for coherence.

Due to the high and continuous hydrogen demand of European ammonia plants, large PV and wind turbine installations are required, resulting in large land usage. Land area for installations is divided into two categories: direct land area and total land area[80–82]. Direct land area refers to the physical footprint of the renewable energy infrastructure itself, including the solar arrays or wind turbines, as well as associated elements like access roads, substations, service buildings, and other immediate necessities. On the other hand, total land area extends beyond the direct physical footprint of the infrastructure. It encompasses the complete land area associated with a PV or wind turbine farm, often represented by the perimeter enclosing all installations. This area factors in the spacing between installations, buffer zones, and other land-use considerations linked to the farm's operation. While direct land area can be calculated more easily, total land area can significantly vary based on the specific location and its geographic and regulatory constraints.

The National Renewable Energy Laboratory (NREL) has produced detailed reports on existing large-scale PV and wind turbine projects, offering data on both direct and total area usage[80–82]. For wind turbines, direct land usage is $1.0 \pm 0.7$ hectares/MW, while total land usage is $31 \pm 22$ hectares/MW. Given that the wind turbines considered in this analysis are larger (2.8 MW) than those in the NREL study (1–3 MW), a smaller area per MW installed is assumed, taking 15 hectares/MW as a reference value, of which 1.0 hectares/MW are direct land usage (7%)[81]. For PV installations, the difference between direct and total land area is less noticeable. The capacity-weighted average for direct land-use requirements is 3 hectares/MW, with 40% of power plants requiring between 2.5 and 3.2 hectares/MW. The total-area capacity-weighted average is 3.6 hectares/MW, with 22% of plants requiring between 3.2 and 4 hectares/MW. A reference value of 3.5 hectares/MW is considered for this analysis of which 3.0 hectares/MW are direct land usage (86%)[80]. Land cost was therefore included in the LCOH calculation, and land price data were collected from the European Commission dataset[83] (Supplementary Fig. 11).

The LCOH also accounts for the losses in the transmission of electricity from renewable power generation to the electrolysis plant. Transmission from PV and wind farm installations to the plant can occur via existing grid lines or by constructing a dedicated transmission line from the farm to the plant. The latter scenario is more probable. The existing grid may lack the capacity to handle additional power at a GW scale without significant upgrades and large-scale electrolysis may result in grid congestion[37,84]. Moreover, a certificate of origin must be introduced to ensure that the electricity utilized by the plant is sourced from renewable infrastructure[37]. Hence, the construction of new, separate transmission lines is assumed.

HVAC transmission systems are more expensive than low voltage alternating current (LVAC) systems, but LVAC systems have limited capacity, which may be insufficient for the plant. Hence, HV transmission is deemed suitable. Additionally, LVAC systems have high transmission losses of up to 30% per 100 km, compared to HVAC's 4% per 100 km[25]. High voltage direct current (HVDC) is another option, with even lower power losses in wires (3% per 1000 km), but it tends to have higher capital costs and is usually optimal for long distances (500–1000 km)[85]. Given the assumption that the renewable installations are located within short distances (<10 km), HVAC is assumed to transmit electricity from PV and WT to the plant. Wind Turbines typically generate LVAC, which must be stepped up to HV using transformers, transferred to the plant, stepped down, and then converted into low-voltage direct current (LVDC) using rectifiers. Conversely, PV systems produce LVDC, requiring an additional step to convert LVDC to LVAC before stepping up and integrating into the dedicated HVAC line. Costs associated with inverters are already included in the IRENA

dataset. Costs and losses associated with transformers, wires, inverters, and rectifiers are shown in Supplementary Table 13. Total transmission losses from the WT to the electrolysis hydrogen system are around 7% (2% transformers, 5% rectifier), and for PV systems, around 10% (3% inverter, 2% transformers, 5% rectifier).

## Battery energy storage systems

Li-ion battery systems for electricity storage can be installed to balance periods of low or no renewable energy generation and ensure a consistent power supply for the electrolysis process. Utility-scale Li-ion battery system costs were collected from NREL[48,86] (Supplementary Fig. 12). The lifetime ranges from 8 to 18 years with a median of 15 years, while round-trip efficiency (system efficiency through a charge/discharge cycle) is 85%[48]. Annual routine maintenance cost is assumed to be 2.5% of the system cost.

## Compressor and tanks

Surplus renewable electricity can also be used to directly synthesize hydrogen to store for balancing periods with low availability of wind and solar. Hydrogen from the electrolyzer is in gaseous form, conventionally from atmospheric pressure to 30 bar, and is typically compressed up to 350 bar for tank storage[87]. Compression is usually done in two ways: using a standard separate compressor and changing the electrolyzer operating pressure. Compressing hydrogen to 30–100 bars has a relatively small efficiency penalty and additional cost. However, major-scale compressors are more efficient for higher pressures, and larger-scale electrolyzers result in a smaller additional cost per unit of hydrogen produced[87]. Hence, as the scale increases, mechanical compression is preferred over electrochemical. Moreover, mechanical compression is required if a pressure higher than the operating pressure of the electrolyzer is needed. Since the EHPS is planned to deliver ambient pressure hydrogen to the ammonia plant, it is not feasible to increase the operating pressure of the electrolyzer. Therefore, mechanical compression is assumed. In general, compression losses are around 5–10% of hydrogen lower heating value (LHV) for compression at 350 bar[73] (2 kWh/kg $H_2$ compressed at 350 bars in the analysis). Although hydrogen compressors are already a mature technology, their cost is expected to decrease over the years (26% cost reduction by 2030 and 50% by 2050[71]). Compressor system cost data were collected from[71] (Supplementary Fig. 13). Compressors require annual maintenance to guarantee normal operations. The annual maintenance cost is around 4% of the system cost[87], while the lifetime is about 10 years[87]. High-pressure hydrogen tanks will store hydrogen for balancing periods. Cost data were collected from[88] (see Supplementary Table 13).

## EU-ETS future price projections

The EU-ETS is a carbon control scheme that issues carbon allowances and ultimately puts a cost on carbon dioxide emissions. EU-ETS allowances are allocated at the member state level, but overall allowances will be reduced as Europe pushes towards net-zero[89]. The impact of EU-ETS on electrolytic hydrogen costs is studied in the robustness analysis. EU-ETS future price projections are based on BloombergNEF report[90] (Supplementary Fig. 14). In the robustness analysis, the pessimistic scenario assumes a high EU-ETS price of 145 EUR/t $CO_2$e, the reference case uses a moderate price of 86 EUR/t $CO_2$e, and the optimistic case is based on a low price of 27 EUR/t $CO_2$e.

## Retrofitting costs

Ammonia plants will undergo retrofitting to replace the SMR hydrogen production system with EHPS. Retrofitting costs include the substitution and upgrade of some plants' components for operations with electrolytic hydrogen. Current research and practical applications, such as the Puertollano project[91], have shown that up to 10–15% of hydrogen derived from electrolysis can be integrated without any modifications. However, to increase this ratio, adjustments are

necessary due to changes in heat flows and operational flexibility. For instance, steam from the ammonia converter cannot be used for the SMR anymore. Anticipated modifications include replacing the electric start-up heater before the ammonia converter, installing additional electric heaters to facilitate flexible operation, and incorporating steam generators for periods of low-load operation when the ammonia converter generates insufficient steam. The estimated cost for these modifications is about 5–10% of the cost of installing a new ammonia loop (detailed calculation in Supplementary Note 8).

The analysis also includes the sunk costs associated with decommissioning the SMR hydrogen production system. These sunk costs represent expenses that have already been incurred and cannot be recovered. However, the potential offset of these sunk costs is also considered by incorporating the estimated residual or scrap value of the decommissioned SMR. It is anticipated that this residual value will be around 10% of the SMR's original cost (detailed calculation in Supplementary Note 8).

## Grid electricity price and carbon intensity

Modeling the price behavior of grid electricity is challenging; electricity prices vary both on a daily and long-term scale[92]. Random events, such as load variations, contingencies, network congestion, and changes in demand, can cause prices to fluctuate throughout the day. In the long run, additional factors such as oil price changes, regulatory policies, political intervention, technological changes, energy mix variation, and grid operations can drastically influence long-term electricity prices[93]. As these factors are difficult to anticipate, the present work focuses on long-term forecasting and neglects short-term fluctuations. One method that has been employed in this context is stochastic process modeling. Geometric Brownian motion (GBM), also known as exponential Brownian motion, is a continuous-time stochastic process in which the logarithm of the randomly varying quantity follows a Brownian motion with drift. GBM is one of the most applied stochastic processes for long-term electricity price forecasts where future values are calibrated on historical time series[94]. Hence, in this study, Monte Carlo simulations and GBM were used to project future electricity prices from 2025 to 2050. The methodology was applied to a dataset provided by the European Commission that incorporates country-based monthly retail and wholesale historical electricity prices for industrial users, spanning from January 2008 to December 2019[95].

GBM operates on the assumption that the logarithmic returns of electricity prices are normally distributed and that these returns can be used to estimate future price paths. The process begins by calculating the logarithmic returns of historical electricity prices for each country. These returns are then used to derive key parameters for the GBM model: the mean, variance, drift, and standard deviation (see Supplementary Eqs. (14) and (15) in Supplementary Note 9). The drift represents the direction trends tend to follow, while the standard deviation measures price volatility. Next, random future price changes, or "shocks", are generated based on these parameters. These shocks represent the unpredictable factors that could impact future electricity prices. The simulation then constructs potential future price paths by applying these shocks iteratively to the last known price. This creates a distribution of possible future price paths for each month until 2050, reflecting the inherent uncertainty in these projections. Once these potential future price paths are established, the 5th, 50th, and 95th percentiles of the simulated prices are used to construct each country's optimistic, average, and pessimistic electricity prices. The 5th and 95th percentiles serve as lower and upper bounds, indicating a 90% confidence interval, while the 50th percentile represents the median price. Finally, the average prices over the period from 2025 to 2050 are calculated for the 5th, 50th, and 95th percentiles and used as input for the optimization (Supplementary Table 13). Grid electricity prices vary from 30 EUR/MWh in Norway to 234 EUR/MWh in the UK, with a mean value of 115 EUR/MWh in the reference case.

Regarding emissions from the grid, historical data and future projections were collected from the JRC-COM-NEEFE (National and European Emission Factors for Electricity Consumption) dataset provided by the European Commission[96]. The dataset contains data from 1990 to 2020 for all countries globally. The indirect emissions from electricity consumption are calculated by dividing total national $CO_2e$ emissions from electricity production from all input energy carriers by the total final electricity consumption. It, therefore, includes upstream emissions while excluding emissions from manufacturing technologies, thereby attributing 0 $CO_2e$ per kWh when electricity is generated from renewable sources. The future carbon content of grid electricity is calculated based on the European Environmental Agency (EEA)'s 2030 projections[97], coupled with the assumption of achieving carbon neutrality by 2050.

## Abatement cost

Abatement Cost (AC) refers to the cost of reducing negative environmental externalities, quantified as tons of $CO_2e$ in this work (Eq. 7).

$$AC = 1000 \times \frac{LCOH - LCOH_{SMR}}{\Gamma_{H2_{SMR}} - \Gamma_{H2}} \qquad (7)$$

where AC is abatement cost in EUR/ton of $CO_2e$ abated; $LCOH_{SMR}$ is LCOH of SMR hydrogen (2 EUR/kg $H_2$); $\Gamma_{H2_{SMR}}$ is $CO_2e$ content of SMR-based hydrogen (12 kg of $CO_2e$/kg $H_2$).

## Wind and solar capacity factors

The European Meteorological derived High-Resolution RES generation time series (EMHIRES) dataset[98] was used to obtain the capacity factor of onshore wind turbines and solar PV for each region where ammonia plants are located. EMHIRES is a European dataset containing information regarding the generation of intermittent renewable energy resources for electricity generation derived from a combination of meteorological data. EMHIRES Part I[99] focuses on wind power generation, while EMHIRES Part II[100] focuses on solar power generation. For Part I and Part II, the hourly wind and solar power generation time series are based on meteorological conditions over 30 years (1986–2015). EMHIRES also calculates the hourly capacity factor (CF) at a NUTS-2 level. This indicates the ratio between the sums of the energy produced and the maximum possible generation. Europe shows a significant variability of weather conditions which directly influence the capacity factors of wind and solar power generation projects, as shown in Fig. 8.

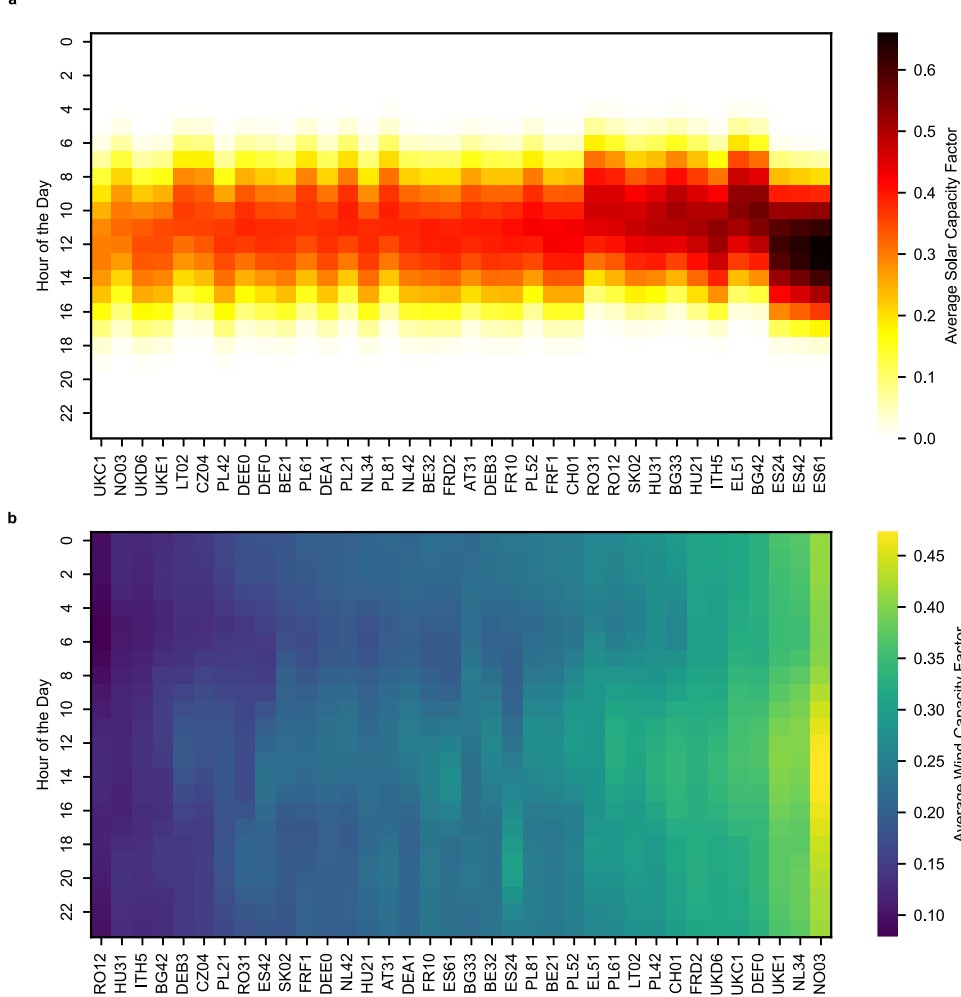

**Fig. 8 | Historical hourly capacity factor for photovoltaics and wind turbines in European regions with ammonia plants.** Capacity factors were derived from the European Meteorological derived High-Resolution RES generation time series (EMHIRES) dataset, which provides comprehensive historical hourly energy data for both solar and wind energy. The average hourly capacity factor was computed by aggregating annual hourly data to yield an average value for each hour. **a** Illustrates the hourly capacity factors for solar energy for each region, indicating the fluctuating nature of solar energy availability throughout the day. **b** Highlights the capacity factor for wind energy. Regions coded according to the NUTS-2 level of the Nomenclature of Territorial Units for Statistics (NUTS) system and ordered by average annual capacity factor.

While the dataset might not fully capture certain geographic-specific factors like terrain conformation, natural obstacles, and shading, it offers a good representation of renewable capacity and weather variability across Europe. This underscores the role of local renewable resources in shaping the economic and technical viability of electrolytic hydrogen production.

To capture the seasonal effect accurately while keeping the computational time within reasonable bounds, one representative year of hourly wind and solar capacity factors (8760 h/iterations) was selected from the 30 years of data available for each region studied. To achieve this, the mean annual capacity factor was calculated for wind and solar energy in each region by averaging the capacity factor over all hours in the dataset. The sum of the mean capacity factors for wind and solar energy was then calculated to give each region an overall measure of energy production capacity for each year. The year with the median sum of capacity factors was selected as the representative year, as it represents the average energy production capacity in each region.

### Robustness analysis

To identify the representative regions, the mean annual capacity factor $\bar{\omega}_{R,i,j}$ was calculated for each region $i$ (280 NUTS-2 regions) and year $j$ (from 1986 to 2015) available in the EMHIRES datasets for both solar and wind energy sources $\omega_R \in \{solar, wind\}$ (Eq. (8)).

$$\bar{\omega}_{R,i,j} = \frac{\sum_{t=0}^{T} \omega_{R,i,j,t}}{T} \qquad (8)$$

European regions were then clustered based on a specific set of rules relating to their capacity factors for solar and wind energy. Regions that exhibit high-capacity factors for renewable energy fall into the top 75th percentile for either solar or wind mean annual capacity factors and have a mean annual capacity factor greater than the 25th percentile for the alternate energy source. Solar-dominated regions are those that

rank within the top 25th percentile for solar energy capacity yet are in the lowest quartile for wind energy potential. Conversely, wind-dominated regions sit in the top 25th percentile for wind energy capacity but find themselves in the lowest quartile for solar energy. Median-capacity regions are characterized by both solar and wind resources surpassing the 25th percentile but not reaching beyond the 75th percentile, reflecting a balanced mix of the two energy sources. Lastly, low-capacity regions are identified by having one of the energy sources—either solar or wind—with a mean annual capacity factor below the 75th percentile, while the other source does not exceed the 25th percentile, signaling a limited potential for renewable energy exploitation.

After classifying all regions into five categories, a representative region for each group was identified, specifically selecting the region that exhibits the most extreme case within its category. For instance, in the wind-dominated category, the region with the highest mean annual wind capacity factor and the lowest solar was chosen. Conversely, in the solar-dominated category, the region with the highest mean solar capacity factor and the lowest wind was selected. This process was repeated for the low-capacity, median-capacity, and high-capacity categories, resulting in the creation of five representative regions that illustrate distinct weather scenarios.

More specifically, NO05 (Vestlandet, Norway) is a representative wind-dominated region, showing the highest mean annual capacity factor at 49.7% (100th percentile) specifically for wind, yet it has relatively poor solar energy, with a capacity factor of just 8.3% (0th percentile). Conversely, ES43 (Extremadura, Spain) is an example of a solar-dominated region, with a high mean annual solar capacity factor of 20.2% (100th percentile), while its wind energy capacity trails significantly with a capacity factor of merely 10.5% (2nd percentile). These selections are made intentionally to represent regions with a strong prevalence of one energy source, wind or solar while having limited potential for the other. In contrast, EL42 (South Aegean, Greece) emerges as a high-capacity region, landing in the 98th percentile for

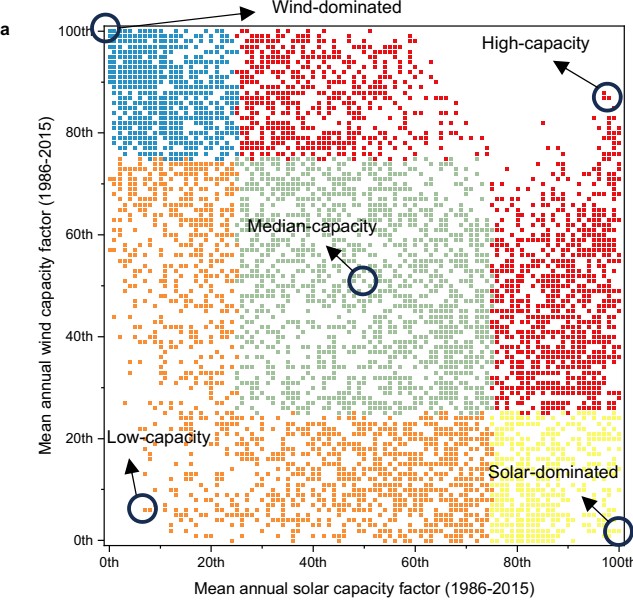

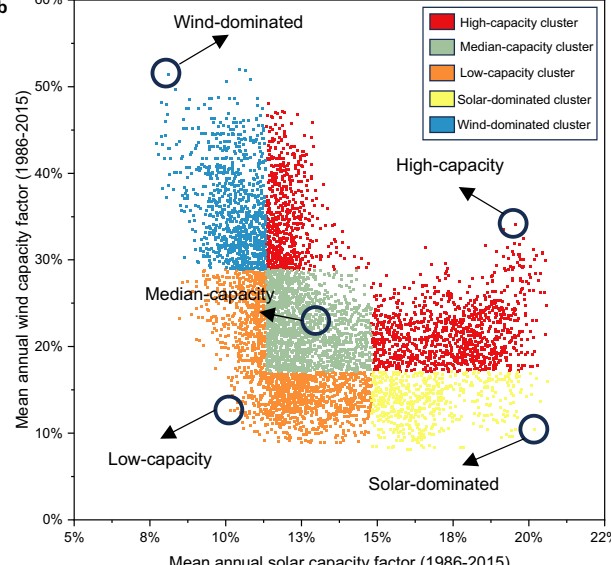

**Fig. 9 | European regions at Nomenclature of Territorial Units for Statistics, level 2 (NUTS-2) clustered based on their mean annual capacity factor along with the five selected representative regions and years.** Each dot represents a specific region, and year means the annual capacity factor. **a** Displays the capacity factor as a percentile rank for each region and year, and **b** expresses the capacity factor as a percentage. Five distinct clusters are identified, each with a representative region, showcasing diverse capacity characteristics: South Aegean, Greece coded as EL42 in NUTS-2 in 1987 exemplifies a High-capacity region with photovoltaic and wind power capacity factors at 19.5% (98th percentile) and 34.1%

(87th percentile), respectively; Střední Čechy, Czech Republic coded as CZ02 in 1989 represents a Median-capacity region with 12.3% (50th percentile) for photovoltaics and 22.1% (51st percentile) for wind power; Kassel, Germany coded as DE73 in 2013 characterizes a low-capacity region, with figures at 10.2% (7th percentile) for photovoltaics and 12.6% (6th percentile) for wind turbines; Extremadura, Spain coded as ES43 in 2005 represents a Solar-dominated region with photovoltaics at 20.2% (100th percentile) and wind at 10.5% (2nd percentile); and Vestlandet, Norway coded as NO05 in 1990 stands as a wind-dominated region with photovoltaics at 8.3% (1st percentile) and wind at 49.7% (100th percentile).

solar energy (19.5% CF) and the 87th percentile for wind energy (34.1% CF). Lastly, DE73 (Kassel, Germany; 10.2% solar CF−7th percentile; 12.6% wind CF− 6th percentile) and CZ02 (Střední Čechy, Czech Republic; 12.3% solar CF−50th percentile; 22.1% wind CF−51st percentile) represent regions with low and median-capacity factors, respectively (Fig. 9).

This approach effectively captures the variability and extremes of climate conditions while reducing the computational burden of simulating every year and region in the dataset. Simultaneously, it ensures that the derived insights remain valuable and applicable to other regions within similar categories, thereby maintaining their relevance across a wider geographical context. Importantly, it also simplifies the task of classifying European ammonia plants into these categories, further enhancing the practicality and applicability of the results (Supplementary Fig. 15).

### Feasibility analysis

The annual grid electricity demand for each ammonia plant under the 1-cap (the one with the lowest AC) was compared to the 2020 national and regional electricity demand. Despite European regions being heterogeneous in terms of land area (min 2148 km$^2$, mean 19,867 km$^2$, max 87,268 km$^2$) and energy consumption (min 2 TWh/year, mean 14 TWh/year, max 86 TWh/year), the ratio of ammonia plant demand to regional demand (Supplementary Table 11) underlines the impact of an electrolytic ammonia plant using grid electricity as backup when renewable output is low.

Data on the theoretical maximum renewable potential at the NUTS-2 level, excluding hydroelectric power, were gathered from the study by Kakoulaki et al.[38] to determine if regional renewable resources could satisfy the energy requirements of the electrolysis system.

Using Gabrielli et al.[39] methodology and the Food and Agriculture Organization of the United Nations (FAO) 2020 land data, available land was estimated by excluding forest areas[101]. It is assumed that 70% of non-forest land is suitable for renewable installations, in line with global usable land estimates that consider institutional and biophysical constraints[101]. This land is then compared with the requirements for PV and WT installations in each region, considering the mean land availability of the corresponding region.

### Reporting summary

Further information on research design is available in the Nature Portfolio Reporting Summary linked to this article.

## Data availability

Data supporting the findings of this study are available within the paper and its supplementary information. Key data sources include renewable power generation costs from IRENA, battery system data from NREL ATB, European ammonia plant locations from Fertilizer Europe (https://www.fertilizerseurope.com/fertilizers-in-europe/map-of-major-fertilizer-plants-in-europe/), SMR hydrogen production sites from FCHO (https://observatory.clean-hydrogen.europa.eu/hydrogen-landscape/production-trade-and-cost/hydrogen-production), geospatial data from EUROSTAT, electrolyzer data from IEA (https://www.oecd-ilibrary.org/energy/the-future-of-hydrogen_1e0514c4-en), wind and solar capacity factors from EMHIRES (https://op.europa.eu/en/publication-detail/-/publication/85b2dc7f-aa61-11e6-aab7-01aa75ed71a1/language-en and https://op.europa.eu/en/publication-detail/-/publication/a6c0cf55-45aa-11e7-aea8-01aa75ed71a1/language-en), grid carbon intensity data from JRC-COM-NEEFE (https://data.jrc.ec.europa.eu/dataset/919df040-0252-4e4e-ad82-c054896e1641), and industrial electricity prices from European Commission (https://energy.ec.europa.eu/data-and-analysis/energy-prices-and-costs-europe/dashboard-energy-prices-eu-and-main-trading-partners_en). Data can be requested through the channels provided on each respective website. The processed data supporting the findings of this study are available in the Supplementary Information and at: https://zenodo.org/records/10771014[102].

## Code availability

The repository required to replicate the main and sensitivity analyses, complete with all necessary code and input data, is hosted on GitHub and can be accessed at https://zenodo.org/records/10771014[102]. Data analysis was performed using Python 3.9.12, utilizing the following libraries: Pandas, NumPy, Seaborn, SciPy, Matplotlib, and Gurobipy. The optimization model was developed with Gurobipy and solved using the Gurobi solver version 10.0.1. Visualizations were created with Origin2023, Python's Matplotlib and Seaborn libraries, and spatial visualization was conducted with ArcMap 10.7.

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

## Acknowledgements

This work was supported by the Hong Kong Research Grant Council (26201721) and start-up support from the Hong Kong University of Science and Technology. M.J.R. and S.M. gratefully acknowledge the support from the Research Grants Council of Hong Kong, through the Hong Kong Ph.D. Fellowship Scheme (HKPFS). M.J.R and A.M. would also like to thank the Green Tech Fund (GTF202020131). S.M. expresses appreciation for the financial support provided by HKUST through the Overseas Research Award (ORA) to undertake an exchange period at ETH Zurich. S.M. is also grateful for the hospitality and guidance offered by the Reliability and Risk Engineering lab at ETH Zurich. The work of P.G. and G.S. was supported by the Swiss Federal Office of Energy (SFOE) as part of the SWEET PATHFNDR project. The views and ideas expressed here belong solely to the authors, not the funding agencies. The authors acknowledge the reviewers for their valuable viewpoints and important feedback, which have significantly improved the relevance, quality, and robustness of the work.

## Author contributions

S.M. and Z.L. initiated the study and secured funding. S.M., M.M.K. and Z.L. designed the study and led the investigation. S.M. developed the methodology and led the data collection with contributions from M.J.R., A.M., and K.R. S.M., M.J.R and A.M. formulated an early version of the optimization model. S.M. designed the final version. S.M., P.G., M.M.K. and Z.L. analyzed the results. S.M. visualized the results and wrote the paper. S.M., M.M.K. and Z.L. reviewed and edited the paper with contributions from P.G., A.M., M.J.R., K.R., F.C. and G.S.

## Competing interests

The authors declare no competing interests.

## Additional information

¹Division of Environment and Sustainability, Hong Kong University of Science and Technology, Clear Water Bay, Kowloon, Hong Kong SAR, China. ²Institute of Energy and Process Engineering, ETH Zurich, Zurich, Switzerland. ³Department of Global Ecology, Carnegie Institution for Science, Stanford, CA, USA. ⁴Department of Mechanical and Aerospace Engineering, Hong Kong University of Science and Technology, Clear Water Bay, Kowloon, Hong Kong SAR, China. ⁵Department of Physics, Technical University of Denmark, Kongens, Lyngby, Denmark. ⁶Ammonia Energy Association, Brooklyn, NY, USA. ⁷Catalytic Processes & Materials, MESA+ Institute for Nanotechnology, Department of Science & Technology, University of Twente, Enschede, The Netherlands. ⁸Koolen Industries, Europalaan 202, Hengelo, The Netherlands. ⁹Chair of Electrode Design for Electrochemical Energy Systems, University of Bayreuth, Bayreuth, Germany. ¹⁰Energy Institute, The Hong Kong University of Science and Technology, Hong Kong SAR, China. ¹¹Division of Public Policy, Hong Kong University of Science and Technology, Clear Water Bay, Kowloon, Hong Kong SAR, China. ¹²These authors contributed equally: Paolo Gabrielli, Alessandro Manzotti, Matthew J. Robson. ✉e-mail: smingolla@connect.ust.hk; magdalena@ust.hk; zhongminglu@ust.hk

