## [Peer Review File · Nature Communications]

REVIEWER COMMENTS

Reviewer #1 (Remarks to the Author):

General comments:

Interesting study addressing a timely topic. The data analysis seems satisfying but would gain being explained more precisely in the main text. The method used seems overall fine even though some aspects need a better clarification or improvement. The results brought by the study should be further compared and put in regards with the existing literature. Similarly, the literature review should be supplemented. The discussion could be improved including more context about green hydrogen discussion and options within EU.

Abstract

- You state that “electrolytic hydrogen” cuts emissions by 90%, maybe it would avoid later confusion to precise with more details how the electrolyser is powered in your study (off-grid wind turbines and solar, grid electricity with guarantee of origin, mix of both?)
- You talk about emissions: which scope are you looking at? And is it CO₂ or CO₂e?
- When you say “cap”, is it based on hourly emissions? Or yearly emissions and then averaged? Please be more precise on the description.

Introduction

- The literature review could be further expanded regarding techno-economic and environmental assessment of hydrogen/ammonia/methanol/FT-diesel. Mentioning the literature focusing on e-fuels LCA may also be relevant. Also please check and mention how studies focusing on global mapping of hydrogen/ammonia/other which does not have “limited number of locations” cover this topic.
- You say “examines all European ammonia production plant”. Maybe mention also here how many ammonia plants you are talking about.

Main

- You present some caps with value that seem a bit arbitrary at first reading. Comparing with caps discussed with existing (or in discussion) regulations could be relevant. Comparing with emissions from SMR also. And why using only these 4 caps? Please also precise if you are talking about a yearly cap (yearly amount of CO₂e divided by yearly H₂ production), an hourly cap or what?
- A paragraph explaining in greater details what is included or not in the emission accounting, the boundary of the system, etc... would be welcomed. Also how you are accounting for the emissions from the grid?
- Maybe say that you are using a hybrid set-up (grid + local power) in a more explicit way and explain to the reader what it means for example in terms of grid tariffs, connection costs, electricity price, etc...

- Please explain clearly from the beginning how you derive the electricity prices. Is it historical prices? (from which year?). Can you give some numbers? Maybe refer to some duration price curves.
- You are talking about hydrogen used for ammonia plant: a description on how the ammonia plants are operated (at full load all the time, in a flexible way?), how they are retrofitted for electrolytic hydrogen use?, how much it cost to retrofit (or do you need new ammonia plants?) would be needed to understand how the hydrogen is produced.
- Please discuss how to additional load induced by the extra electrolyzers will be handled by the existing grid. May it have any influence on the electricity price? Which power unit may be used to cover the additional load?
- In Fig.3 why not include LCOH SMR lines in all graphs?
- Did you include the retrofitting of existing ammonia plants in your abatement cost? It would be relevant to include or at least discuss it.
- Looking at results from Table 1 some of the results are surprising and would be worth some explanations: in most places AC goes from lowest with no-cap to maximum with 0-cap with a growing or constant trend, however in Poland it goes up and down, why?
- It would be nice to have some comparisons about what 61 km² or 342 km² represents i.e. compared to existing PV plants, city sizes, whatever. What is the “available land” in the region? Please explain in a few words and/or refer to the method section

General comment for the main section:

Some of the points addressed in the main paragraph seem to have been mentioned in the method section. In that case, it may help the reader to also summarize the most important assumptions in the main section text and refer to the method section for further details.

As you discuss about ammonia in most of the paper, it may be nice to give some ammonia production costs also (LCOA).

Please explain how your results compare with the existing literature.

Discussion

In general, it would help the reader to be more precise when describing the power supply you are talking about. For example, when you say “powered by dedicated renewables” do you mean off-grid plants? Or still a bit of grid can come in?

Some more detailed explanation on why you are getting some results would be welcomed. For example (among other examples), H₂ has a low carbon content in Spain: why? Is it because, Spanish grid is “clean” due to a lot of solar/hydro/wind in the mix? Is it because fossil electricity is expensive so mostly local renewable power is used? Same type of questions could be answered to explain some LCOH results.

You talk a lot about this 1 kg CO₂e/kg H₂ cap or lower but how would it look with higher caps like 1.5 or 2 kg CO₂e/kg H₂?

The discussion section should also be enhanced discussing with a greater extent the following aspects:

- How does your study fits with the discussed EU regulations/certification schemes for green hydrogen?
- You are talking about local retrofitting, but why not looking at H2 pipeline network within Europe and more global energy system modeling? What differences could raise between your localized study and energy system analysis models?

Method

- Would not it be possible to retrofit ammonia plant at a limited cost and then operate it semi-flexibly? It would be worth looking in more detail into “flexible ammonia plant” literature. Maybe perform a sensitivity analysis on this parameter if it has a significant influence on the results and mention it in the result/discussion section.
- Please quantify the time to run a scenario and the specification of the computer used to do it
- When you say that “emissions from the production of renewable facilities [...] is significantly lower than SMR”, it would be nice to supplement with further explanation (or not saying it, if you didn’t look at it with enough details): is it in location where the system is significantly oversized? Does it include land use change emissions? Which share of grid-electricity is used? Does grid electricity also include this additional emissions for solar and wind?
- You are discarding SOEC technology because is not mature yet, but if your time horizon goes to 2050 why not considering it? (it should be mature enough by then)
- Is the method used to calculate electricity price still valid to estimate 2050 prices in an energy system that may look completely different by then and were historical data may be “outdated”/inadequate? Please discuss.
- Please explain how did you include the different components lifetime in your LCOH calculation.

The method used for carbon emission accounting should be further detailed and compared with the state-of-the art methods used, for example, in the LCA field.

Remember to also point to the limitations of the methods used and their implication on the results you obtain.

Other

Check your reference list, some references are appearing twice.

Be sure to write CO₂e in the units when you talk about CO₂ equivalent.

Reviewer #3 (Remarks to the Author):

The paper's emphasis on providing insights for industry and policymakers through a comprehensive analysis of the European ammonia sector and the impact of emission caps adds value and significance to the research. The contrast with previous studies strengthens the originality and significance of the research.

This paper would benefit from providing more methodological details, discussing uncertainties, and offering a more comprehensive analysis of the findings to enhance the clarity and robustness of the study. It would be helpful to explain the scientific or policy justifications, relevant literature, or regulatory frameworks that influenced the selection of the emission caps under consideration. The authors cite two references to support hydrogen storage in favor of battery technology for the 0-cap. A more comprehensive examination of the literature would be helpful to confirm consensus amongst previous studies. The spiral

bar chart is visually appealing although challenging to interpret at first. The exclusion of the carbon intensity of technology manufacturing and installation from the analysis overlooks a significant portion of overall emissions associated with the electrolyzer's lifecycle. The authors should consider this aspect to gain a more comprehensive understanding of the emissions impact. In addition to capacity factors, other factors such as transmission losses, curtailment rates, and grid congestion should be considered in a robustness analysis to provide a more complete understanding of the system's resilience. The use of the EMHIRE dataset for calculating capacity factors is reasonable and provides consistency in the analysis. However, it is important to acknowledge that the dataset may not fully capture region-specific factors such as terrain, shading, regional weather patterns that affect wind and solar energy generation.

It may be beneficial to further elaborate on grid integration and the availability of renewable resources, specifically in relation to the electrolyzer technology selection and the feasibility of low-carbon hydrogen production. Explore how these factors were considered and their potential impact on the analysis.

The authors should explicitly state the assumptions made in the GBM and Monte Carlo simulations and discuss their potential impact on the forecasted results.

While constructing synthetic regions based on capacity factors is reasonable, it does not fully capture the real-world variability and uncertainties associated with renewable energy generation, including seasonal variations, weather patterns, and grid integration constraints.

By addressing these points, the authors can enhance the methodology, analysis, and overall quality of the research, providing more comprehensive and robust insights.

RESPONSE TO REVIEWERS' COMMENTS

Reviewer #1:

General comments

Comment #1

Interesting study addressing a timely topic. The data analysis seems satisfying but would gain being explained more precisely in the main text. The method used seems overall fine even though some aspects need a better clarification or improvement. The results brought by the study should be further compared and put in regards with the existing literature. Similarly, the literature review should be supplemented. The discussion could be improved including more context about green hydrogen discussion and options within EU.

Response to Comment #1

We are grateful to the reviewer for the encouraging and insightful feedback on our research. The suggestions have been instrumental in refining our manuscript.

In response to the comments and recommendations from the reviewers, we have thoroughly updated our manuscript. The revised version now includes a more explicit explanation of our methodology, supplemented with additional figures and equations to elucidate the system boundaries and the optimization model. We have enhanced both the *Methods* and *Main* sections of the article, providing a concise summary of the critical model assumptions and a description of the model. These summaries serve to clarify our approach, with further comprehensive details available in the *Methods* and Supplementary Information sections.

We have incorporated the latest data into all input parameters to ensure the representation of the advancements in low-carbon hydrogen technologies is as precise as possible. Despite significant updates to the model, the consistency of our results with our initial findings gives credibility to our work and confirms the reliability of our methodology.

Our findings have been benchmarked against existing research and industry projects, underscoring the robustness of our results. We have also meticulously outlined current and proposed regulatory frameworks and certifications for low-carbon hydrogen at both national and international levels. This detailed account ensures our model assumptions are well-contextualized within the current policy landscape, making our research timely and relevant to ongoing discussions.

Furthermore, we have expanded our team to include a new co-author, an industry specialist associated with the Ammonia Energy Association. Their expertise in the current state-of-the-art and future perspectives for the decarbonization of the European ammonia industry has significantly enriched the substance and applicability of our study, as well as supporting the validity of our methods.

We are confident that these revisions have greatly improved the robustness and impact of our paper.

Abstract

Comment #2

You state that “electrolytic hydrogen” cuts emissions by 90%, maybe it would avoid later confusion to precise with more details how the electrolyser is powered in your study (off-grid wind turbines and solar, grid electricity with guarantee of origin, mix of both?)

Response to Comment #2

Thanks for the reviewer’s valuable comment. In the revised *Abstract*, we explained that hydrogen is produced “*via water electrolysis powered by dedicated renewable energy sources with grid backup.*” - Lines 8-9.

We also included a detailed description of the electrolytic hydrogen production system (EHPS) in the *Main*, subsection *Hydrogen production system and model description*:

Lines 126-143: “This study assumes that the existing European ammonia plants will be retrofitted by replacing the SMR production system with an electrolytic hydrogen production system (hereafter EHPS). The EHPS includes solar photovoltaic panels (PV), wind turbines (WT), electrolyzers, battery energy storage systems (BESS), hydrogen compressors, and high-pressure tanks for hydrogen storage (**Error! Reference source not found.**). Two EHPS configurations are modeled: (i) when the EHPS is mainly powered by renewable resources but also maintains a grid connection for backup, it is defined as semi-islanded configuration with hybrid (PV and WT) renewable power generation. (ii) When the plant is islanded or off-grid, operating entirely independently of any grid connection, relying solely on its renewable power generation (**Error! Reference source not found.**).

Error! Reference source not found.**Error! Reference source not found.**

Fig. 1 | System boundaries for the electrolytic hydrogen production system (EHPS) and the ammonia plant. The EHPS is primarily powered by newly dedicated renewable installations (semi-islanded with grid connection for backup), with the 0-cap scenario being an off-grid exception with no grid electricity import.”

This approach aligns with the requirements of the “European Union's 'Fit for 55' package, particularly the Renewable Energy Directive II delegated act²⁹ (SI – Text i). The act sets out requirements for additionality, as well as temporal and geographic correlation, for electricity used in hydrogen and fuels derived from it” - **Lines 149-150, SI – Text i**. In doing so, we mirror existing and proposed policy guidelines to generate insights that can inform ongoing policy efforts.

We hope that our revision can help readers better understand our modeling of electrolytic hydrogen for ammonia productions.

Comment #3

You talk about emissions: which scope are you looking at? And is it CO2 or CO2e?

Response to Comment #3

Thank you for raising this point. In the revised *Abstract* we explained that the emission caps refer to CO2e:

Lines 12-14: “However, an optimal lifespan average well-to-gate emission cap of 1 kg carbon dioxide equivalent (CO2e)/kg H2 achieves even deeper emissions reduction of 95%...”

In the *Main*, subsection *Hydrogen production system and model description*, we explain in detail the scope considered in our analysis:

Lines 166-174: “*Well-to-gate emissions encompass Scope 1 (direct emissions from operations, negligible in electrolytic production), Scope 2 (indirect greenhouse gas emissions from the generation of purchased electricity), and partial Scope 3 emissions (upstream activities like the extraction, refining, and transport of fuel used for electricity production, hereafter “Scope 3 upstream emissions”)* (see *Methods*, subsection “*Grid electricity price and carbon intensity*”). *Other Scope 3 emissions, such as those embedded in technology manufacturing (hereafter “Scope 3 embedded emissions”), are excluded, aligning with most low-carbon hydrogen certification systems (SI - Table ii) and the emission accounting framework for low-carbon hydrogen outlined by the International Partnership for Hydrogen and Fuel Cells in the Economy (IPHE)*²².”

Our definition of boundary and scopes of greenhouse gas emission is consistent with existing frameworks, which enhances the relevance and impact of our work, aiding policymakers in defining cost-effective low-carbon hydrogen.

Accordingly, we use CO_{2e} in our analysis and result interpretations. It should be noted that some references we cited only talked about CO₂, and we carefully use the right terms in the revised manuscript.

Comment #4

When you say “cap”, is it based on hourly emissions? Or yearly emissions and then averaged? Please be more precise on the description.

Response to Comment #4

Thanks for the reviewer’s comment. We agree that the definition of the “cap” should be clarified to avoid confusion. In the revised *Abstract* we explained that we study emission caps on the lifespan average well-to-gate CO_{2e} emissions – **Lines 12-14**.

We added a detailed description of the carbon emission caps in *Main*, subsection *Hydrogen production system and model description*:

Lines 164-166: “*Emission caps are set to limit the average well-to-gate CO_{2e} emissions from the hydrogen produced throughout the lifetime of the plant (hereafter simply “emissions”) in the optimization models (see Methods, subsection “Carbon emission caps”).*”

To provide further clarity on this point, we have also included the following equations in our manuscript: **eq. 3** and **eq. 4** in *Methods*, subsection *Carbon emission caps*:

Lines 707-716: “*The well-to-gate CO_{2e} content for each unit of hydrogen Γ_{H2} is calculated by dividing the total lifetime operational emissions from grid-imported electricity ($\sum_{l=1}^L \sum_{t=1}^T \gamma_E M_{E,t}$) by the total amount of hydrogen produced over lifetime of the EHPS ($\sum_{l=1}^L \sum_{t=1}^T D_{H2,t}$) (**Error! Reference source not found.**)*.”

$$\Gamma_{H2} = \frac{\sum_{l=1}^L \sum_{t=1}^T \gamma_E M_{E,t}}{\sum_{l=1}^L \sum_{t=1}^T D_{H2,t}} \quad \text{eq. 1}$$

where $M_{E,t}$ is the imported grid electricity at time t . $D_{H2,t}$ is the hourly hydrogen output of the EHPS system, and γ_E is the average life-cycle CO_{2e} emission intensity of the grid from 2025 to 2050 calculated for each country, based on the historical data provided by the European Commission. Thus,

the EHPS can import grid electricity as long as the average well-to-gate CO₂e emissions of hydrogen do not exceed the emission cap constraint considered in this study (3-cap, 1-cap, 0.5-cap, 0.1-cap, 0-cap) (**Error! Reference source not found.**).

$$\sum_{l=0}^L \sum_{t=0}^T \gamma_E M_{E,t} \leq \varepsilon_{H_2} \left(\sum_{l=0}^L \sum_{t=0}^T D_{H_2,t} \right) \quad \text{eq. 2}$$

where ε_{H_2} represents the emission cap under consideration in kg CO₂e/kg H₂.

We understand that the time duration requirement of emission caps (e.g., hourly, monthly, yearly, or life span) can affect the LCOH. Our research framework can be modified in future work to accommodate these varying time durations for emission caps in a dedicated study. - **Lines 600-605**

Introduction

Comment #5

The literature review could be further expanded regarding techno-economic and environmental assessment of hydrogen/ammonia/methanol/FT-diesel. Mentioning the literature focusing on e-fuels LCA may also be relevant. Also please check and mention how studies focusing on global mapping of hydrogen/ammonia/other which does not have “limited number of locations” cover this topic.

Response to Comment #5

Thank you for the reviewer’s comment. We included new paragraphs in the revised *Introduction* that summarizes the techno-economic and environmental analysis of electrolytic hydrogen to contextualize our study and underline the research gap. We also included global scale analysis of electrolytic hydrogen, which highlight the importance of examining the feasibility at a regional level of the transition to electrolytic hydrogen, aspect that thoroughly included in our study.

Lines 30-35: “Using renewable energy for water electrolysis is considered a long-term sustainable pathway to produce low-carbon hydrogen for uses like transport, power, and energy storage. While direct electrification is often more cost-effective for road transport¹³, electrolytic hydrogen, and synthesized e-fuels are low-carbon alternatives for shipping¹⁴ and aviation¹⁵, where other options are currently limited. In addition, electrolytic hydrogen can be utilized in Power-to-X and Power-to-H₂-to-Power systems¹⁶, for long-term energy storage and grid stabilization.”

Lines 79-104: “Research into the economics of electrolytic hydrogen and ammonia production has been extensive, encompassing detailed system-based analyses and broad global reviews. *Campion et al.*³¹ assessed hydrogen systems in three ammonia plants worldwide, finding the most cost-effective strategy combines local renewables with grid electricity, with emissions tied to grid carbon intensity. *Nayak-Luke and Bañares-Alcántara*³² expanded the scope to a 534 location worldwide finding that by 2030 many could produce ammonia at costs competitive with fossil fuels, with production flexibility being crucial for reducing expenses.

Operational flexibility has been pinpointed as a method to decrease hydrogen production costs, as outlined by *Guerra et al.*³³. *Wang et al.*²³ and *Fasihi et al.*³⁴ demonstrated that flexible ammonia plant operations could mitigate costs and overcapacity^{23,35}. Despite this, current ammonia production through the HB process exhibits limited adaptability, necessitating technological advancements. An alternative approach involves connecting plants to the electricity grid in a semi-islanded configuration, which can potentially reduce costs. However, this strategy risks increasing emissions unless properly constrained by emission-limiting policies.

*Salmon and Bañares-Alcántara*²⁵ studied electrolytic ammonia production in Australia, focusing on the economic and emission implications of grid connectivity. *Terlouw et al.*²⁷ investigated semi-islanded

hydrogen systems in renewable-rich islands concluding that can be both cost-effective and have low environmental burden under a specific emission cap. Ricks et al.³⁶ highlighted a potential pitfall: grid-connected electrolysis, although compliant with US “clean-carbon” regulation, may increase emissions compared to fossil-based hydrogen unless it is matched hourly with clean energy.

The shift to electrolytic production appears to be feasible, but locally challenging. Bartels et al.³⁷ showed that large-scale electrolysis facilities may have local grid impacts when only relying on grid electricity. Kakoulaki et al.³⁸ pointed out that despite abundant national renewable resources, regional shortages might arise. Lastly, Gabrielli et al.³⁹, Rosa and Gabrielli⁷ and Tonelli et al.⁴⁰ illustrated how, despite global resources exceeding the amount necessary for electrolytic production, local scarcities of land and water may pose significant hurdles.”

We hope this can help the reader better understand the motivation and contribution of our study.

Comment #6

You say, “examines all European ammonia production plant”. Maybe mention also here how many ammonia plants you are talking about.

Response to Comment #6

Thanks for comment. The number of plants is specified in the revised *Introduction*, **Lines 118-120**: “*The study analyzes hydrogen production across 38 major European ammonia plants (a list of ammonia plants and locations is provided in SI – Table i and SI – Text ii), factoring in regional costs and historical weather data.*”

Main

Comment #7

You present some caps with value that seem a bit arbitrary at first reading. Comparing with caps discussed with existing (or in discussion) regulations could be relevant. Comparing with emissions from SMR also. And why using only these 4 caps? Please also precise if you are talking about a yearly cap (yearly amount of CO₂e divided by yearly H₂ production), an hourly cap or what?

Response to Comment #7

Thank you for the reviewer’s comment. In the Response to Comments #3 and #4, we explained the definition of emission caps including boundary, scopes, and timeframe.

In the revised *Main*, subsection *Hydrogen production system and model description*, we have reviewed current and anticipated regulatory frameworks to provide a clearer context for our readers, updating the manuscript accordingly – **Lines 126-195**.

We included an additional emission cap according to the EU taxonomy in Europe, which stands at 3 kg CO₂e/kg H₂ – **Lines 70-71**. As this target is set to be revised in 2024, it is crucial to study this alongside the 1, 0.5, 0.1, and 0-caps already considered in the submitted manuscript. These 5 caps are chosen to proposed certification and regulations for hydrogen – **Lines 159-160, SI - Table ii**. The aim is to probe the implications of increasingly stringent and ambitious targets on the technical and economic feasibility of electrolytic hydrogen production. Importantly, “*any caps higher than 3 kg CO₂e/kg H₂ are effectively encompassed by the ‘no-cap’ scenario*” – **Lines 162-163**. In the European Union context, testing any caps higher than 3 kg CO₂e/kg H₂ would not change the optimization output. We explain this in the revised *Main*, subsection *Electrolytic hydrogen results in deep industry decarbonization, even without emission caps*:

Lines 304 – 318: “*Without any emission target, on average, 32% of the EHPS’s total annual energy demand comes from the grid (SI – Fig. iv). One reason for the small fraction of grid electricity import even when certain emission caps would allow more lies in lower leveled cost of renewable electricity*

(LCOE) from newly installed infrastructure as compared to grid electricity. For example, an EHPS fully powered by grid electricity, using the average European grid electricity price of 115 EUR/MWh, would result in an LCOH exceeding 7 EUR/kg H₂, which is 75% higher than the average LCOH of semi-islanded plants. Most European ammonia plants exhibit similar LCOH under both the 3-cap and the no-cap. In fact, 30/38 plants emit less than 3 kg CO₂e/kg H₂ with electrolytic hydrogen production, even without the imposition of emission caps.

[...]

Replacing SMR with water electrolysis for all the European ammonia plants results in an average emission reduction of about 85%, even without enforcing any emission caps attributable to the average well-to-gate carbon content of hydrogen being 1.85 kg CO₂e/kg H₂ (Fig. 4a and SI – Table iv).”

The emissions associated with electrolytic hydrogen production are compared to SMR. We provided more details about SMR emissions:

Lines 175–181: “The well-to-gate emissions from electrolytic hydrogen production are compared with the ones from SMR. The direct emissions (Scope 1) of hydrogen production from natural gas through SMR are around 9-10 kg of CO₂/kg H₂²². Further emissions occur in natural gas production, processing, and transport. Scope 3 upstream emissions for natural gas can vary widely on a country base, depending on production methods and emission mitigation efforts (median value 2.4 kg CO₂e/kg H₂²²). Therefore, considering direct and upstream emissions, the total well-to-gate emissions from hydrogen production via SMR are 12 kg CO₂e/kg H₂.”

We hope our revision can help readers better understand our selections of various emission caps, as well as emissions from SMR.

Comment #8

A paragraph explaining in greater details what is included or not in the emission accounting, the boundary of the system, etc... would be welcomed. Also, how you are accounting for the emissions from the grid?

Response to Comment #8

We appreciate the comment. Indeed, it is important to clarify this aspect. As explained in Response to Comment #3, our emission caps refer to the lifespan well-to-gate Scope 1, Scope 2, and partial Scope 3 (upstream emissions) CO₂e emissions. As explained in Response to Comment #2, the EHPS is primarily powered by newly dedicated renewable installations (semi-islanded with grid connection for backup), with the 0-cap scenario being an off-grid exception with no grid electricity import.

In the revised *Methods*, subsection *Grid electricity price and carbon intensity*, we clarified which types of emissions are considered in the grid:

Lines 939-947: “Regarding emissions from the grid, historical data and future projections were collected from the JRC-COM-NEEFE (National and European Emission Factors for Electricity Consumption) dataset provided by the European Commission⁹⁴. The dataset contains data from 1990 to 2020 for all countries globally. The indirect emissions from electricity consumption are calculated by dividing total national CO₂e emissions from electricity production from all input energy carriers by the total final electricity consumption. It, therefore, includes upstream emissions while excluding emissions from manufacturing technologies, thereby attributing 0 CO₂e per kWh when electricity is generated from renewable sources. The future carbon content of grid electricity is calculated based on the European Environmental Agency (EEA)'s 2030 projections⁹⁵, coupled with the assumption of achieving carbon neutrality by 2050.”

Comment #9

Maybe say that you are using a hybrid set-up (grid + local power) in a more explicit way and explain to the reader what it means for example in terms of grid tariffs, connection costs, electricity price, etc...

Response to Comment #9

We appreciate your comment and concur with your suggestion. In response, we have expanded our explanation of the hydrogen production system's boundaries and included a descriptive figure to enhance reader comprehension, as detailed in Response to Comment #2.

In the revised manuscript we clarified that we included grid connection and dedicate transmission lines cost and losses in our LCOH calculations – **Lines 145-150**.

First, we explained in the *Methods*, subsection *Grid connection upgrades*, that:

Lines 740-753: *“Existing fossil-based ammonia plants are already grid-connected, albeit with a few MW of power capacity. However, when retrofitting these plants—replacing SMR hydrogen with electrolytic hydrogen production—the peak capacity demand from the grid will increase. This means that when a large amount of energy is required, the grid should be capable of supplying it. As a result, grid connection upgrades to accommodate this increase in peak capacity demand are included in the LCOH calculation. This includes the installation of new high voltage alternating current (HVAC) wires and a transformer to step down from high voltage to low voltage at the plant side (unit cost in SI - table xii).*

[...]

Given the typical location of existing plants in industrial areas, a proximity to the grid connection point (within 5 km) is assumed, and line losses are negligible due to the short distance. Hence, only transformer losses (1% HV-LV) and rectifier losses (5% at plant side) are included.”

The shift to electrolytic hydrogen production does increase the demand of electricity compared to SMR. However, this increase is unlikely to impact grid prices or cause grid congestions:

Lines 477-481: *“While this increased electricity consumption necessitates infrastructure upgrades on the plant side, as previously discussed, the plant's energy demand is relatively small, representing, on average, 6.3% of the current energy demand at the regional level and 0.8% at a national level. This increase is unlikely to have a significant impact on regional consumption patterns, causing grid congestions, or to influence price dynamics notably³⁷.”*

We have also clarified that the total installed cost for utility-scale PV and wind turbines taken from the International Renewable Energy Agency (IRENA) incorporates connection and wiring costs in *Methods*, subsection *Renewable power generation*:

Lines 844-853: *“Given the assumption that the renewable installations are located within short distances (<10km), HVAC is assumed to transmit electricity from PV and WT to the plant. Wind Turbines typically generate LVAC, which must be stepped up to HV using transformers, transferred to the plant, stepped down, and then converted into Low Voltage Direct Current (LVDC) using rectifiers. Conversely, PV systems produce LVDC, requiring an additional step to convert LVDC to LVAC before stepping up and integrating into the dedicated HVAC line. Costs associated with inverters are already included in the IRENA dataset. Costs and losses associated with transformers, wires, inverters, and rectifiers are shown in SI – Table xii. Total transmission losses from the WT to the electrolysis hydrogen system are around 7% (2% transformers, 5% rectifier), and for PV systems, around 10% (3% inverter, 2% transformers, 5% rectifier).”*

Comment #10

Please explain clearly from the beginning how you derive the electricity prices. Is it historical prices? (from which year?). Can you give some numbers? Maybe refer to some duration price curves.

Response to Comment #10

We agree with your comment and have taken steps to provide additional clarity regarding the electricity price assumptions in our model.

As detailed in the updated *Methods*, subsection *Grid electricity and carbon intensity*, we highlighted the method for projecting electricity prices in the future and the dataset used for projections:

Lines 907-938: *“Modelling the price behavior of grid electricity is challenging; electricity prices vary both on a daily and long-term scale⁹⁰. Random events, such as load variations, contingencies, network congestion, and changes in demand, can cause prices to fluctuate throughout the day. In the long run, additional factors such as oil price changes, regulatory policies, political intervention, technological changes, energy mix variation, and grid operations can drastically influence long-term electricity prices⁹¹. As these factors are difficult to anticipate, the present work focuses on long-term forecasting and neglects short-term fluctuations. One method that has been employed in this context is stochastic process modeling. Geometric Brownian motion (GBM), also known as exponential Brownian motion, is a continuous-time stochastic process in which the logarithm of the randomly varying quantity follows a Brownian motion with drift. GBM is one of the most applied stochastic processes for long-term electricity price forecasts where future values are calibrated on historical time series⁹². Hence, in this study, Monte Carlo simulations and GBM were used to project future electricity prices from 2025 to 2050. The methodology was applied to a dataset provided by the European Commission that incorporates country-based monthly retail and wholesale historical electricity prices for industrial users, spanning from January 2008 to December 2019⁹³.*

[...]

The process creates a distribution of possible future price paths for each month until 2050, reflecting the inherent uncertainty in these projections. Once these potential future price paths are established, the 5th, 50th, and 95th percentiles of the simulated prices are used to construct each country’s optimistic, average, and pessimistic electricity prices. The 5th and 95th percentiles serve as lower and upper bounds, indicating a 90% confidence interval, while the 50th percentile represents the median price. Finally, the average prices over the period from 2025 to 2050 are calculated for the 5th, 50th, and 95th percentiles and used as input for the optimization (SI - Table xii). Grid electricity prices vary from 30 EUR/MWh in Norway to 234 EUR/MWh in the UK, with a mean value of 115 EUR/MWh in the reference case.”

We recognize the price duration curve to be a valuable tool for integrating existing grid operations into our optimization model. Implementing an effective dispatch strategy could further reduce the LCOH. While our analysis currently concentrates on multi-regional assessments, we believe focusing on specific regions with detailed grid data could yield more precise LCOH validation, although it may not be technically feasible in our broader study.

In our manuscript's robustness analysis section, we have comprehensively detailed how grid price dynamics can affect the LCOH, particularly if plants leverage low-cost electricity from the grid. Additionally, we have quantified the proportion of electricity that hydrogen production consumes relative to total regional electricity consumption. This measure helps assess feasibility across regions and can guide future localized studies that aim to optimize dispatch strategies, integrate with current grid operations, minimize LCOH, and prevent disruptions to the existing grid.

Comment #11

You are talking about hydrogen used for ammonia plant: a description on how the ammonia plants are operated (at full load all the time, in a flexible way?), how they are retrofitted for electrolytic hydrogen use?, how much it cost to retrofit (or do you need new ammonia plants?) would be needed to understand how the hydrogen is produced.

Response to Comment #11

Many thanks for the important feedback. First, as explained in Comment #2, we included a new figure (Fig. 1) describing both the key subsystems of the EHPS and electrolytic ammonia plant.

Second, we have included explanations on how ammonia plants are operated in *Methods*, subsection *Ammonia production process and EHPS*:

Lines 639-645: “Ammonia plants operate continuously at full load⁵⁴. The HB process has been optimized over decades for steady-state operation, requiring consistent reactant supply, product removal, and specific temperature and pressure condition⁵⁵. This standard has been enabled by the dispatchability of natural gas. In line with this, even when switching to electrolysis for hydrogen production, it is assumed to maintain the continuous plant operation. This approach aligns with many of today’s large-scale electrolytic plant projects (an example is the Greenko, GIC, and Gentari partnership for renewable-based electrolytic ammonia production in India⁵⁶).”

We also discuss the implications of the plant's flexibility:

Lines 651-653: “Despite recent efforts by technology licensors to develop more ‘flexible’ plants that can adjust operations in accordance with renewable energy availability, achieving this flexibility at a large scale presents inherent challenges, particularly with the air separation unit and the ammonia synloop^{23,35,54}.”

In this paper, we propose “retrofitting current ammonia plants by substituting the existing SMR production system with an Electrolytic Hydrogen Production System (EHPS).” – **Line 126-128**. We have included a comprehensive evaluation of retrofitting costs with the assistance of an industry expert as a new author contributing to the study:

Methods, subsection *Grid connection upgrades*, **Lines 284-289:** “These expenses include the upgrading and replacement of specific components (such as electric start-up heaters and steam generators) to ensure optimal operation with electrolytic hydrogen. They also involve enhancements of transmission lines, which are needed to accommodate an increase in grid demand. Furthermore, the sunk costs associated with decommissioning the SMR hydrogen production system are included in these retrofitting expenses.”

Methods, subsection *Technology input data*, **Lines 740-753:** “Existing fossil-based ammonia plants are already grid-connected, albeit with a few MW of power capacity. However, when retrofitting these plants—replacing SMR hydrogen with electrolytic hydrogen production—the peak capacity demand from the grid will increase. [...] As a result, grid connection upgrades to accommodate this increase in peak capacity demand are included in the LCOH calculation. This includes the installation of new high voltage alternating current (HVAC) wires and a transformer to step down from high voltage to low voltage at the plant side (unit cost in SI – table xii). To account for energy losses in transmission, an increase in the price of delivered electricity has been incorporated.[...] The additional power required to compensate for these transmission losses leads to a proportional increase in the cost of power.”

Methods, subsection *Retrofitting costs*, **Lines 891-906:** “Retrofitting costs include the substitution and upgrade of some plants' components for operations with electrolytic hydrogen. Current research and practical applications, such as the Puertollano project⁸⁹, have shown that up to 10-15% of hydrogen derived from electrolysis can be integrated without any modifications. However, to increase this ratio, adjustments are necessary due to changes in heat flows and operational flexibility. For instance, steam from the ammonia converter cannot be used for the SMR anymore. Anticipated modifications include replacing the electric start-up heater before the ammonia converter, installing additional electric heaters to facilitate flexible operation, and incorporating steam generators for periods of low-load operation when the ammonia converter generates insufficient steam. The estimated cost for these modifications is about 5-10% of the cost of installing a new ammonia loop (detailed calculation in SI – Text ix).

The analysis also includes the sunk costs associated with decommissioning the SMR hydrogen production system. These sunk costs represent expenses that have already been incurred and cannot be

recovered. However, the potential offset of these sunk costs is also considered by incorporating the estimated residual or scrap value of the decommissioned SMR. It is anticipated that this residual value will be around 10% of the SMR's original cost (detailed calculation in SI – Text ix)."

It is important to note that *"the costs associated with retrofitting the hydrogen system constitute a relatively minor part of the total expenditure, accounting for approximately 0.5% of the overall cost (SI – Fig. ii)."* - **Lines 283-289.**

To summarize, in the revised version we did our best to clarify how an ammonia plant operates, the boundaries of the EHPS, and how ammonia plants are retrofitted for electrolytic production.

Comment #12

Please discuss how to additional load induced by the extra electrolyzers will be handled by the existing grid. May it have any influence on the electricity price? Which power unit may be used to cover the additional load?

Response to Comment #12

Thanks for the comment. We have now elaborated on this topic in our revised version. Except for the 0-cap (off-grid), the hydrogen production system described operates semi-islanded, meaning most of its energy demand, primarily for powering electrolyzers, is met by newly installed renewable installations. However, during periods of low renewable capacity, the plant may occasionally depend on grid imports to maintain continuous operation.

In the updated feasibility analysis we have studied the impact of grid connection on local grid, specifically under the 1-cap:

Lines 474-486: *"Results show that, under the 1-cap, the EHPS mean grid electricity demand is 491 GWh per year (min 26 GWh, max 1,488 GWh across Europe). The shift to electrolytic hydrogen production does increase the demand for electricity compared to the SMR process (on average, 56 MWh are imported from the grid each hour). While this increased electricity consumption necessitates infrastructure upgrades on the plant side, as previously discussed, the plant's energy demand is relatively small, representing, on average, 6.3% of the current energy demand at the regional level and 0.8% at a national level. This increase is unlikely to have a significant impact on regional consumption patterns, causing grid congestions, or to influence price dynamics notably³⁷. Nevertheless, while the overall energy demand of the grid may not be significantly affected, the peak power demand during operation can be substantial³⁷ (see Methods, "Feasibility analysis"). There is a need to investigate how regional or local grids will adapt to the increased demand of EHPS. This includes the necessary expansion of power generation capacity and the adjustment of pricing policies to ensure that hydrogen production via electrolysis remains cost-effective."*

Our feasibility analysis serves as a basis for future works to study how regional or local grid should prepare for the increased demand of electrolytic ammonia plants including the expansion of power generation, and price strategy. Such study however should be conducted using more granular grid data.

Comment #13

In Fig.3 why not include LCOH SMR lines in all graphs?

Response to Comment #13

We appreciate your feedback. In response, we have included the LCOH for SMR in all subfigures of Fig. 3 (Fig.4 in the revised manuscript) as well as updated all the results – **Fig. 4, Lines 344-349:**

Fig. 1 | Cost-effective emission cap. a, Minimum LCOH and area needed for system installation (size of circles) as function of carbon emission reduction for all ammonia plants in Europe. Pessimistic and optimistic scenarios in SI - Fig. vi. **b**, Malopolskie (PL21). **c**, Sør-Østlandet (NO03). **d**, Észak-Magyarország (HU31).

Comment #14

Did you include the retrofitting of existing ammonia plants in your abatement cost? It would be relevant to include or at least discuss it.

Response to Comment #14

We agree that retrofitting costs should have been more extensively discussed in the manuscript. As detailed in Response to Comment #11, retrofitting costs have been included in both LCOH and abatement cost.

Comment #15

Looking at results from Table 1 some of the results are surprising and would be worth some explanations: in most places AC goes from lowest with no-cap to maximum with 0-cap with a growing or constant trend, however in Poland it goes up and down, why?

Response to Comment #15

We appreciate your observation on the trends in abatement cost (AC), particularly the trend observed in Poland. We have further explained it in the revised manuscript:

Main, subsection The 1-cap is feasible and cost-effective emission reduction target, Lines 351-366: “The AC is calculated by dividing the difference between the LCOH of electrolytic hydrogen and the LCOH of SMR hydrogen (reference LCOH being 2 EUR/kg H₂) by the difference in their carbon content. In essence, this ratio quantifies the additional expenditure per ton of CO_{2e} abated (see Methods, subsection “Abatement Cost”).

[...]

More stringent caps induce higher AC, except for regions with carbon-intensive grid electricity. For example, Polish region PL42 records an average LCOH of 2.98 EUR/kg H₂ under no-cap, and a 52% reduction in emissions compared to hydrogen produced via SMR. The AC is 152 EUR/ton CO_{2e}. While a stricter 3-cap causes a cost increase of only 7%, avoided emissions increase by 44% compared to the no-cap scenario, and the estimated AC drops to 132 EUR/ton CO_{2e}. The trend observed in Poland is a result of the large increase in avoided emissions compared to the relatively small increase in cost when the first cap is implemented, ultimately leading to a lower AC.”

As demonstrated by this example, the AC not only depends on the cost differential between electrolytic and SMR hydrogen but also on the effectiveness of the emission caps in reducing emissions. This pattern is not observed in regions with a less carbon-intensive electricity mix. In these regions, implementing more stringent emission caps results in only a minimal reduction in emissions, but incurs higher costs. Thus, the abatement cost in these areas tends to increase with more stringent emission caps.

Comment #16

It would be nice to have some comparisons about what 61 km² or 342 km² represents i.e. compared to existing PV plants, city sizes, whatever. What is the “available land” in the region? Please explain in a few words and/or refer to the method section

Response to Comment #16

Thanks for the comment. Indeed, we agree is better to give early on a definition of available land to help the reader understand the results of the feasibility analysis.

In our study, the term "available land" refers to the area within a specific region that can be used for the installation of renewable power generation systems such as photovoltaics (PV) and wind turbines (WT) to power plants' operations. We have updated the revised manuscript to provide a more detailed explanation of how it is calculated:

Main, subsection The transition to electrolytic hydrogen is challenging for the 0-cap, Lines 495-498: “Additionally, the installation would occupy, on average, 5.2% (min 0.4%, max 21.4%) of the available land in the region under the 1-cap. The computation of available land adopted the methodology described by Gabrielli et al.³⁹, utilizing country-specific land data provided by the Food and Agriculture Organization of the United Nations (FAO)⁵³, as detailed in the methods section.”

Methods, subsection Feasibility analysis, Lines 1033-1037: “Using Gabrielli et al.³⁹ methodology and FAO’s 2020 land data, available land was estimated by excluding forest areas⁹⁹. It is assumed that 70% of non-forest land is suitable for renewable installations, in line with global usable land estimates that consider institutional and biophysical constraints⁹⁹. This land is then compared with the requirements for PV and WT installations in each region, considering the mean land availability of the corresponding region.”

In this revision, we updated our model, data and results based on the valuable comments from reviewers. The values of land requirement are updated accordingly, and we also use make examples to help understand the size of land requirement:

Main, subsection The transition to electrolytic hydrogen is challenging for the 0-cap, Lines 503-510: “These results point to land area required for renewable installations as one major area of concern when considering the production of hydrogen from renewables, since ammonia plants require massive

installations (on average 206 km² for the 1-cap and 405 km² for the 0-cap), with 1.4 and 2.5 times these capacities in the low-renewable regions. As a reference, the median size of urban area with a minimum population of one million in Europe is 300 km². In comparison, the largest solar park in the world (Bhadla Solar Park in India), has a nominal capacity of 2,245 MW and covers an area of 56 km², while the Asian Renewable Energy Hub in Australia, currently under planning, will span over 6,500 km² for 15 GW solar and wind installations.”

We believe our revision now clarified our methodology for available land calculation and the dimension of the renewable installations required to power electrolytic ammonia plants in Europe.

General comment for the main section

Comment #17

Some of the points addressed in the main paragraph seem to have been mentioned in the method section. In that case, it may help the reader to also summarize the most important assumptions in the main section text and refer to the method section for further details.

Response to Comment #17

Thank you for the feedback. In the revised manuscript, we have summarized and elaborated on the key assumptions and modeling decisions within the *Main* section, referring readers to the *Methods* or *SI* section for a more in-depth explanation.

Comment #18

As you discuss about ammonia in most of the paper, it may be nice to give some ammonia production costs also (LCOA).

Response to comment #18

Thank you for the comment. We recognize that our initial manuscript did not specifically mention ammonia costs. We cited two study about levelized cost of ammonia from SMR and renewable energy in the revised *Introduction*. These sentences made our introduction more coherent.

Lines 58-61: *“The levelized cost of ammonia (LCOA) is significantly higher for electrolytic hydrogen (1,000-2,500 EUR/tons NH₃) than SMR (200-1,000 EUR/tons NH₃)^{22,24}. A lower production cost of clean hydrogen is required for renewable ammonia production to be cost-competitive.”*

Comment #19

Please explain how your results compare with the existing literature.

Response to Comment #19

We have included these comparisons in the revised manuscript, providing a broader context and further validation for our study.

We compared our LCOH analysis with recent literature discussed in the *Main*, subsection *Electrolytic hydrogen can be cost-competitive with SMR except for the 0-cap*:

Lines 200-203: *“The lowest LCOH is with a semi-islanded configuration in Norway (NO03) with 1.99 EUR/kg H₂, while the highest LCOH is 12.61 EUR/kg H₂ for off-grid plants in SK02. This range aligns with recent estimates from IEA²² (2-10 EUR/kg H₂ in Europe).”*

This consistency provides further validation for our study’s methodologies and findings.

Additionally, we have examined preliminary estimates from real-world projects regarding total cost, production, and installed capacities and found a strong similarity with our results, suggesting that our study not only aligns with academic research but also with practical applications in the field.

Lines 204-210: *“Over the course of a 25-year plant lifetime, the total cost of the EHPS, comprising both capital expenditures (CAPEX) and operational expenditures (OPEX), averages 6.7 billion EUR for the no-cap scenario, with OPEX making up 55% and CAPEX 45% of the total (SI – Fig. ii). In contrast, for the 0-cap scenario, the total cost rises to approximately 10.8 billion EUR, with a higher proportion attributable to CAPEX at 65% and a smaller portion to OPEX at 35%. These estimates align with anticipated investments for major proposed projects in renewable-based ammonia production, which range between 4 and 11 billion euros⁴¹⁻⁴⁴.”*

We have included these comparisons in the revised manuscript, providing a broader context and further validation for our study.

Discussion

Comment #20

In general, it would help the reader to be more precise when describing the power supply you are talking about. For example, when you say “powered by dedicated renewables” do you mean off-grid plants? Or still a bit of grid can come in?

Response to Comment #20

We agree this aspect requires further clarification. As discussed in responses to Comment #2 and Comment #9, electrolytic ammonia plants are assumed to be 'semi-islanded', primarily relying on renewable energy installations located near the plant. Grid electricity is only used as a backup for balancing periods.

Only in the case of a zero-emissions cap (0-cap) plants are 'islanded' or off-grid. In this case, the electrolytic hydrogen systems are disconnected from the grid and rely solely on their dedicated renewable energy installations for power.

As discussed in Response to Comment #17, we include a new section in the beginning of the *Main*, to explain our electrolytic hydrogen production system, emission definitions, and key assumptions. We also refer readers to the *Methods* or *SI* for a more in-depth explanation.

Comment #21

Some more detailed explanation on why you are getting some results would be welcomed. For example (among other examples), H₂ has a low carbon content in Spain: why? Is it because, Spanish grid is “clean” due to a lot of solar/hydro/wind in the mix? Is it because fossil electricity is expensive so mostly local renewable power is used? Same type of questions could be answered to explain some LCOH results.

Response to Comment #21

Thanks for your comment. We updated our model, data, and results, and in our revision, we include more explanatory paragraphs. Hereafter some examples:

Main, subsection *Electrolytic hydrogen can be cost-competitive with SMR except for the 0-cap*,

Lines 211-220: *“The cost of imported electricity is the most significant expense under both the no-cap and 3-cap scenarios, comprising 30% and 25% of the total cost, respectively (SI – Fig. ii). Consequently, in the absence of any emission thresholds (no-cap), the lowest LCOH is recorded in Norway and Poland (1.99 EUR/kg H₂ in NO03; 2.98 EUR/kg H₂ PL42; and 3.05 EUR/kg H₂ PL61), mainly driven by the below-average price of electricity. The weight of grid import diminishes rapidly with the enforcement of stricter emission caps. Despite regional variations in local cost components and renewable capacity factors, a direct correlation exists between the LCOH and the average price of*

grid electricity in the region where the plant is situated. However, this correlation weakens as the emission caps become more stringent, dropping from an *r*-squared value of 0.55 in the no-cap scenario to 0.01 in the 0.1-cap scenario (SI – Fig. iii).”

Lines 228-238: “[...] in regions with cheap, low-carbon grid electricity, such as Alsace, Sør-Østlandet (NO03, highlighted in Fig. 2b), and Aragón, experience negligible increases in LCOH with a more stringent emission cap. NO03 is the only region where electrolytic hydrogen is estimated to be cost-competitive with SMR hydrogen [...]. However, a 0-cap brings about a sharp increase in the LCOH. For example, the ammonia plant in Sør-Østlandet presents the lowest LCOH of all plants, with above-average use of grid electricity (54% of the electricity comes from the grid); here, grid electricity has the lowest carbon intensity in Europe and lower price compared to the mean value. However, when the plant is off grid, the installed capacity of wind turbines and electrolyzers increase by 369% and 417%, respectively, to balance the lack of grid backup, resulting in a 236% increase in LCOH.”

Throughout the revised manuscript we added similar in-depth explanation of our key results to help reader interpret our findings. An example:

Main, subsection The largest installations are required to meet the 0-cap, Lines 258-268: “With increasingly stringent caps, the installed capacity of these components exhibits exponential growth (Fig. 3 and SI – Fig. iv). Under the 0-cap, the installed capacity is nearly double that of the average installed capacity under the 0.1-cap. This trend is particularly pronounced in wind-dominated regions. The installed capacity of wind turbines is estimated to approximately triple from the 0.1-cap to the 0-cap scenario, while the installed capacity of electrolyzers quadruples. PV installations, however, grow more steadily from the no-cap to the 0.1-cap scenarios. On average, PV systems constitute 57% of the renewable power capacity for semi-islanded configurations, whereas this share drops to 46% for islanded (off-grid) systems. This result stems from the difference between solar and wind resource profiles. Under the 0-cap, the consistent availability of wind energy leads to an increased use of wind installations, reducing the reliance on solar energy. Consequently, costs in wind-dominated regions surge significantly more than in solar-rich areas.”

Comment #22

You talk a lot about this 1 kg CO₂e/kg H₂ cap or lower but how would it look with higher caps like 1.5 or 2 kg CO₂e/kg H₂?

Response to Comment #22

We appreciate your question concerning the implications of higher emission caps. In the revised manuscript, we offer a more detailed explanation of this rationale.

“Replacing SMR with water electrolysis for all the European ammonia plants results in an average emission reduction of about 85%, even without enforcing any emission cap (the average well-to-gate carbon content of hydrogen is 1.85 kg CO₂e/kg H₂ (Fig. 4a. and SI – Table iv).” – **Lines 316-318**. As explained in Response to Comment #7, “The reason for the minimal use of grid electricity import, even when emission caps would permit increased usage, is due to the lower levelized cost of electricity (LCOE) from newly installed renewable energy sources as compared to that of grid electricity.” – **Lines 306-311**. Hence, in the European context, it is more pertinent to study stringent emission caps, as the shift to electrolytic hydrogen is already reducing emissions significantly even without regulations.

However, in the revised manuscript we also tested the “3 kg CO₂e/kg H₂ emission cap to include the latest EU regulation.” – **Lines 70-71**. As expected, the 3-cap does not result in significant differences in costs and emissions compared to the 1-cap (only a 4% reduction in LCOH on average) nor the no-cap (+2% LCOH).

With this revision we can reiterate that caps higher than 1 kg CO₂e/kg H₂ would not yield additional insights for this study.

Nevertheless, it is acknowledged that this methodology could be adapted to regions outside the EU, where the energy mix is more carbon-intensive and less expensive. In such regions, higher emission caps might be more appropriate to study, as the impact on costs and emissions could be more pronounced and thus more relevant. – Lines 313-315

Comment #23

The discussion section should also be enhanced discussing with a greater extent the following aspects: How does your study fits with the discussed EU regulations/certification schemes for green hydrogen?

Response to Comment #23

Thank you for the suggestion. We substantially revised our manuscript to highlight how our study fits the discussed EU regulatory framework and certification schemes for sustainable low-carbon hydrogen. We have incorporated a discussion of the 'Fit for 55' package and the Delegated Act under the Renewable Energy Directive II, which includes the most updated requirements for hydrogen. We have also included references to the EU Taxonomy and its targets and requirements for low-carbon fuels and hydrogen, including derivatives such as ammonia. – Lines 69-78, and SI – Text i.

In the Response to Comment #7, we provided a detailed update of emission caps, which represents a broad spectrum of current proposals, and is consistent in terms of boundary and scopes of emission accounting. We included a table (SI – Table ii) presenting an extensive comparison of each proposed regulation for hydrogen production, including a description of the scope and boundaries considered. In the *Discussion*, we provided policy suggestions under 1-cap regulations accordingly.

By aligning our study with these evolving regulatory frameworks, we hope to give insights that are both timely and relevant in the context of the EU's transition towards a low-carbon ammonia sector.

Comment #24

You are talking about local retrofitting, but why not looking at H₂ pipeline network within Europe and more global energy system modeling? What differences could raise between your localized study and energy system analysis models?

Response to Comment #24

Thank you for the insightful question. Developing an H₂ pipeline network within Europe is one of the potential solutions. However, this represents a highly challenging research task within global energy system modeling. It requires an understanding of the future demand for renewables by different sectors across various regions, the expansion of renewable generation, and the optimization of demand and supply between renewable electricity and H₂. While this study could be conducted in specific regions with detailed grid data, it remains challenging on a larger multi-regional scale to refine the emission caps for electrolytic hydrogen. Practically, coordinating the expansion of renewable power generation with the construction of an H₂ pipeline network in Europe also poses significant challenges. In short, investigating continental and global hydrogen supply chains is indeed of tremendous importance, but it is also an ambitious endeavour that necessitates a completely distinct model design, data calibration, and dedicated assumptions.

The focus of our current study reflects the structure of the European fertilizer industry as is, where hydrogen is used directly for on-site ammonia production. Several ammonia producers are already examining options to retrofit existing plants, confirming the relevance of this setup.

Our research indicates that with well-defined emission caps and adequate financial incentives, ammonia plants have the potential to initiate their own hydrogen decarbonization processes. As discussed in the Response to Comment #12 about grid expansion and pricing strategies, studying H₂ pipeline network can be a follow-up study to assess other strategies to reduce the LCOH, especially in areas where electrolytic hydrogen and ammonia production is challenging.

To reflect this message, we revised the *Discussion*:

Lines 578-599: *“Conducting the analysis at the regional level can help tailor emission caps and similar policies to local conditions. Some countries present a combination of favorable conditions for the deployment of electrolytic hydrogen that can allow near-zero emissions while bearing minimal or absent cost increases compared to fossil-based hydrogen production. Policymakers may agree to set more stringent decarbonization targets for some countries, while others may be subjected to less stringent or delayed measures. Regionally diversified and phased policy approaches can avoid not excessively penalizing local industries. European governments may also decide to support strategies other than producing renewable electricity in the vicinity of current ammonia plants. Due to the heterogeneity of electrolytic hydrogen costs, it is imaginable that some ammonia producers may consider locating renewable infrastructure in regions with a higher renewable potential to reduce costs. However, this could potentially deplete local renewable resources. Further investigations are warranted to ascertain the regions and conditions under which ammonia plants might be given precedence for the use of limited renewable resources over other sectors.*

Another viable strategy could involve retaining the current plants while directly importing low-carbon hydrogen from regions with a superior renewable potential and lower production costs. The large-scale transportation of hydrogen would necessitate substantial pipeline infrastructure, introducing additional carbon emissions, costs, and risks, factors that could be incorporated into the model. For a detailed regional analysis, it is essential to utilize precise, high-resolution geospatial data to investigate strategies for providing low-carbon, cost-effective electrolytic hydrogen. This work could serve as a foundation for future studies to determine whether and how regions capable of producing low-cost hydrogen could meet the demand from other areas. The study could be extended beyond Europe, considering a global supply chain.”

We hope our revision will provide useful insights for future studies looking into electrolytic hydrogen and ammonia supply chains for cost-effective low-carbon production.

Method

Comment #25

Would not it be possible to retrofit ammonia plant at a limited cost and then operate it semi-flexibly? It would be worth looking in more detail into “flexible ammonia plant” literature. Maybe perform a sensitivity analysis on this parameter if it has a significant influence on the results and mention it in the result/discussion section.

Response to Comment #25

Thanks for bringing this up. Indeed, it is an important question. We already developed a follow-up study regarding the impact of degree of flexibility on the levelized cost of ammonia under different renewable potentials. This requires an upgrade of our model presented in this study to include downstream processes (e.g., air separator unit and ammonia synloop).

To reflect this message, we included a new paragraph in the end of the revised *Discussion* about future study of flexible ammonia production:

Lines 600-614: *“The study can be further improved in multiple aspects to formulate rigorous emission cap regulations for sustainable investment in low-carbon electric hydrogen for ammonia production. [...] Second, investigating the potential of flexible ammonia production is essential. This approach could more efficiently harness intermittent renewable energy, influencing both LCOH and the determination of cost-effective emission caps. [...]”*

We also explained the reason why in this study we assume to maintain continuous operations:

Lines 642-660: *“In line with this, even when switching to electrolysis for hydrogen production, it is assumed to maintain the continuous plant operation. This approach aligns with many of today’s large-*

scale electrolytic plant projects (an example is the Greenko, GIC, and Gentari partnership for renewable-based electrolytic ammonia production in India⁵⁶).

[...]

Despite recent efforts to develop more ‘flexible’ plants that can adjust operations in accordance with renewable energy availability, achieving this flexibility at a large scale presents inherent challenges, particularly with the air separation unit and the ammonia synloop^{23,35,54}. In large-scale plants, air separation units typically employ cryogenic distillation and have a minimum load limit between 50% and 70%, below which extensive ramp-up times would be required²³. The ammonia reactor and compressors also pose operational challenges due to the need to maintain precise conditions, including specific hydrogen-to-nitrogen ratios, temperatures, and pressures^{54,60}. [...] Consequently, while efforts are being made to increase plant flexibility, the prevailing industry trend still favors continuous operation with hydrogen/energy storage support.”

Comment #26

Please quantify the time to run a scenario and the specification of the computer used to do it

Response to Comment #26

Certainly. In SI – Table xi, we now provide the computer specifications used to run the optimization: Processor 11th Gen Intel(R) Core(TM) i7-11800H, 16.0 GB RAM, and the performance of the different optimization algorithms tested (Mixed-integer linear programming (MILP) outperform other options with <1 min/plant).

In the revised *Methods*, subsection *Optimization Model*, we summarized the time for running optimization scenarios:

Lines 702-705: “While heuristic methods, and particularly a hybrid approach combining Differential Evolution (DE) and Nelder-Mead, provided results equivalent to those of the MILP method, they were significantly slower in finding the solution (less than 1 minute for MILP versus approximately 20 minutes for the DE and Nelder-Mead combination; see SI – Table xi).”

Comment #27

When you say that “emissions from the production of renewable facilities [...] is significantly lower than SMR”, it would be nice to supplement with further explanation (or not saying it, if you didn’t look at it with enough details): is it in location where the system is significantly oversized? Does it include land use change emissions? Which share of grid-electricity is used? Does grid electricity also include this additional emissions for solar and wind?

Response to Comment #27

We appreciate the reviewer's valuable feedback.

We developed a new subsection *Scope 3 embedded emissions*, in the revised *Method* to define the life cycle CO₂e emissions from technology manufacturing and transport:

Lines 718-730: “Including emissions from the manufacturing of components introduces significant uncertainty, largely due to the intricate nature of tracing emissions along the comprehensive technology supply chain. This complexity is particularly pronounced for technologies such as electrolyzers, solar PV modules, or wind turbine components, often subject to international trade²². Furthermore, actual emission inventory data are frequently unavailable or inaccessible²².”

While there are inherent complexities in estimating emissions related to the manufacturing of key technology components, several studies offer valuable insights into average emissions⁶⁷⁻⁶⁹. Emissions embedded in technology γ_k are calculated for PV, WT, Li-ion batteries, electrolyzers, hydrogen compressors and storage tanks, and multiplied by the respective installed capacity \dot{P}_k (see SI – Text iv).

It is noteworthy that due to anticipated reductions in the emission intensity of electricity generation, as projected by IEA scenarios²², emissions from material production and technology manufacturing are expected to decrease. Consequently, indirect emissions from materials and manufacturing processes involved in hydrogen production could be less in the future than today.”

In Response to Comments #3 and #4, we explained that the lifespan average well-to-gate CO₂e emissions excludes the Scope 3 embedded emissions. However, we underlined that future regulation should pay the attention to the rebounded Scope 3 embedded emission from technology manufacturing and transport as the emission caps become stricter based on the current lifespan average well-to-gate CO₂e emissions (see **SI – Fig. vii**).

SI - Fig. vii | Average lifecycle emissions (including scope 1, scope 2, partial scope 3 upstream emissions and technology manufacturing) of electrolytic hydrogen in Europe.

Lifecycle emissions, are on average of 3.1 kg CO₂e/kg H₂ for electrolytic hydrogen produced in Europe under the no-cap. The lifecycle emissions reduce to 2.3 kg and 2.2 kg CO₂e/kg H₂ under the 0.5- and 1-caps, and increase again under the 0-cap 3.0 kg CO₂e/kg H₂ (see SI – Fig. vii). It is important to highlight that the emissions values we gathered from LCA studies also include emissions from land use changes. Fig. vii illustrates that including emissions from the manufacturing of renewable installations would increase on average 1.19 kg CO₂e/kg H₂ under the no-cap scenario (56% from WT and 44% from PV) and rise to 2.76 kg CO₂e/kg H₂ under the 0-cap scenario when larger overcapacities are required (67% for WT and 33% for PV).

The reviewer has rightly pointed out that plants situated in regions with low renewable capacity and stringent emission caps will necessitate larger installations if they operate off-grid, which in turn would increase the Scope 3 embedded technology carbon footprint. An example is SK02 under the 0-cap, which requires triple the average installed capacity compared to other European plants subject to the same emissions limitations, with 11.3 GW from solar and 1.4 GW from wind power. Even in this extreme case, the lifecycle emissions would be 6.8 kg CO₂e/kg H₂, marking a 40% reduction when compared to SMR.

In the updated *Discussion*, we have clarified this seemingly paradoxical finding:

Lines 565-577: *“It is crucial to underscore that a 0-cap scenario not only implies higher costs and increased land usage for renewables but can also potentially lead to higher overall lifecycle emissions compared to scenarios with less stringent caps. Present policies do not account for ‘Scope 3 embedded technology’ emissions from component manufacturing, and this study therefore also omits these emissions. However, under the 0-cap scenario, where renewable installations are significantly larger, the total carbon content of hydrogen - when accounting for emissions from component manufacturing (including Scope 3) - is 33% higher compared to the lifecycle emissions under the 1-cap scenario, given global average emissions components as shown in SI – Fig. vii (details about calculation of Scope 3 embedded emissions in Methods, subsection “Scope 3 embedded emissions” and SI, text iv). As we look to the future, the cleaner the grid of EU becomes, the more significant the relative contribution of component manufacturing to overall emissions will become. This suggests that future research could usefully adapt this model to inform the next generation of policy regulations, considering the full lifecycle emissions of hydrogen production systems.”*

Lastly, as explained in Response to Comment #8, grid emissions derive from the European Commission dataset:

Lines 939-945: *“JRC-COM-NEEFE dataset [...], includes upstream emissions while excluding emissions from manufacturing technologies, thereby attributing 0 CO_{2e} per kWh when electricity is generated from renewable sources.”*

Comment #28

You are discarding SOEC technology because is not mature yet, but if your time horizon goes to 2050 why not considering it? (it should be mature enough by then)

Response to Comment #28

Thanks for the insightful comment. This is a very important point.

In the revision we have conducted additional analysis to include the impact of electrolyzer technology selection. We believe that with the insight from the reviewer, we developed more interesting findings. See the revised *Main*, subsection *Robustness analysis*:

Lines 429-459: *“For this study, Alkaline (ALK) electrolyzers, were selected given their maturity and widespread use. However, other alternative electrolyzer technologies may become commercially viable in the future (see Methods, subsection “Electrolyzers”). To account for this, additional analyses have assessed two additional electrolyzers technologies. The first model represents a low-cost but less efficient electrolyzer (i.e., membraneless (ML): 54% cheaper but also 29% less efficient than ALK^{50,51}), while the second represents a more expensive yet highly efficient electrolyzer (i.e., solid oxide electrolyzer (SOE): 280% more expensive and 17% more efficient than ALK⁵²).*

Despite their superior efficiency, the significantly higher costs associated with SOE electrolyzers result in a higher LCOH across all regions and emission cap scenarios, with increases ranging from 7% to 43% (SI – Table viii). The lowest increase, 7%, is observed in low-capacity regions, where higher efficiency can help to reduce the larger installed capacity of renewable infrastructure as well as reliance on grid import. Conversely, in high-capacity regions and in regions dominated by wind energy, the deployment of SOE simply leads to a higher LCOH. This can be attributed to the fact that the high renewable potential results in smaller installation sizes. As such, the benefits of increased efficiency cannot offset the higher costs of the more efficient electrolyzers. Similarly, electrolyzers that are less efficient but also less expensive (ML), generally result in a higher LCOH. However, the pattern here is completely opposite to SOE. A less efficient electrolyzer implies a higher energy demand, which in turn necessitates larger renewable installations. This effect is particularly pronounced in low-capacity regions, where it results in a 28% cost increase. On the other hand, regions with high renewable capacity, either solar or wind, might experience a slight (1%) reduction in LCOH under the 0-cap scenario. This happens because in these regions, under the most stringent emission cap, the electrolyzer

capacities are exponentially larger than under the less stringent caps, allowing the lower electrolyzer costs to offset the impact of reduced efficiency.

To summarize, despite the current options either being too expensive or having low efficiency, in general, more efficient electrolyzers provide higher benefits in regions with low renewable energy availability. This is primarily because the enhanced efficiency can leverage the reduction in renewable installations needed. Conversely, cheaper electrolyzers can reduce the LCOH in renewable-rich regions, particularly under stringent emission caps as these regions with plentiful renewables can deploy more capacity at lower costs, thus allowing the lower equipment costs to offset the impact of lesser efficiency. Therefore, the choice of electrolyzer technology should be carefully matched to the local conditions, particularly the availability of renewable resources and the emission cap in place.”

In the revised manuscript, we bring attention to these findings for more in-depth analysis to elucidate the trade-off between cost and efficiency in electrolyzer selection. Nevertheless, it is important to acknowledge that regardless of the electrolyzer technology selected, the main conclusions of our study remain valid.

Comment #29

Is the method used to calculate electricity price still valid to estimate 2050 prices in an energy system that may look completely different by then and were historical data may be “outdated”/inadequate? Please discuss.

Response to Comment #29

We appreciate the comment. This is an important consideration.

We acknowledge that forecasting electricity prices for a time as distant as 2050 comes with inherent uncertainties due to possible changes in technology, policy, market dynamics, and energy mix.

To address the uncertainty of future electricity price, we defined optimistic and pessimistic scenarios to verify the consistency of our main conclusions under different electricity price assumptions. In addition, we also conducted robustness analysis to test the impact of electricity price on LCOH and evaluate the robustness of our results against changes in electricity prices. A dedicated discussion is included in **SI-Text x**.

With these efforts, our main conclusion of the emissions caps and critical factors can be reliable to support future studies.

Comment #30

Please explain how did you include the different components lifetime in your LCOH calculation. The method used for carbon emission accounting should be further detailed and compared with the state-of-the-art methods used, for example, in the LCA field.

Response to Comment #30

Thanks for pointing this out. In the revised manuscript we clarified that we take component lifetimes from the literature and assume components are replaced at the end of their respective lifetimes at projected future component prices. Please refer to the revised *Methods*, subsection *Technology input data* and **SI – Table xii** that highlights the time for replacement.

Regarding emission accounting, we have improved our definition of the boundaries and scopes for emissions related to electrolytic hydrogen and clarified that our primary data sources are derived from a synthesis of life cycle assessment (LCA) studies.

Moreover, our methodology is in line with the one proposed by most low-carbon hydrogen certification systems (**SI - Table ii**) as articulated in Response to Comment #8 (**Lines 156 – 159**). Nonetheless, we also included in the revised paper extensive discussions about the implications of including additional

carbon emissions, such as those associated with the manufacturing of technology and components (further details in Response to Comment #27).

We hope that our revision can help readers better understand our modeling framework, emission definition and data sources. We will make these datasets publicly open once the study is published to help others develop further studies.

Comment #31

Remember to also point to the limitations of the methods used and their implication on the results you obtain.

Response to Comment #31

We concur with the suggestion. We do believe that there are multiple aspects for method improvements. We included a new paragraph in the end of the *Discussion* section about limitations and the implications on the results.

Lines 600-614: *“The study can be further improved in multiple aspects to formulate rigorous emission cap regulations for sustainable investment in low-carbon electrolytic hydrogen for ammonia production. First, an analysis of how varying timeframes for measuring CO₂e emissions, from hourly to yearly, influence the LCOH is crucial. Hourly emission caps may be challenging in industrial processes that require continuous output and a dedicated investigation is needed. Second, investigating the potential of flexible ammonia production is essential. This approach could more efficiently harness intermittent renewable energy, influencing both LCOH and the determination of cost-effective emission caps. Third, the performance of the optimization model (See Methods, subsection “Optimization model”) can be enhanced by incorporating robust optimization techniques to better handle the uncertainties associated with renewable energy variability. In this study, robustness analyses were instead performed to verify the consistence of the main findings (See Methods, subsection “Robustness analysis”). Finally, the study should be extended to other hard-to-abate sectors with heavy hydrogen usage. The feasible cost-effect emission caps for these hard-to-abate can be different from the findings for ammonia production in Europe, and these differences should be thoroughly investigated to formulate sector-based emission regulations.”*

We hope that this revision about strategies to further reduce LCOH can serve as a catalyst for future studies.

Other

Comment #32

Check your reference list, some references are appearing twice.

Response to Comment #32

Thanks for noticing it. We carefully reviewed and corrected the reference list in the updated manuscript.

Comment #33

Be sure to write CO₂e in the units when you talk about CO₂ equivalent.

Response to Comment #33

In the revised manuscript, we have ensured the correct use of the term CO₂e and CO₂. We also carefully checked the references we cited to verify whether the unit is CO₂ only or CO₂e.

Thank you for bringing this to our attention.

Reviewer #3:

Comment #1

The paper's emphasis on providing insights for industry and policymakers through a comprehensive analysis of the European ammonia sector and the impact of emission caps adds value and significance to the research. The contrast with previous studies strengthens the originality and significance of the research.

Response to Comment #1

We express our gratitude to the reviewer for the positive and constructive feedback. These comments have provided valuable guidance for improving our manuscript, and we hope to meet the reviewer's expectations through the responses and corresponding modifications.

Indeed, our primary goal was to provide thorough insights into the European ammonia sector and assess the impact of emission caps, providing insights for industry stakeholders and policymakers. We are pleased that our efforts in contributing valuable research to this domain have been recognized.

Comment #2

This paper would benefit from providing more methodological details, discussing uncertainties, and offering a more comprehensive analysis of the findings to enhance the clarity and robustness of the study. It would be helpful to explain the scientific or policy justifications, relevant literature, or regulatory frameworks that influenced the selection of the emission caps under consideration.

Response to Comment #2

Thank you for the reviewer's constructive comment. All the comments from the reviewers are relevant and through a meticulous revision process guided by these valuable insights, we hope to enhance the overall quality of the study.

In the revised manuscript we focused on providing more methodological details. The main upgrades follows:

1. Inclusion of 2 new figures, showing the boundaries of the analysis (see Fig. 1) and the methodological framework (see Fig. 7);
2. Detailed explanation of retrofitting costs (see *Methods*, subsection *Retrofitting costs*, Lines 891-906);
3. Detailed description of the optimization model, including explanatory equations for the objective function (see eq. 1 and eq. 2), emission constraints (see eq. 3 and eq. 4), and energy and mass balance constraints (see SI – Text v);
4. Elaborate in details how representative regions are selected (see *Methods*, subsection *Robustness analysis*; Fig. 9);
5. Include limitations of the study (see *Discussion*, Lines 535-614).

We expanded the uncertainty analysis, providing more detailed explanation of the results of the robustness analysis.

Lines 415-428: *“Among the input parameters evaluated, the price of grid electricity is found to be the most impactful under less stringent emission caps, with its significance diminishing under more stringent caps. Under the no-cap scenario, high electricity prices result in an average 21% increase in LCOH (SI – Table vii). This increase is primarily driven by a 39% rise in low-capacity regions, which cannot leverage inexpensive renewable generation, compared to an 8% increase in high-capacity regions. Conversely, low electricity prices result in an average reduction in LCOH of 73%, a trend that is more homogeneous across all regions. Low electricity prices would lead to predominantly using grid electricity, eliminating the need for additional renewable installations when optimizing costs without any emission constraints.*”

The second most influential parameter in determining the LCOH is the cost and performance of the Alkaline (ALK) electrolyzer, with it causing a variation in LCOH of approximately $\pm 20\%$ (SI – Table vii), depending on the specific case under consideration. Given the substantial impact of electrolyzer costs and performance on the LCOH, further analyses were conducted to delve deeper into this relationship and potentially identify strategies for optimizing these parameters to enhance the economic viability of hydrogen production.”

We also included further analysis regarding the role of technology development, such as electrolyzer selection:

Main, subsection Robustness analysis, Lines 452-457: “To summarize, despite the current options either being too expensive or having low efficiency, in general, more efficient electrolyzers provide higher benefits in regions with low renewable energy availability. This is primarily because the enhanced efficiency can leverage the reduction in renewable installations needed. Conversely, cheaper electrolyzers can reduce the LCOH in renewable-rich regions, particularly under stringent emission caps as these regions with plentiful renewables can deploy more capacity at lower costs, thus allowing the lower equipment costs to offset the impact of lesser efficiency.”

In general, throughout the entire revision, we tried our best to explain the mechanisms driving the results presented. Hereafter an example:

Main, subsection The largest installations are required to meet the 0-cap, Lines 261-267: “The installed capacity of wind turbines is estimated to approximately triple from the 0.1-cap to the 0-cap scenario, while the installed capacity of electrolyzers quadruples. PV installations, however, grow more steadily from the no-cap to the 0.1-cap scenarios. On average, PV systems constitute 57% of the renewable power capacity for semi-islanded configurations, whereas this share drops to 46% for islanded (off-grid) systems. This result stems from the difference between solar and wind resource profiles. Under the 0-cap, the consistent availability of wind energy leads to an increased use of wind installations, reducing the reliance on solar energy.”

In the revised manuscript, we explained that our model boundaries and emission caps selection are based on existing and proposed regulatory frameworks, including the recent European Union's 'Fit for 55' package. *“The European Union's 'Fit for 55' package, particularly the Renewable Energy Directive II delegated act²⁹ sets out requirements for additionality, temporal and geographic correlation concerning the electricity used in hydrogen and fuels derived from it.”* – Lines 69-78, 149-150, and SI – Text i. In doing so, the study mirrors existing policy guidelines to generate insights that can inform ongoing policy efforts.

Alongside with the 1, 0.5, 0.1, and 0-caps already included in previous version, we also included an emission cap in accordance with the latest *“EU taxonomy in Europe, which stands at less than 3 kg CO_{2e}/kg H₂²⁸”* – Lines 69-71. *“Those 5 caps cover proposed certifications and regulations for hydrogen. The aim is to probe the implications of increasingly stringent emissions reduction targets on the LCOH on the life-span well-to-gate carbon dioxide-equivalent (CO_{2e}) content of hydrogen”* – Lines 156-163, and SI - Table ii.

In the European context, explicitly testing caps higher than 3 kg CO_{2e}/kg H₂ does not provide any relevant insight. *“In fact, 30/38 plants emit less than 3 kg CO_{2e}/kg H₂ with electrolytic hydrogen production, even without the imposition of emission caps. [...] Replacing SMR with water electrolysis for all the European ammonia plants results in an average emission reduction of about 85%, even without enforcing emissions caps attributable to the average well-to-gate carbon content of hydrogen being 1.85 kg CO_{2e}/kg H₂ (Fig. 4a and SI – Table iv)”* – Lines 311-318. This reduction is due to the lower levelized cost of renewable electricity (LCOE) from newly installed infrastructure when compared to conventional grid electricity. Hence, *“any caps higher than 3 kg CO_{2e}/kg H₂ are effectively encompassed by the 'no-cap' scenario.”* – Lines 162-163.

Nevertheless, it is acknowledged that applying this methodology *“in regions outside EU where the energy mix is more carbon-intensive and less expensive, higher emission caps might be more appropriate to study, as the impact on costs and emissions could be more pronounced and thus more relevant.”* – **Lines 311-315.**

Comment #3

The authors cite two references to support hydrogen storage in favor of battery technology for the 0-cap. A more comprehensive examination of the literature would be helpful to confirm consensus amongst previous studies.

Response to Comment #3

We appreciate the reviewer’s comment. In response, we have expanded our literature review, including a more comprehensive set of references to clarify the choice between hydrogen and battery storage. These additions have been incorporated into our revised manuscript.

We have updated our study with new data showing lower compressor and tank costs, alongside a slight increase in Li-ion battery costs. Hence, in the updated results no ammonia plant opt for batteries. This aligns with several studies demonstrating the cost-effectiveness of hydrogen storage compared to battery storage.

Lines 272-276: *“Compressed hydrogen is favored over li-ion batteries. Although a significant cost reduction in utility-scale batteries is forecasted in the coming year⁴⁶, hydrogen storage is estimated to be cheaper for large-scale applications, as also previously shown by other studies⁴⁷⁻⁴⁹. One notable observation is that the greater availability of wind energy throughout the day reduces the need for storage in wind-rich regions compared to regions primarily reliant on solar energy.”*

It is important to note that we observed numerous studies comparing hydrogen and battery technology for **grid storage**. As their scope diverges from our study, focus on hydrogen for ammonia production, we deliberately excluded these references.

We trust that this comprehensive review addresses the reviewer’s concerns and strengthens the analysis of the choice between hydrogen and battery storage, confirming our findings with additional references.

Comment #4

The spiral bar chart is visually appealing although challenging to interpret at first.

Response to Comment #4

We updated original Fig. 3 (now **Fig. 4**). We increase readability by showing only the most relevant emission caps (no-cap, 1-cap, 0.1-cap, and 0-cap), increase font size, and add further explanation in the caption.

Comment #5

The exclusion of the carbon intensity of technology manufacturing and installation from the analysis overlooks a significant portion of overall emissions associated with the electrolyzer’s lifecycle. The authors should consider this aspect to gain a more comprehensive understanding of the emissions impact.

Response to Comment #5

This is a very important point. Thanks for bringing it up.

In our analysis, we excluded Scope 3 embedded technology emissions to align with existing and proposed regulatory frameworks (see **Lines 168 – 174**). However, we acknowledge the importance of

discussing the weight of such emissions and therefore we included extensive analysis and discussion in our revised manuscript.

SI - Fig. vii | Average lifecycle emissions (including scope 1, scope 2, partial scope 3 upstream emissions and technology manufacturing) of electrolytic hydrogen in Europe.

Few take-home messages from SI - Fig. vii follows:

1. The inclusion of Scope 3 embedded technology emissions raises the average lifecycle CO₂e content of electrolytic hydrogen to 2.2 kg CO₂e/kg H₂ under the 1-cap and to 3.0 kg CO₂e/kg H₂ under the 0-cap scenario (see *Discussion, Lines 565 – 577*).
2. Including emissions from the manufacturing of renewable installations increase on average 1.19 kg CO₂e/kg H₂ under the no-cap scenario (56% from WT and 44% from PV) and rise to 2.76 kg CO₂e/kg H₂ under the 0-cap scenario when larger overcapacities are required (67% for WT and 33% for PV). (see *Fig. vii*).
3. Large installations of PV and wind turbines lead to higher lifecycle emissions under the 0-cap condition, despite the absence of grid electricity usage. This is further accentuated in areas with limited renewable energy capacity. (see *Fig. vii*)
4. Notwithstanding this addition, there would still be an approximate 75% reduction in emissions compared to SMR technology, even without factoring in the manufacturing emissions for SMR. (see *Fig. vii*)

Plants situated in regions with low renewable capacity and stringent emission caps would necessitate larger installations if they operate off-grid, which in turn would increase the Scope 3 embedded technology carbon footprint. An example is SK02 under the 0-cap, which requires triple the average installed capacity compared to other European plants subject to the same emissions limitations, with 11.3 GW from solar and 1.4 GW from wind power. Even in this extreme case, the lifecycle emissions would be 6.8 kg CO₂e/kg H₂, marking a 40% reduction when compared to SMR:

Lines 565-577: “It is crucial to underscore that a 0-cap scenario not only implies higher costs and increased land usage for renewables but can also potentially lead to higher overall lifecycle emissions compared to scenarios with less stringent caps. Present policies do not account for ‘Scope 3 embedded technology’ emissions from component manufacturing, and this study therefore also omits these emissions. However, under the 0-cap scenario, where renewable installations are significantly larger, the total carbon content of hydrogen - when accounting for emissions from component manufacturing (including Scope 3) - is 33% higher compared to the lifecycle emissions under the 1-cap scenario, given global average emissions components as shown in SI – Fig. vii (details about calculation of Scope 3

embedded emissions in SI, text iv). As we look to the future, the cleaner the grid of EU becomes, the more significant the relative contribution of component manufacturing to overall emissions will become. This suggests that future research could usefully adapt this model to inform the next generation of policy regulations, considering the full lifecycle emissions of hydrogen production systems.”

We hope that this extensive discussion may provide a more comprehensive understanding of the emission impact of technology.

Comment #6

In addition to capacity factors, other factors such as transmission losses, curtailment rates, and grid congestion should be considered in a robustness analysis to provide a more complete understanding of the system's resilience.

Response to Comment #6

We appreciate the suggestion: In the revised manuscript we explained that transmission losses range from 7% to 10%, curtailment rates range from 14% (no-cap) to 39% (0-cap), and that grid congestions are unlikely to occur due to the small fraction of energy imported from the grid.

Two transmission losses are considered: (i) from grid electricity and (ii) from renewable installations:

Methods, subsection Grid connection upgrades, Lines 740 – 753:

“Existing fossil-based ammonia plants are already grid-connected, albeit with a few MW of power capacity. However, when retrofitting these plants—replacing SMR hydrogen with electrolytic hydrogen production—the peak capacity demand from the grid will increase. This means that when a large amount of energy is required, the grid should be capable of supplying it. As a result, grid connection upgrades to accommodate this increase in peak capacity demand are included in the LCOH calculation. This includes the installation of new high voltage alternating current (HVAC) wires and a transformer to step down from high voltage to low voltage at the plant side (unit cost in SI - table xii). [...] Given the typical location of existing plants in industrial areas, a proximity to the grid connection point (within 5 km) is assumed, and line losses are negligible due to the short distance. Hence, only transformer losses (1% HV-LV) and rectifier losses (5% at plant side) are included.”

Methods, subsection “Renewable power generation”, Lines 831 – 854:

“The LCOH also accounts for the losses in the transmission of electricity from renewable power generation to the electrolysis plant. [...] The construction of new, separate transmission lines is assumed. [...] Given the assumption that the renewable installations are located within short distances (<10km), HVAC is assumed to transmit electricity from PV and WT to the plant. Wind Turbines typically generate LVAC, which must be stepped up to HV using transformers, transferred to the plant, stepped down, and then converted into (LVDC) using rectifiers. Conversely, PV systems produce LVDC, requiring an additional step to convert LVDC to LVAC before stepping up and integrating into the dedicated HVAC line. Costs associated with inverters are already included in the IRENA dataset. [...] Total transmission losses from the WT to the electrolysis hydrogen system are around 7% (2% transformers, 5% rectifier), and for PV systems, around 10% (3% inverter, 2% transformers, 5% rectifier).”

Regarding curtailment rates, we included the following paragraph and figure **SI – Fig. iv**:

Lines 296-302: *“Increasingly stringent emission caps not only affect the optimal design and costs of the EHPS but also affect the renewables curtailment rate. The curtailment rate increases exponentially with more stringent emission targets due to the larger over-capacities installed for a limited number of low-resource hours per year. For instance, the curtailment rate rises from 14% with no emission cap, to 22% with a 0.1-cap, and to 39% under a 0-cap when the plant operates off-grid (SI – Fig. iv). An*

increase in curtailment rate is typically observed during peak renewable energy production periods when the plant cannot handle the excess energy.”

Lastly, in *Methods*, subsection *Feasibility analysis*, we compared the imported electricity from grid and the total regional electricity consumption to illustrate the impact on existing grid. We cited a reference that uses the same ratio to study the impact of increased demand on existing grid.

Lines 474-486: *“Results show that, under the 1-cap, the EHPS mean grid electricity demand is 491 GWh per year (min 26 GWh, max 1,488 GWh across Europe). The shift to electrolytic hydrogen production, compared to the SMR process, does increase the demand for electricity (on average, 56 MWh are imported from the grid each hour). While this increased electricity consumption necessitates infrastructure upgrades on the plant side, as previously discussed, the plant's energy demand is relatively small, representing, on average, 6.3% of the current energy demand at the regional level and 0.8% at a national level. This increase is unlikely to have a significant impact on regional consumption patterns, causing grid congestions, or to influence price dynamics notably³⁸. Nevertheless, while the overall energy demand of the grid may not be significantly affected, the peak power demand during operation can be substantial³⁸ (see *Methods*, “*Feasibility analysis*”). There is a need to investigate how regional or local grids will adapt to the increased demand of EHPS. This includes the necessary expansion of power generation capacity and the adjustment of pricing policies to ensure that hydrogen production via electrolysis remains cost-effective.”*

The feasibility analysis serves as a basis for future studies to investigate how regional or local grid should prepare for the increased demand from electrolytic hydrogen for ammonia production including the expansion of power generation, and price strategy. This follow-up study can further help grid operator and ammonia plant better prepare for the emission cap regulation. Such studies should be done at a sub-regional resolution with more granular grid data. We hope that our revised work can open the way to future investigations to provide cost-effective low-carbon hydrogen.

Comment #7

The use of the EMHIRES dataset for calculating capacity factors is reasonable and provides consistency in the analysis. However, it is important to acknowledge that the dataset may not fully capture region-specific factors such as terrain, shading, regional weather patterns that affect wind and solar energy generation.

Response to Comment #7

We appreciate your comment and concur with your suggestion. In response we have included the following paragraph to acknowledge the limitations of using capacity factors in *Methods*, subsection *Wind and solar capacity factors*:

Lines 963-966: *“While the dataset might not fully capture certain geographic-specific factors like terrain conformation, natural obstacles and shading, it offers a good representation of renewable capacity and weather variability across Europe. This underscores the role of local renewable resources in shaping the economic and technical viability of electrolytic hydrogen production.”*

While a more granular, region-specific analysis could potentially yield further insights and accuracy, it is beyond the scope of our study, given its continental scale. Future studies could aim at improving the precision of capacity factors examining high-resolution geospatial data to understand which areas are feasible for such installations and to analyze potential natural constraints such as terrain and shading. – **Lines 963-966.**

Comment #8

It may be beneficial to further elaborate on grid integration and the availability of renewable resources, specifically in relation to the electrolyzer technology selection and the feasibility of low-carbon

hydrogen production. Explore how these factors were considered and their potential impact on the analysis.

Response to Comment #8

Thanks for bringing this up. We appreciate the insightful comment and agree that including a discussion regarding grid integration and electrolyzer technology selection will improve our work.

Grid integration has been extensively discussed in the revised manuscript. In *Methods*, subsection *Feasibility analysis*, we explained that the most of the EHPS energy demand, primarily for powering electrolyzers, is met by newly installed renewable installations. However, during periods of low renewable capacity, the plant may occasionally depend on grid imports to maintain continuous operation. European plants would import an average of only 15-30% of the electricity demand (this percentage varies depending on the stringency of the emission cap). This translates in an average of 56 MWh imported from the grid each hour.

*Lines 477-486: “While this increased electricity consumption necessitates infrastructure upgrades on the plant side, the plant’s energy demand is relatively small, representing, on average, 6.3% of the current energy demand at the regional level and 0.8% at a national level. This increase is unlikely to have a significant impact on regional consumption patterns, causing grid congestions, or to influence price dynamics notably³⁷. Nevertheless, while the overall energy demand of the grid may not be significantly affected, the peak power demand during operation can be substantial³⁸ (see *Methods*, “Feasibility analysis”). There is a need to investigate how regional or local grids will adapt to the increased demand of EHPS. This includes the necessary expansion of power generation capacity and the adjustment of pricing policies to ensure that hydrogen production via electrolysis remains cost-effective.”*

Regarding electrolyzer technology selection, as outlined in Response to Reviewer 1, Comment #28, this study initially assumes the use of an Alkaline (ALK) electrolyzer. However, we have expanded the robustness analysis to examine alternative electrolyzer technologies: a cheaper but less efficient membraneless (ML) and an expensive but high-efficiency solid oxide SOE. Our results follow:

Lines 452-457: “To summarize, despite the current options either being too expensive or having low efficiency, in general, more efficient electrolyzers provide higher benefits in regions with low renewable energy availability. This is primarily because the enhanced efficiency can leverage the reduction in renewable installations needed. Conversely, cheaper electrolyzers can reduce the LCOH in renewable-rich regions, particularly under stringent emission caps as these regions with plentiful renewables can deploy more capacity at lower costs, thus allowing the lower equipment costs to offset the impact of lesser efficiency.”

We bring attention to these findings for more in-depth analysis to elucidate the trade-off between cost and efficiency in electrolyzer selection. Nevertheless, it is important to note that regardless of the electrolyzer technology selected, the main conclusions of our study remain valid.

We recognize that electrolyzer technology selection plays a significant and expansive role in grid integration, renewable energy demand, and the feasibility and cost of ammonia production. We hope our revised study offers valuable insights for further research, serving as a starting point for a more comprehensive analysis in a dedicated separate work.

Comment #9

The authors should explicitly state the assumptions made in the GBM and Monte Carlo simulations and discuss their potential impact on the forecasted results.

Response to Comment #9

Thank you for the reviewer's comment, we agree this was not clear. We extended the description of the GBM and Monte Carlo simulations in **SI – Text x**:

“The Geometric Brownian Motion (GBM) model assumes a normal distribution of logarithmic returns and constant volatility, which simplifies the complexities of real-world financial markets where actual price returns may not follow a normal distribution and volatility can be dynamic.”

To address the uncertainty of future electricity price, we defined optimistic and pessimistic scenarios to verify the consistency of our main conclusions under different electricity price assumptions. In addition, we also conducted robustness analysis to test the impact of electricity price on LCOH and evaluate the robustness of our results against changes in electricity prices.

SI – Text x: *“However, the robustness analysis and inclusion of pessimistic and optimistic scenario in the study compensates for these simplifications by evaluating a wide array of electricity price data across different regions.”*

Results show that our main conclusions are robust and can support future studies.

SI – Text x: *“This approach has confirmed that the key conclusions of the study are consistent, even when taking into account variations in electricity costs and their potential volatility.”*

This detailed exploration, now included in the revised manuscript, offers a more comprehensive understanding of the role of electricity prices, reinforcing the validity of our model despite its simplifications.

Comment #10

While constructing synthetic regions based on capacity factors is reasonable, it does not fully capture the real-world variability and uncertainties associated with renewable energy generation, including seasonal variations, weather patterns, and grid integration constraints.

Response to Comment #10

Thanks for the reviewer's insightful comment. In this revision, we identified and used "representative" regions, and we discuss how current plant locations align with these categories.

To identify representative regions, we calculated the mean annual capacity factor for each region and year available in the EMHIREs datasets for solar and wind energy. Regions were clustered into five categories based on their solar and wind energy capacity factors.

After classifying all regions into five categories, we identified a representative region for each group, specifically selecting the region that exhibits the most extreme case within its category.

For detailed information, see *Methods*, subsection *Robustness analysis - Lines 979-1021, and Fig. 9*.

Fig. 9 | European regions (NUTS-2 level) clustered based on their mean annual capacity factor along with the 5 selected representative regions and years. Each dot represents a specific region and year means annual capacity factor in percentile. a, percentile; b, capacity factor in percentage.

This approach allows us to capture the variability and extremities of climate conditions while also reducing the computational intensity of testing all regions. Moreover, because these representative regions are based on actual historical hourly data for solar and wind, they accurately capture real-world variability and uncertainty in power generation.

This methodology simplifies determining which European regions fall into these five categories. This not only enhances the relevance of our results but also improves the practicality of our findings, as it

becomes straightforward to classify each ammonia production plant into one of these categories (see SI – Fig. xv). This has been exhaustively detailed in the revised manuscript.

SI - Fig. i | European regions clustered in based on their solar and wind capacity factor. Pink dots indicate the location of existing ammonia plants.

We acknowledge that our chosen regions may not encompass all variations in renewable energy potential, yet they suffice for a robust regional analysis. We concur with the reviewer’s consideration, and therefore included a new section in the revised *Discussion*:

Lines 600-614: “*The study can be further improved in multiple aspects to formulate rigorous emission cap regulations for sustainable investment in low-carbon electrolytic hydrogen for ammonia production. [...] Third, the performance of our optimization model (See Methods, subsection “Optimization model”) can be enhanced by incorporating robust optimization techniques to better handle the uncertainties associated with renewable energy variability. In this study, we performed robustness analysis instead to verify the consistence of our findings (See Methods, subsection “Robustness analysis”).*”

REVIEWER COMMENTS

Reviewer #1 (Remarks to the Author):

General comment:

Good article supplemented by an extensive and useful supplementary material. The article seems to be soon ready for publication with a few minor adjustments. Maybe you can consider adding a Github link to your code to facilitate other researchers re-use/validate your work?

However, I still have one main concern: all your conclusions for 0-cap seems to be based on ammonia plants that do not run flexibly while most of the literature agree that flexible ammonia plants are a key point to reduce the costs in off-grid systems and that it is technically feasible, some industrials also argue in that sense. Please explain explicitly why flexible ammonia plant option has been discarded also in the “main” (not only the method) and make sure to emphasize that you took this assumption when you present your results in the main, abstract, etc... . In any case, I would recommend to implement the “flexible ammonia plant” feature in your model (even if the minimal load is “only” 50%) and see how it influences your results in the 0-emission case to avoid potentially misleading conclusions. You mention it in the discussion but I don’t think that this is enough given the potential importance of this assumption on your results and conclusions. Ideally, you could add a sensitivity analysis on this parameter and explain some of the results in the main.

Abstract:

You talk about 85% emission reduction using grid-backup. Maybe add a small clarification explaining that is a grid back-up relatively “clean” or covering a small share of the electricity consumption (?).

You say: “Conversely, a 100% emissions 16 reduction target dramatically increases costs (mean LCOH: 6.3 EUR/kg H₂) and land area for 17 renewables installations, likely hindering the transition to electrolytic hydrogen in most regions.”

This may be true but only for non-flexible ammonia plant and grid electricity price “not too high” (?): add a precision here, to avoid misleading conclusions about 100% green off-grid systems.

Say something about the year used to derive the costs you present in the abstract to avoid any confusion (is it for 2020, 2030, 2050?). Did you use 2019 grid electricity prices? (or 2021? Or Future? Or both?). A few words should be enough to give that information.

Introduction:

Seems fine

Main:

When you say that electrolytic hydrogen is competitive except with zero caps (for example) make sure to remind the reader under which these conclusions are valid (i.e. constant ammonia production with non-flexible operation, expensive on-site H₂ storage –opposed to cheap large scale H₂ cavern storage-, ...). Because off-grid solutions are probably not that expensive if it is assumed that ammonia plant can run flexibly.

Please be a bit more specific on the emissions from the grid: are you using marginal or average emissions? And how do you justify that choice? Because using one accounting method or another may have a significant impact on the results. It would be worth adding that point in the discussion.

I already made that comment in the first version but I think that your changes in the text could be improved a little more.

What is the total share of electricity provided by the grid? If the grid consumption is high due to the additional load of multiple electrolyzers, with which production units this extra electricity demand will be covered? Do we need to build more i.e. gas turbines (stable load to ensure continuous ammonia production)? Or the existing grid is sufficient? You discuss this point in this paragraph: “The transition to electrolytic hydrogen is challenging for the 0-cap” but you would gain to explain it a bit more clearly (and relate this topic with marginal/average grid emissions). Maybe you could also change the title of this subsection to something related to “additional grid capacity needed”.

As a general comment, I think that you can expand a bit more the comparison of your results with what has been found in the literature.

Discussion:

You say: “It is crucial to underscore that a 0-cap scenario not only implies higher costs and increased land usage for renewables but can also potentially lead to higher 566 overall lifecycle emissions compared to scenarios with less stringent caps.” Do you also include scope 3 emissions when calculating the grid carbon intensity? I may have missed it but I did not find that information.

Reviewer #3 (Remarks to the Author):

Upon reviewing the revised manuscript, you have addressed the previous comments with diligence and attention to detail. Your efforts to refine the manuscript have not only enhanced its clarity but also enriched the overall quality of the work. The revisions have satisfactorily resolved my concerns raised, and it is evident that a considerable amount of thought and work went into making these changes. I appreciate your responsiveness to the review process and commend you on your dedication to improving the manuscript.

Reviewer #3 (Remarks on code availability):

Code note available. Webpage 404, page not found error.

RESPONSE TO REVIEWERS' COMMENTS (SECOND ROUND OF REVIEW)

Reviewer #1:

General comments

Comment #1

Good article supplemented by an extensive and useful supplementary material. The article seems to be soon ready for publication with a few minor adjustments.

Response to Comment #1

We are grateful for your feedback. Our team is pleased that the enhancements made to the manuscript after the initial round of revisions have been acknowledged.

We extend our thanks to all the reviewers for their insightful comments and questions, which have undoubtedly elevated the value, applicability, and significance of our work.

Comment #2

Maybe you can consider adding a GitHub link to your code to facilitate other researchers re-use/validate your work?

Response to Comment #2

Thank you for your suggestion. We agree that providing easy access to our research's data and code is crucial for replication and further development by other researchers. In line with your recommendation, we have now included a publicly accessible GitHub link in the main manuscript (<https://github.com/hkust-suscity/Electrolytic-ammonia-production-in-Europe>). This link directs readers to a repository containing all the necessary code to replicate our main analysis, including the sensitivity analysis, as well as all the input data required to reproduce our results. We believe this addition will significantly enhance the usability and transparency of our modeling framework.

Comment #3

However, I still have one main concern: all your conclusions for 0-cap seems to be based on ammonia plants that do not run flexibly while most of the literature agree that flexible ammonia plants are a key point to reduce the costs in off-grid systems and that it is technically feasible, some industrials also argue in that sense. Please explain explicitly why flexible ammonia plant option has been discarded also in the “main” (not only the method) and make sure to emphasize that you took this assumption when you present your results in the main, abstract, etc.... In any case, I would recommend implementing the “flexible ammonia plant” feature in your model (even if the minimal load is “only” 50%) and see how it influences your results in the 0-emission case to avoid potentially misleading conclusions. You mention it in the discussion, but I don't think that this is enough given the potential importance of this assumption on your results and conclusions. Ideally, you could add a sensitivity analysis on this parameter and explain some of the results in the main.

Response to Comment #3

We appreciate the insightful feedback provided. We concur to clarify even more the assumption of continuous ammonia production throughout the manuscript.

First, we modified the abstract: “[...] These emissions can be mitigated by producing hydrogen via water electrolysis powered by dedicated renewable energy sources with grid backup **to maintain continuous plant operations.**” **Lines 7-9.**

Second, we included the following paragraph at the beginning of the Main section: “This study assumes that the existing European ammonia plants will be retrofitted by replacing the SMR production system with an electrolytic hydrogen production system (hereafter EHPS). **The other subsystems of the ammonia plant, including the air separator unit (ASU), the ammonia synthesis loop, and the cryogenic storage for ammonia, will be maintained as they are in the existing facility. Consequently, the EHPS is designed to deliver a continuous hydrogen supply thanks to on-site hydrogen storage in pressurized tanks and grid backup, ensuring uninterrupted operation of the ammonia plant, which requires steady-state conditions** (detailed information Methods, subsection “Ammonia production process and EHPS”).” **Lines 129-136.**

We also modified the first results’ subtitle, emphasizing that “Electrolytic hydrogen can be cost-competitive with SMR except for the 0-cap **when maintaining continuous production.**” **Lines 209-210**; as well as the discussion section: “However, more stringent regulations like the 0-cap lead to substantial cost increases **when maintaining continuous production without any grid backup**, rendering the subsidies insufficient to offset the cost hike.” **Lines 633-635.**

Furthermore, we acknowledge the importance of a more detailed discussion on the potential impacts of flexible ammonia production on our findings.

We added a new section in the Main: “Lastly, the feasibility of flexible plant operation was explored. As detailed in Methods, subsection “Ammonia production process and EHPS”, ammonia plants typically operate at full capacity to satisfy the steady-state conditions required by the Haber-Bosch process. It is therefore assumed a continuous hydrogen supply to the synthesis loop thanks to hydrogen storage and grid backup to prevent operational disruptions. Recent industry efforts, however, have been directed towards investigating the potential for more adaptable plant operations, specifically aligning ammonia production with the variable output of intermittent renewable energy sources. The extent of operational flexibility is directly correlated with the capability to adjust to fluctuating energy inputs. This flexibility is limited by the least flexible component within the system (i.e., ASU and ammonia synloop), suggesting that constraints on any single technology’s operations can limit the flexibility of the entire ammonia plant.

Technological innovations are making ammonia plants more adaptable to variable power inputs. Electric heaters, variable load compressors, and better catalysts for ammonia synthesis allow for quicker adjustments to power input changes, more manageable load variations, and improved operational ramp-up and ramp-down. Future electrolytic ammonia plants could operate more efficiently and with greater flexibility than today’s standard.

In light of these developments, the impact of partially relaxing the hourly output constraint of the EHPS was tested. Specifically, the plants were assumed to operate down to a 50% minimum load while maintaining the same total annual hydrogen production, as detailed in the Methods, subsection “Robustness Analysis”. Results indicate that even “partially flexible” ammonia production can lead to cost reductions, attributable to the downsizing of renewable energy installations, grid import, and hydrogen storage. The extent of cost reduction varies, ranging from 6% under the “no-cap” to 32% when operating off-grid (0-cap), and is especially pronounced in regions dependent on wind energy, with a reduction of up to 46%, while solar-dominated regions experience a smaller impact, with a 19% reduction (SI – Table x).

This result is significant as it demonstrates that increasing the flexibility of the plant can partially mitigate the substantial cost increments encountered under a 0-cap (off-grid) scenario. These preliminary findings underscore the importance of prioritizing plant flexibility in the design of next-generation plants as a

strategic approach to decrease the production costs and land requirements of off-grid electrolytic ammonia plants.” Lines 499-528.

These results are supported by an additional table in SI:

Representative region	Emission-cap			
	No-cap	1-cap	0.1-cap	0-cap
High-capacity	-5%	-5%	-9%	-19%
Low-capacity	5%	-4%	-14%	-32%
Median-capacity	-6%	-6%	-11%	-46%
Solar-dominated	-3%	-3%	-13%	-19%
Wind-dominated	-16%	-16%	-22%	-46%
Average	-5%	-7%	-14%	-32%

SI - Table i | Impact of plant flexibility ($\delta_{H_2}D_{H_2,t}$) on LCOH. LCOH variation in % compared to the reference case (continuous ammonia production).

These results were also included in the Discussion section: “To reduce costs, land requirements, and lifecycle emissions associated with hydrogen production at off-grid plants, it is essential to prioritize increasing plant flexibility. Enhanced flexibility allows plants to align production more closely with the availability of renewable energy, thus reducing the need for extensive renewable generation installations (such as PV and wind turbines) and storage capacity. Although current plants offer limited operational flexibility, focused efforts from industry and targeted government R&D can stimulate advancements in this area. Enhanced plant flexibility could lead to improved competitiveness of islanded electrolytic plants and support the achievement of deep decarbonization goals in regions with favorable renewable resources, especially with high wind potential.” Lines 658-666; as well as highlighted in the abstract: “Increasing plant flexibility is an effective strategy to reduce costs, particularly in off-grid plants (mean reduction: 32%).” Lines 19-20.

Detailed explanations on how we included the flexibility analysis have been included in the revised Methods, subsection “Ammonia production process and EHPS”: “Nevertheless, the robustness analysis examines the implications of relaxing the constraint on hydrogen production output. Specifically, D_{H_2} , which was a fixed input parameter in the main analysis, is now treated as a decision variable optimized each hour $D_{H_2,t}$. However, the total annual hydrogen demand, remains consistent with the continuous operation scenario.

A minimum load parameter, denoted as δ_{H_2} , has been set at 50% to guarantee that the hydrogen output to the ammonia synthesis loop from the EHPS never falls below a safe minimum load (see eq. 5). It should be noted that while this assumption of minimum load is rather optimistic. Wang et al.²³, for example, consider a minimum load of 60% with ramp rates of 20% per hour.

$$\delta_{H_2}D_{H_2,t} \leq D_{H_2,t} \leq \frac{1}{\delta_{H_2}}D_{H_2,t}, \forall t \in \{0, \dots, T\} \quad \text{eq. 5}$$

Compared to the reference analysis with continuous production, the costs associated with the ASU, ammonia synthesis loop, and cryogenic storage tanks (Δ_f) must be included in the LCOH (eq. 6).

$$\text{LCOH} = \frac{z_{\text{cost}} + \Delta_f}{\sum_{l=1}^L \sum_{t=1}^T D_{\text{H}_2,t}} \quad \text{eq. 6}$$

Given that production may occur at a 50% load during certain hours, it is necessary to oversize the ASU and synthesis loop to accommodate higher loads during peak production times. The maximum hydrogen volume entering the synthesis loop is set to be smaller than $1/\delta_{\text{H}_2}$. This constraint ensures the installed capacities of the ASU and the synthesis loop do not exceed twice the size required for continuous operations. For large-scale ammonia plants, the doubling of installed capacity would mean that the ASU costs approximately 150 million EUR, and the capital expenditures (CAPEX) for the ammonia synthesis loop, inclusive of auxiliaries and balance of system, amount to 300 million EUR.

However, this analysis remains preliminary and somewhat simplistic. The installed capacity and operation of both subsystems should be modeled and optimized in greater detail, taking into account other cost factors such as control systems improvements. Furthermore, as per Wang's findings, variations from nominal load in the HB loop will likely result in reduced efficiency. Therefore, additional energy losses must be contemplated, alongside the need for more intermediate storage for nitrogen and other minor enhancements to prevent reactor poisoning.

Despite the simplicity of the model, such increases in energy demand and inefficiencies are expected to have a minimal impact on the LCOH and LCOA, as the majority of electricity usage and costs are predominantly incurred during the production of hydrogen.” **Lines 820-845.**

With this revision, we hope to have sufficiently addressed the reviewer's concerns and enriched our paper with a detailed analysis of flexible production.

Abstract

Comment #4

You talk about 85% emission reduction using grid-backup. Maybe add a small clarification explaining that is a grid back-up relatively “clean” or covering a small share of the electricity consumption (?).

Response to Comment #4

Thanks for the comment. In the revised abstract we clarified that the projected 85% reduction in emissions is an average figure across European plants. The range in emission reduction (36% to 100%) is due to the differences in grid price and projected carbon intensity across different regions: “[...] electrolytic hydrogen cuts emissions, **on average**, by 85% (**36%-100% based on grid price and carbon intensity from 2025 to 2050**), even without enforcing any emission limit.” **Lines 12-13.**

We also emphasized that the 1-cap exhibits a significantly narrower emission reduction range: “[...] an optimal lifespan average well-to-gate emission cap of 1 kg carbon dioxide equivalent (CO₂e)/kg H₂ achieves even deeper emissions reduction of 95% (**92%-100%**) [...]” **Lines 13-15.**

Comment #5

You say: “Conversely, a 100% emissions 16 reduction target dramatically increases costs (mean LCOH: 6.3 EUR/kg H₂) and land area for 17 renewables installations, likely hindering the transition to electrolytic hydrogen in most regions.” This may be true but only for non-flexible ammonia plant and grid electricity price “not too high” (?): add a precision here, to avoid misleading conclusions about 100% green off-grid systems.

Response to Comment #5

We appreciate your feedback highlighting the need for clarification in our discussion about 100% renewable-powered off-grid plants. We agree with your assessment and have revised addressing this concern.

We have clarified in the revised abstract that achieving a 0-cap (off-grid plant entirely powered by renewables) is particularly challenging in regions characterized by suboptimal renewable resources and constrained land availability: “Conversely, a 100% emissions reduction target dramatically increases costs (mean LCOH: 6.3 EUR/kg H₂) and land area for renewables installations, likely hindering the transition to electrolytic hydrogen in regions with poor renewables and limited land.” **Lines 17-19.**

To further elaborate on the strategies that could mitigate these challenges, we have included an additional statement emphasizing that increasing the flexibility of off-grid plants can be an effective cost-reduction strategy: “Increasing plant flexibility is an effective strategy to reduce costs, particularly in off-grid plants (mean reduction: 32%).” **Lines 19-20.**

With this modification, we hope to avoid any misleading conclusions about 100% renewable-based production.

Comment #6

Say something about the year used to derive the costs you present in the abstract to avoid any confusion (is it for 2020, 2030, 2050?). Did you use 2019 grid electricity prices? (Or 2021? Or Future? Or both?). A few words should be enough to give that information.

Response to Comment #6

In the revised abstract, we have clarified that our analysis is based on projections of grid prices and carbon intensity for the period from 2025 to 2050: “[...] based on grid price and carbon intensity from 2025 to 2050 [...]” **Lines 12-13.**

Additionally, we have explicitly written that the Levelized Cost of Hydrogen (LCOH) is computed based on the hydrogen production over this same timeframe: “[...] mean levelized cost of hydrogen produced from 2025 to 2050 or LCOH: 4.1 EUR/kg H₂ [...]” **Lines 16-17.**

Introduction

Comment #7

Introduction: Seems fine

Response to Comment #7

We appreciate your comments on the last round of revision. We ensure that the introduction provides sufficient background, clear statement of research gaps and our contributions to help readers understand our study and explore future research questions.

Main

Comment #8

When you say that electrolytic hydrogen is competitive except with zero caps (for example) make sure to remind the reader under which these conclusions are valid (i.e. constant ammonia production with non-flexible operation, expensive on-site H₂ storage –opposed to cheap large scale H₂ cavern storage-, ...). Because off-grid solutions are probably not that expensive if it is assumed that ammonia plant can run flexibly.

Response to Comment #8

Thank you for highlighting this point. We recognize the importance of explaining from the very beginning that our model assumes a continuous hydrogen output to ensure the uninterrupted operation of the ammonia plant. Consequently, we have included the following paragraph at the beginning of the Main section to address this clarification: “This study assumes that the existing European ammonia plants will be retrofitted by replacing the SMR production system with an electrolytic hydrogen production system (hereafter EHPS). The other subsystems of the ammonia plant, including the air separator unit (ASU), the ammonia synthesis loop, and the cryogenic storage for ammonia, will be maintained as they are in the existing facility. Consequently, the EHPS is designed to deliver **a continuous hydrogen supply thanks to on-site hydrogen storage in pressurized tanks and grid backup**, ensuring uninterrupted operation of the ammonia plant, which requires **steady-state conditions** (detailed information Methods, subsection “Ammonia production process and EHPS”).” **Lines 129-136.**

We also modified the caption of Fig. 1, including the following sentence: “The ammonia plant is assumed to operate continuously, at full load. Hence the EHPS must supply a constant volume of hydrogen D_{H_2} to the ammonia synloop.” **Lines 152-153.**

Additionally, we modified the title of the subsection: “**Electrolytic hydrogen can be cost-competitive with SMR except for the 0-cap when maintaining continuous production.**” **Lines 209-210.**

With these modifications, we clarify that our main results are based on the assumption of continuous, full-load operation of the ammonia plant.

Comment #9

Please be a bit more specific on the emissions from the grid: are you using marginal or average emissions? And how do you justify that choice? Because using one accounting method or another may have a significant impact on the results. It would be worth adding that point in the discussion.

Response to Comment #9

Thank you for your comment. In this study, we use the “average life-cycle emissions factor for electricity from the grid”. These life-cycle emission factors include the emission from electricity production and the emission of fuel supply. This is consistent with the EU methodology of quantifying the greenhouse gas emissions of hydrogen. We further explained this in the revised Main section: “Well-to-gate emissions encompass Scope 1 (direct emissions from operations, negligible in electrolytic production), Scope 2 (indirect greenhouse gas emissions from the generation of purchased electricity), and partial Scope 3 emissions (upstream activities like the extraction, refining, and transport of fuel used for electricity production, hereafter “Scope 3 upstream emissions”) (see Methods, subsection “Grid electricity price and carbon intensity”). **Hence, following EU regulations, the emissions from the grid considered in the study are based on the average carbon intensity of electricity consumed in the Member State where the fuel is produced⁴¹.**” **Lines 176-182.**

The “marginal life-cycle emission factor” is useful for quantifying the increase in grid emissions resulting from the additional electricity consumption by electrolytic hydrogen used in ammonia production. We pointed out the importance of considering in real-world implementation marginal emissions, especially in regions with greater impact on local electricity demand: “Nevertheless, four plants in Hungary (HU21 and HU31), the Netherlands (NL34), and Greece (EL51) would result in a local demand increase of over 15%, with the Dutch plant in Zeeland (NL34) potentially reaching up to 24.5%. While the overall energy demand of the grid may not be significantly affected, the peak power demand during operation can be substantial³⁷ (see Methods, “Feasibility analysis”). There is a need to investigate how regional or local grids will adapt to the increased demand of the EHPS. This includes the necessary expansion of power generation capacity

and the adjustment of pricing policies to ensure that hydrogen production via electrolysis remains cost-effective. **In these analyses, the marginal grid emission factors should be used in optimizing the expansion of power generation capacity and grid dispatch to accurately evaluate and minimize the emissions resulting from the EHPS-induced demand^{56,57}.** Lines 553-562.

Comment #10

I already made that comment in the first version, but I think that your changes in the text could be improved a little more.

What is the total share of electricity provided by the grid? If the grid consumption is high due to the additional load of multiple electrolyzers, with which production units this extra electricity demand will be covered? Do we need to build more i.e. gas turbines (stable load to ensure continuous ammonia production)? Or the existing grid is sufficient? You discuss this point in this paragraph: “The transition to electrolytic hydrogen is challenging for the 0-cap” but you would gain to explain it a bit more clearly (and relate this topic with marginal/average grid emissions). Maybe you could also change the title of this subsection to something related to “additional grid capacity needed”.

Response to Comment #10

Thank you for your comment. The total proportion of electricity supplied by the grid is, indeed, low: “The grid reliance drops to 16% under the 1-cap, and predictably to zero when the plant operates off-grid (SI – Fig. iv). Grid imports are prevalent during prolonged periods with suboptimal renewable energy conditions.” Lines 354-356.

We acknowledge that: “the shift to electrolytic hydrogen production does increase the electricity demand compared to the SMR process (on average, 56 MWh are imported from the grid each hour). While this increased electricity consumption necessitates infrastructure upgrades on the plant side, as previously discussed, the **plant's grid energy demand is relatively small**, representing, on average, **6.3% of the current energy demand at the regional level and 0.8% at a national level**. This increase is unlikely to have a significant impact on regional consumption patterns, causing grid congestions, or to influence price dynamics notably³⁷.” Lines 543-549.

In this second revision, we added the following paragraph to clarify the impact of this additional grid demand: “According to the EU's 2030 and 2050 targets, it is anticipated that future expansions of the grid will be predominantly accommodated by a greater proportion of renewable energy sources, complemented by widespread adoption of utility-scale storage technologies⁵⁴.” Lines 549-552.

Nevertheless, as explained in Response to Comment#9, we also pointed out that in real-world projects grid expansion and its implication on grid carbon intensity should be studied, especially in regions with higher-than-average impact on grid: “Nevertheless, four plants in Hungary (HU21 and HU31), the Netherlands (NL34), and Greece (EL51) would result in a local demand increase of over 15%, with the Dutch plant in Zeeland (NL34) potentially reaching up to 24.5%. While the overall energy demand of the grid may not be significantly affected, the peak power demand during operation can be substantial³⁷ (see Methods, “Feasibility analysis”). There is a need to investigate how regional or local grids will adapt to the increased demand of the EHPS. This includes the necessary expansion of power generation capacity and the adjustment of pricing policies to ensure that hydrogen production via electrolysis remains cost-effective. **In these analyses, the marginal grid emission factors should be used in optimizing the expansion of power generation capacity and grid dispatch to accurately evaluate and minimize the emissions resulting from the EHPS-induced demand^{56,57}.** Lines 553-562.

We modified the title of the subsection from “The transition to electrolytic hydrogen is challenging for the 0-cap” to “**The challenges in transitioning to electrolytic hydrogen**” to clarify that each emission cap comes with different challenges, including potential grid impact for less stringent caps. Moreover, we also modified the definition of the three feasibility dimensions considered: “The analysis considered three key factors: **additional grid capacity needed**, renewable energy needs, and land requirement in each plant’s region [...]” **Lines 538-540.**

We trust that these changes have enhanced the paragraph and its clarity.

Comment #11

As a general comment, I think that you can expand a bit more the comparison of your results with what has been found in the literature.

Response to Comment #11

Thank you for your comment, which encouraged us to further expand the literature review for cross-validation of our findings as well as highlight our novel contributions.

Our findings are consistent with those in the existing literature, particularly in terms of the cost range for producing electrolytic hydrogen and ammonia: “This range aligns with recent estimates from IEA²² (2-10 EUR/kg H₂ in Europe). **Assuming continuous operation of ammonia plants, these values translate into an ammonia production cost ranging from around 700 EUR/ton NH₃ under less stringent emission caps to 1200 EUR/ton NH₃ for off-grid plants. These estimates are consistent with cost projections found in other studies²².**” **Lines 215-218**, and the observation that off-grid plants require increased capacity in renewable installations and electrolyzers, leading to higher costs compared to semi-islanded plants: “These figures underscore the substantially higher resource requirements for fully off-grid setups (SI – Fig. iv). **These results align with findings from other case studies. For instance, Campion et al.³¹ demonstrated that in Northern Chile, a region dominated by solar energy, the installed capacity of PV and electrolyzers is more than twice as high for off-grid plants compared to semi-islanded configurations.**” **Lines 308-312.**

Additionally, we confirm the potential benefits of flexible production in off-grid settings: “The observed decrease in LCOH for flexible off-grid operations closely aligns with the estimated 25%-40% reduction previously reported in Australian, Argentina, and Chile case studies^{23,31,50}. The reduction is especially pronounced in regions dependent on wind energy, with a reduction of up to 46%, while solar-dominated regions experience a smaller impact, with a 19% reduction (SI – Table x).” **Lines 520-524.**

However, differently from previous studies, our research provides a comprehensive investigation into how emission caps influence the economic and technical viability of transitioning to electrolytic hydrogen in Europe. We consider various emission targets, assumptions for technology costs and performance, as well as diverse renewable conditions, by conducting an analysis on a continental scale. To underscore both the corroboration of our findings with established research and the novel contributions of our work, we included several paragraphs in this second round of revision.

We have enhanced the introduction to emphasize our work’s contribution more clearly: “While previous research has extensively analyzed trade-offs between the technical, economic, and environmental feasibility of electrolytic hydrogen production, the influence of emission caps on these trade-offs has not been

considered. Determining low-carbon hydrogen emission standards for the ammonia industry is complex since plants are spread across various regions, each with distinct cost components, electricity prices, grid emissions, and renewable potential. Accounting for this variation is vital to understand how the stringency of emissions targets affects the size, cost, and land use of low-carbon ammonia plants. However, studies that consider regional variations in renewable energy potential and costs, employ high-resolution analysis of renewable energy profiles and plant operations, and consider future advancements in technology, are currently missing. It is therefore not well understood how regional conditions, including renewables resource profiles and grid emission intensities, shape the relationship between emission standards and costs, particularly as these emissions standards approach zero. Non-linear relationships may lead to outsized costs, grid congestion, extensive renewables curtailment, and land scarcity, impacts that could be avoided with more deliberate, model-informed policy designs.” **Lines 105-118.**

In the revised Main, we highlighted that “These results corroborate previous research conducted at select sites in Europe²⁷ and the United States⁴⁷, which found that producing hydrogen via water electrolysis, when powered by a combination of dedicated renewable energy sources with a grid backup for uninterrupted plant operations, is generally more cost-effective than using exclusively additional renewable energy sources and produces fewer carbon emissions than relying solely on grid electricity. The degree of these benefits is influenced by factors such as the cost of electricity from the grid, the grid's carbon intensity, and the availability of renewable resources.

By assessing the implications of increasingly stringent emission targets, this research enhances the comprehensive understanding of the variability in LCOH across different European locations. The subsequent sections detail the findings, which are essential for defining suitable emission standards necessary for guiding the ammonia industry's shift toward low-carbon electrolytic hydrogen production.” **Lines 264-274.**

In the revised Main, subsection “Robustness analysis”, we also highlighted that “The robustness analysis revealed the impact of grid electricity price, electrolyzer technology selection, and flexible operation of ammonia production on the LCOH across a variety of emission caps and different degrees of renewable energy availability. These insights deliver a thorough assessment of the economic viability for grid-connected (non-zero caps) and off-grid (0-cap) electrolytic hydrogen and ammonia production. This enhanced understanding fills a gap in the literature²³ and informs strategic decision-making and policy development for sustainable transition towards net-zero electrolytic hydrogen and ammonia production.” **Line 529-535** to emphasize our new contributions.

We hope that with this revision, it will be easier for readers to contextualize our work within the existing literature and identify the contributions of our study.

Discussion

Comment #12

You say: “It is crucial to underscore that a 0-cap scenario not only implies higher costs and increased land usage for renewables but can also potentially lead to higher 566 overall lifecycle emissions compared to scenarios with less stringent caps.” Do you also include scope 3 emissions when calculating the grid carbon intensity? I may have missed it, but I did not find that information.

Response to Comment #12

Thanks for the reviewer’s question. Our analysis includes partial Scope 3 emissions (upstream activities for electricity production) but not Scope 3 embedded emissions in technology manufacturing: “Well-to-gate

emissions encompass Scope 1 (direct emissions from operations, negligible in electrolytic production), Scope 2 (indirect greenhouse gas emissions from the generation of purchased electricity), and **partial Scope 3 emissions (upstream activities like the extraction, refining, and transport of fuel used for electricity production, hereafter “Scope 3 upstream emissions”)** (see Methods, subsection “Grid electricity price and carbon intensity”).” **Lines 176-180.** Further details were also included in the last revision within the subsection "Scope 3 embedded emissions" in the Methods section.

In the Discussion section, following the suggestions received from the first round of revisions, we have incorporated a paragraph that addresses the potential impact of Scope 3 embedded emissions related to the manufacturing and installation of technologies. In this second round of revisions, we have revised that section to improve clarity: “It is crucial to underscore that a 0-cap scenario not only implies higher costs and increased land usage for renewables but can also potentially lead to higher overall lifecycle emissions compared to scenarios with less stringent caps **when further accounting for emissions associated with the manufacturing of components including electrolyzers, wind turbines, PV, and hydrogen storage systems.**” **Lines 643-646.**

Thank you for the review’s comment. We hope that our modification of this sentence is clearer to the readers.

Reviewer #3:

Comment #1

Upon reviewing the revised manuscript, you have addressed the previous comments with diligence and attention to detail. Your efforts to refine the manuscript have not only enhanced its clarity but also enriched the overall quality of the work. The revisions have satisfactorily resolved my concerns raised, and it is evident that a considerable amount of thought and work went into making these changes. I appreciate your responsiveness to the review process and commend you on your dedication to improving the manuscript.

Response to Comment #1

Thank you for your encouraging comments on our revised manuscript. The whole team is pleased to see our hard work reflected in the improvements you have acknowledged. We really appreciate the valuable feedback from all reviewers, as it has greatly enhanced our paper.

Comment #2

Code notes available. Webpage 404, page not found error.

Response to Comment #1

Thanks for bringing this up. In the revised manuscript, we have included a publicly accessible link (<https://github.com/hkust-suscity/Electrolytic-ammonia-production-in-Europe>) to the GitHub page hosting a repository with all the code necessary to replicate our main analysis and sensitivity analysis, along with all the input data required to reproduce our results. We believe this enhancement will greatly improve the usability and transparency of our modeling framework, thereby aiding other researchers in utilizing and improving our work.

REVIEWERS' COMMENTS

Reviewer #1 (Remarks to the Author):

Very high quality article suitable for publication. You have addressed all the last comments I had.
Congrats for your hard work!

RESPONSE TO REVIEWERS' COMMENTS (THIRD ROUND OF REVIEW)

Reviewer #1:

Comment #1

Very high quality article suitable for publication. You have addressed all the last comments I had. Congrats for your hard work!

Response to Comment #1

We are truly pleased to hear that the reviewer has acknowledged the hard work put into this manuscript. We extend our sincere thanks for the constructive comments and feedback that have significantly contributed to the enhancement of our paper's quality and rigor.